# A Comprehensive Review of Organochlorine Pesticide Monitoring in Agricultural Soils: The Silent Threat of a Conventional Agricultural Past

**Evangelia N. Tzanetou †and Helen Karasali *,†**

Laboratory of Chemical Control of Pesticides, Scientific Directorate of Pesticides' Control and Phytopharmacy, Benaki Phytopathological Institute, 8 St. Delta Street, Kifissia, 14561 Athens, Greece; ev.tzanetou@bpi.gr
* Correspondence: e.karassali@bpi.gr; Tel.: +30-210-8180-314
† These authors contributed equally to this work.

**Abstract:** Soil constitutes the central environmental compartment that, primarily due to anthropogenic activities, is the recipient of several contaminants. Among these are organochlorine pesticides (OCPs), which are of major concern, even though they were banned decades ago due to their persistence and the health effects they can elicit. In this review, an overview of monitoring studies regarding OCPs in soils published over the last 30 years along with the development of analytical methods and extraction procedures for their determination in soil are presented. The presented synopsis verifies the soil contamination by OCPs during the last several decades. Soil pollution by OCPs should be an essential aspect of the characterization of whole soil quality, considering that a significant percent of soils on a global scale are in the borderline of suitability for cultivation and pertinent activities. The latter, to an extent, is attributed to the presence of organic contaminants, especially those of persistent chemical natures.

**Keywords:** soil monitoring; degradation products; analytical methods; extraction; OCP occurrence

## 1. Introduction

Organochlorine pesticides (OCPs) are persistent organic pollutants (POPs) extensively used in agriculture to control insect pests in a broad variety of crops. POPs are a cluster of toxic, bio-accumulative, bio-magnified, and persistent compounds with a likelihood of long-distance movement in the environment [1]. OCPs are synthetic compounds with boundless chemical stability. They are considered as egregious environmental contaminants responsible for ecological instability around the globe [2]. OCPs, such as dichlorodiphenyltrichloroethane (DDT) and dieldrin, were among the first synthetic insecticides developed and used worldwide. Although their use was discontinued worldwide, their persistence and their extensive historical use has left numerous sites with raised soil concentrations, which require remediation.

OCPs are hydrophobic compounds with very high adsorption coefficients, meaning a considerable number of them can be adsorbed and strongly bound to soil particles through agricultural procedures [3]. They remain in the surface layers of soils upon adsorption without leaching down the soil profile and persist in the soil, having half-lives ranging from months to years [4].

OCPs were the leading chemicals used in the control of brown Muridae species, which are the most destructive insect pests in cocoa trees [2]. Despite being banned worldwide, OCPs are still available in many countries via illegal routes [3]. High global demands for OCPs in agricultural practice, in opposition to environmental regulations, were due to their excellent efficacy in pest control and cost-effectiveness [5].

OCPs, including hexachlorocyclohexanes (HCHs), DDTs, aldrin, dieldrin, endrin, chlordane, heptachlor, and hexachlorobenzene (HCB), have been related to causing cancer,



injury to the nervous system, generative disorders, and disturbance of the immune system in humans [6]. Some of them are extremely toxic and have a large variety of chronic effects, including endocrine dysfunction, mutagenesis, and carcinogenesis, while others are supposed to act as endocrine disruptors affecting hormone function [7]. In 2001, the Stockholm Convention on Persistent Organic Pollutants (POPs) was signed and came into force in 2004. OCPs were listed in the Stockholm Convention as persistent organic pollutants (POPs) that are to be banned by the United Nations Environment Program (UNEP) [8,9]. The goal of this convention was to protect human health and the environment from POPs by the banning or restriction of their production.

Aldrin, HCB, chlordane, dieldrin, endrin, heptachlor, mirex (MRX), and toxaphene were obliterated internationally under the 2001 Stockholm Convention. Furthermore, chlordecone and HCHs were added to the convention in 2009, whereas endosulfan was added later in 2013. DDT remains available for vector control, as it has been approved by the World Health Organization, but is otherwise banned. In agriculture, OCPs have acted as insecticides, acaricides and fumigants to control pests in a variety of crops. In the field of public health, they have played a pivotal role in eliminating certain parasitic diseases such as malaria [10,11].

Environmental pollution with OCPs may be associated with point sources (industrial emissions and waste plant effluents) or, more commonly, with diffuse sources (atmospheric transport and deposition), which are the most important pathways for their transportation to distant sites [12].

Soils and sediments possess various microenvironmental conditions that impact air and water exchange and post-depositional procedures. Soil pollution increases worries regarding soil utilities, biodiversity, and food security but also regarding the off-site transportation of pollutants via wind- and water-forced erosion. Such off-site transportation may harm the function of sink ecosystems and correspond to further exposure paths to soil pollutants for humans and other non-target organisms [13].

Regardless of the numerous consequences of soil pollution, the monitoring of pesticide residues in soil is not required in many countries, in contrast to water monitoring [13]. Furthermore, large-scale worldwide studies on soil pollution via pesticide residues are rare and are often restricted to one single pesticide or to only a few compounds [14]. Various studies have previously described the allocation of currently used and of no-longer-approved pesticides in soil at the national or regional levels, but the various sampling periods, several sampling strategies, numerous analytical methods, and various analyte lists among these studies avert a complete impression of the distribution of pesticides residues worldwide [12–17].

From soil-monitoring programs and studies, those with the theme of OCP monitoring in soil, the manufacturing of which is totally banned globally, have been brought into consideration. Subsequently, there were no other complete studies on OCP soil monitoring comprising extensive characteristic sampling locations in American countries. In Europe, monitoring surveys were started in 1990 in Spain [17]. In the following years, outcomes from more complete studies were issued, which represented the whole EU [13]. In Asia, and especially in China and India, an adequate number of monitoring studies have been implemented, which was only a few considering the vast expanse of the countries and the continent. In Africa, fewer monitoring results are available than in other areas, as studies have only covered limited positions. With regard to Oceania, no published studies have been identified.

### 1.1. Research Methodology—Inclusion/Exclusion Criteria

For initial screening, relevant data were collected based on the existence of the terms "organochlorine pesticides" and "soil" in the article title, abstract, and keywords of Scopus. As a consequence, one thousand nine hundred and fourteen potentially eligible articles were identified. Afterwards, forty-five records were identified through personal collection, citation, and Google searches. The total number of findings was

reduced following the removal of duplicate articles and scanning of their eligibility. Furthermore, the literature search was limited to publications written in English between 1991 and 2021. Information on each paper was extracted, such as the type and number of OCPs, the extraction process, the analytical method used for their determination in soil, acceptable validation data, and monitoring results. Finally, from reading the full text, one hundred and sixty-four articles and ten books were deemed eligible for this review. At this stage, all articles that were included addressed soil contamination by OCPs and their analysis via several extraction procedures. An overview of the inclusion and exclusion criteria is given below, in Figure 1.

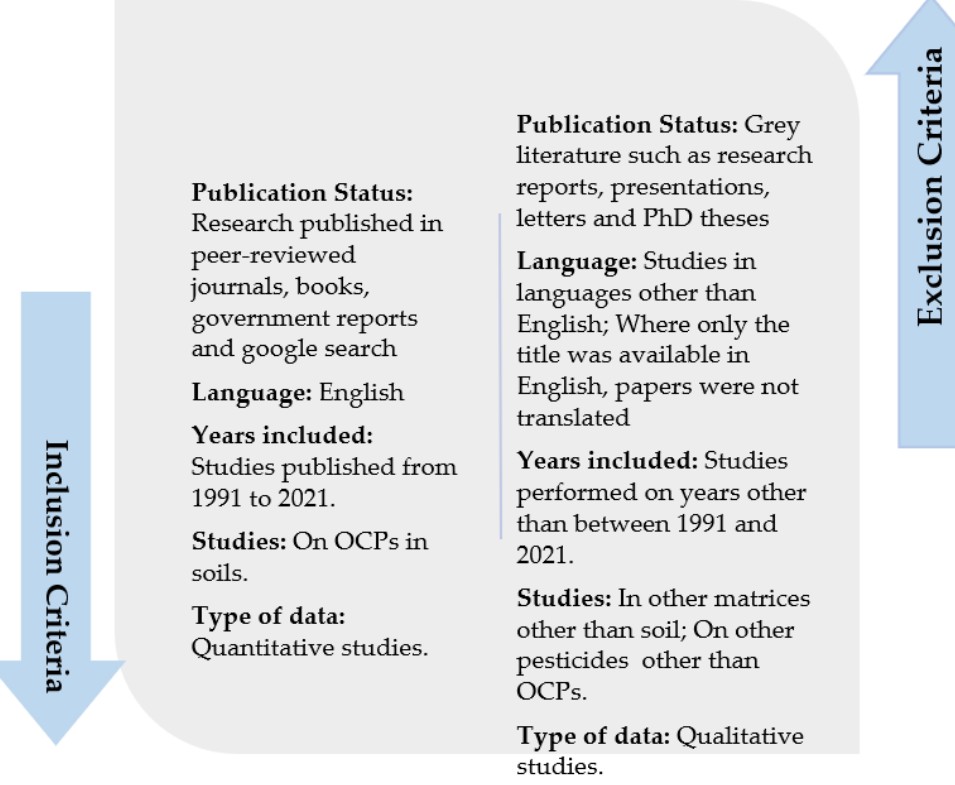

**Figure 1.** Overview of inclusion and exclusion criteria for the systematic review.

### 1.2. General Overview of Organochlorine Pesticides

Organochlorine pesticides (OCPs) belong to the chemical classes of chlorinated cyclodienes, chlorodiphenylethanes, chlorinated benzenes, and cyclohexanes [18]. Dicofol (DCF), DDT, dichlorodiphenyldichloroethylene (DDE), dichlorodiphenyldichloroethane (DDD), metolachlor, and perthane belong to the chemical class of dichlorodiphenylethanes, whereas aldrin, dieldrin, endrin, chlordane, endosulfan, and heptachlor belong to the class of chlorinated cyclodienes. Chlordecone, HCH, also known as benzene hexachloride (BHC), hexachlorobenzene (HCB), MRX, and toxaphenes are chlorinated benzenes and cyclohexanes [18].

These substances belong to the class of persistent organic pollutants (POPs) with high persistence in the environment, having long half-lives ($DT_{50}$) from 60 days to 12 years in soils and sediments [19,20].

They are usually hydrophobic and persistent compounds, and they are resistant to photolytic, biological, and chemical degradation [21]. Their lipophilic ability permits them to be connected to fatty tissues in both animal and human bodies. OCPs may be found in higher concentrations in some tissues, such as liver or kidney, because of their lipophilic nature leading to bioconcentration [22]. Their lipophilicity is also responsible for their persistence in the environment through accumulation in sediment, soil, and plants. Due to

their low cost, OCPs such as DDT, HCH, aldrin, and dieldrin are among the most widely used pesticides in developing countries in Asia.

OCPs come into the environment through various routes, such as their application, the dumping of industrial waste into landfills, and their release from manufacturing plants [18]. As they can travel long distances before deposition on soils, OCPs can be detected hundreds or thousands of miles away from their application points [23]. Various OCPs are volatile, while others are strongly absorbed in soil particles [24]. They may also be taken up by vegetation or penetrate soil and consequently contaminate groundwater [25]. In aquatic ecosystems, OCPs can adsorb or desorb on solids and further mitigate in bottom sediments, where they bioaccumulate in fish and other aquatic organisms. DDTs and dieldrin persist in soil for decades and thus ultimately enter the food chain because of adsorption [26].

### 1.3. Context of This Review

In this article, a comprehensive review of soil monitoring studies regarding OCPs is presented; therefore, the review combines all accessible data to gather past and current worldwide information on their residue in agricultural soils. The worldwide status of agricultural soils from the viewpoint of pollution by OCPs was discussed. Finally, the methodological approaches and strategies of the studies regarding the soil monitoring of OCPs were briefly reviewed to attain a critical understanding of weakly substantiated facts and gaps which need to be filled in future studies. There were plenty of studies which placed emphasis on the analysis of OCPs yet comprised soil monitoring results as part of their method validation process. Among them, only those with a proper quantity of samples and wider sampling locations were included. Those with an inadequate number of samples and sampling sites were omitted since they were not sufficient to represent the soil status of an entire region.

### 1.4. Practices Used in Soil Monitoring and Surveys of OCPs

In this review, more than 59 monitoring studies and investigations performed all over the world and released in the last 32 years were considered. Due to the absence of an established worldwide monitoring protocol, it was anticipated that the methods and instrumental processes used would not be homologous across all the studies. Although the key objective of this paper is to review the monitoring results, to enable accurate understanding of the results, we also reviewed the methods and procedures of obtaining results in reference studies. This way, the reader can recognize the method by which the results were obtained in each reference study, as there was no unified method of obtaining data in all the studies.

## 2. Determination of Organochloride Pesticide Residues in Soil

With the signing of the Stockholm Convention on POPs and the development of monitoring programs, there is an increased need for fit-for-purpose laboratories, particularly in developing countries, to identify and detect such obsolete organic chemicals with persistent characteristics. Among POPs, OCPs are an exemplary category with historic prevalence in a plethora of environmental compartments on a global scale.

It is obvious that the establishment of an analytical laboratory that would apply modern methodologies at currently acceptable international standards is a relatively expensive task. In practice, most laboratories can generally identify and quantify about 10–20 individual OCPs and their metabolites, regardless of the sample matrix, considering simple analytical systems. The availability of suitable analytical standards is a fundamental requirement, as the scarcity of analytical standards or standards of questionable quality can be a significant source of error in OCP analyses. Their standards could be provided from commercial chemical supply companies or agencies involved in the certification of reference materials.

Yet, the overall cost can limit the participation of scientists in developing countries in the context of internationally acknowledged quality control and interlaboratory schemes. This is evident from the relative lack of publications and information on POPs from some countries in Africa, South Asia, and South/Central America.

### 2.1. Analytes to Be Determined and Their Physico-Chemical Properties—A Step before Chemical Analysis

OCPs are synthetic chemicals that are not naturally present in the environment. They were introduced in the 1940s due to their effectiveness against various insects. Nine OCPs were initially listed in the so-called dirty dozens of POPs regulated by the Stockholm Convention on POPs in 2001 [9], including aldrin, dieldrin, endrin, DDT, chlordane, HCB (also classified as an industrial chemical), MRX, toxaphene, and heptachlor.

Beginning from this list of OCPs, technical chlordane is a chlorinated cyclodiene consisting of a complex mixture of isomers, chlorinated hydrocarbons, and by-products that is composed of at least 147 compounds. The composition varies with the manufacturing process. It contains trans-chlordane (TC also called β-chlordane) and cis-chlordane (CC also called α-chlordane) (43–75%), and lower levels of heptachlor (10–20%), cis- and trans-nonachlor, and chlordenes. The two major constituents are racemic in technical mixture but could undergo enantioselective degradation in the environment; therefore, the ratio of TC to CC has been used to indicate the degree of "weathering" in the environment [27]. Analytical standards, in high purities >95%, are available for the mixture, as well as for several major components (α-, β-, γ-chlordanes) [28,29], but not for all major components of the mixture, and this makes them the most difficult to determine.

Heptachlor is considered together with chlordane because of its close structural resemblance and since technical-grade products each contain about 10–20% of the other compound. Technical-grade heptachlor contains about 72% heptachlor and 28% related compounds (20–22% trans-chlordane and 4–8% nonachlor). Heptachlor epoxide is known as an impurity in the commercial heptachlor, which is expected to be rapidly degraded into heptachlor epoxide in the environment. However, analytical standards for heptachlor epoxide are not as widely available as heptachlor standards [30].

When one refers to DDT, they are generally referring to p,p′-DDT, which was produced and used for its insecticidal properties. However, technical-grade DDT is composed of up to 14 chemical compounds, of which only 65–80% is the active ingredient, p,p′-DDT. The other components include 15–21% of the nearly inactive o,p′-DDT, up to 4% of p,p′-DDD, and up to 1.5% of 1-(p-chlorophenyl)-2,2,2-trichloroethanol [31]. DDE and DDD are chemicals similar to DDT that contaminate commercial DDT, DDE, and DDD, which are also the major metabolites and environmental breakdown products of DDT with similar properties. Specifically, the (p,p′-DDE + p,p′-DDD)/ p,p′-DDT ratio indicates new or historical DDT input into soils. DDT, DDE, and DDD are sometimes collectively referred to as DDX [32].

The composition of technical-grade aldrin was reported to consist of 85.5% of the active substance. Isodrin and dieldrin, as well as other compounds, have been found as impurities in aldrin samples. Aldrin can also be rapidly converted in the environment to its epoxide dieldrin, while isodrin may form the epoxide endrin. Endrin is further degraded to form endrin aldehyde and endrin ketone. Isodrin and endrin are the endo–endo stereoisomers of aldrin and dieldrin. All of them are commercially available [33].

Technical-grade HCH typically contains 10–15% gamma HCH, known as lindane (LND), as well as the alpha (α), beta (β), delta (δ), and epsilon (ε) forms of HCH. Technical-grade LND, almost pure (>99%), is also used. All of them are commercially available in high purities and as a mixture. Most OCPs have optically active or chiral isomers (e.g., α-HCH, o,p′-DDT, the main constituents of technical chlordane, cis-/trans-chlordane, heptachlor, as well as chlorobornanes in toxaphene). While OCPs are racemic mixtures when manufactured, microbial degradation in soils can result in non-racemic patterns in environmental samples. Enantiomer fractions of several OCPs isomers can be used to

identify emissions of this pesticide from soils. Furthermore, mixtures of several OCPs in known concentrations are commercially accessible to be used as surrogates for achiral analysis [34].

Overall, OCPs present high lipophilicity, low polarity, high thermal stability, and volatility. The chemical stability of several OCPs or their metabolites is high because their molecules are constructed from C-C, C-H, and C-Cl bonds, which tend to be chemically inactive under normal environmental conditions. The physical and chemical properties of the most popular OCPs insecticides are listed in Table 1. Specifically, chemical structure, molecular weight, the logarithm of the octanol–water partition coefficient (log Kow), solubility, vapor pressure (VP), and the Henry's Law constant (H) are some of the properties selected to be presented below. As shown in Table 1, the majority of the OCPs are insoluble or slightly soluble in water. Furthermore, all OCPs have a log Kow > 3, which indicates that they have the tendency to be absorbed by the organic matter present both in soils and sediments. The vapor pressure is another important physicochemical property. Compounds with a high vapor pressure are generally volatile and may readily enter the atmosphere once applied in the field. Heptachlor is reported below as the most volatile. In addition, as compounds with high values of H will tend to volatilize, MRX pesticide could also be considered volatile.

**Table 1.** * Chemical and physical chlorinated pesticide compounds (OCPs) belonging to insecticide pesticide type and their physicochemical properties.

| Analyte; IUPAC Name; Molecular Formula, etc. | Molecular Weight (g mol$^{-1}$) | Molecular Structure | Regulatory Status: EC Regulation 1107/2009 Repealing 91/414 (Introduced/First Reported) | Soil Degradation DT 50 (Days) | Vapor Pressure (VP) at 20 °C (mPa) | Henry's Law Constant (H) at 25 °C (Pa m$^3$ /mol) | Octanol–Water Partition Coefficient, LogKow (pH 7, 20 °C) | Solubility in Water (mg L$^{-1}$)/ Solubility in Organic Solvents |
|---|---|---|---|---|---|---|---|---|
| DDT is an Unstated isomer mix containing roughly 75–85% p,p'-DDT, 10–15% o,p'-DDT, and a small amount of o,o'-DDT. Any balance is composed of transformation products DDE and DDD; 1,1,1-trichloro-2,2-bis(4-chlorophenyl)ethane; C$_{14}$H$_9$C$_{l5}$, CAS 50-29-3 p,p'-DDT major isomer; 1,1-dichloro-2,2-bis(4-chlorophenyl)ethane; o,p'-DDT, an isomer of DDT which is normally around 10–15% of the DDT isomeric mix; 1-chloro-2-(2,2,2-trichloro-1-(4-chlorophenyl)ethyl)benzene o,o'-DDT a minor isomer; 1-chloro-2-(2,2,2-trichloro-1-(2-chlorophenyl)ethyl)benzene | 354.49 | | Not approved (1944) | 6200 Other sources: 3 months in tropical regions, 4–30 years in temperate regions | 0.025 for DDT, 0.025–0.8 (at 20–25 °C) for p,p' DDT | $8.43 \times 10^{-1}$ for DDT 0.86–8.2 (at 20–25 °C for p,p-'DDT) | 6.91, 6.36 | 0.006 (at 20 °C for DDT), 0.025 (at 25 °C for p,p'-DDT): insoluble/readily soluble in aromatic and chlorinated solvents (e.g., ethyl ether, acetone, cyclohexanone, dichloromethane, benzene, and xylene) |
| DDD consists of three isomeric forms: p,p'-DDD, o,p'-DDD, and o,o'-DDD. p,p'-DDD is the dominant isomer: impurities or metabolites of technical DDT; 1,1-dichloro-2,2-bis(4-chlorophenyl)ethane; C$_{14}$H$_{10}$C$_4$,CAS 72-54-8 | 320.02 | | Not approved (1944) | 1000 | 0.18 | $4.0 \times 10^{-6}$ | 6.02 | 0.090 (at 20 °C): very slightly soluble. |
| p,p'-DDE impurities or metabolites of technical DDT; p,p' Dichloro diphenyl dichloroethane; CAS 72-55-9 | 318.02 | | No commercial use | 5000 | 0.024 | $2.1 \times 10^{-5}$ | 6.51 | 0.12 (at 25 °C): slightly soluble/lipids, many organic solvents |

Table 1. *Cont.*

| Analyte; IUPAC Name; Molecular Formula, etc. | Molecular Weight (g mol$^{-1}$) | Molecular Structure | Regulatory Status: EC Regulation 1107/2009 Repealing 91/414 (Introduced/First Reported) | Soil Degradation DT 50 (Days) | Vapor Pressure (VP) at 20 °C (mPa) | Henry's Law Constant (H) at 25 °C (Pa m$^3$ /mol) | Octanol–Water Partition Coefficient, LogKow (pH 7, 20 °C) | Solubility in Water (mg L$^{-1}$)/ Solubility in Organic Solvents |
|---|---|---|---|---|---|---|---|---|
| Endrin, known also as Aldrin epoxide; 1,2,3,4,10,10-hexachloro-6,7-epoxy-1,4,4a,5,6,7,8,8a-octahydro-exo-1,4-exo-5,8-dimethanonaphthalene; C$_{12}$H$_8$Cl$_6$O, CAS 72-20-8 | 380.91 |  | Not approved (1950s) | 4300 Other sources: up to 12 years; DT$_{50}$ 4–14 days (FAO) | $2 \times 10^{-7}$, other sources: 0.02720- (at 25 °C) | $1.48 \times 10^{-1}$, other sources: 0.05–0.76 (at 20–25 °C) | 3.2, 5.20 Other sources: 5.6, 5.34, 5.45 (calculated) | 0.24 (at 20 °C): slightly soluble. Acetone: 17 g/100 mL; benzene: 13.8 g/100 mL; carbon tetrachloride: 3.3 g/100 mg/L; hexane: 7.1 g/100 mL; xylene: 18.3 g/100 mL |
| Dieldrin, a chiral cyclodiene molecule, can be the metabolite of aldrin; (1R,4S,4aS,5R,6R,7S,8S,8aR)-1,2,3,4,10,10-hexachloro-1,4,4a,5,6,7,8,8a-octahydro-6,7-epoxy-1,4:5,8-dimethanonaphthalene; C$_{12}$H$_8$Cl$_6$O, CAS 60-57-1 | 380.91 |  | Not approved (1949) | 1400 Other sources: 800–900 days, 2.6–12.5 years | 0.024, other sources: 0.02–2.4, $3.1 \times 10^{-3}$ mmHg (at 20 °C) $5.89 \times 10^{-3}$ mmHg (at 25 °C) | $6.50 \times 10^{-2}$, other sources: 0.02–5.88 | 3.7, 3.69–6.2 | 0.14 (at 20 °C), 0.2 (at 20–25 °C): slightly soluble/moderately soluble in common organic solvents except aliphatic petroleum solvents and methyl alcohol |
| Chlordane is a chiral molecule. Chlordane is a complex mixture of isomers, other chlorinated hydrocarbons, and a range of by-products: 1,2,4,5,6,7,8,8-octachloro-2,3,3a,4,7,7a-hexahydro-4,7-methanoindene; C$_{10}$H$_6$Cl$_8$, CAS 57-74-9 | 409.78 |  | Not approved (circa 1950) | 350 Other sources: approx 1 year; 283 days to 3.8 years; 4 years (FAO) | 1.3 Other sources: cis-chlordane $2.2 \times 10^{-5}$ mmHg (crystal) $3.0 \times 10^{-6}$ mmHg trans-chlordane $2.9 \times 10^{-5}$ mmHg (crystal) $3.9 \times 10^{-6}$ mmHg | $0.39 \times 10^{-3}$, other sources: 2.92–9.5 | 2.78, 5.54 | 0.1–1.83 at 20 °C: slightly soluble/miscible: miscible with most aliphatic and aromatic organic solvents; ethanol, cyclohexane, and isopropanol, including acetone |
| Heptachlor is a molecule with 5 chiral centers: 1,4,5,6,7,8,8-heptachloro-3a,4,7,7a-tetrahydro-4,7-methanoindene; C$_{10}$H$_5$Cl$_7$, CAS 76-44-8 | 373.32 |  | Not approved (1951) | 285 Other sources: 9–10 months, about 2 years | 53 highly volatile | $3.53 \times 10^2$, 112–845 volatile | 5.44, 4.4–5.5 | 0.056 at 20 °C: very slightly soluble/soluble in many organic solvents, e.g., in acetone 75, benzene 106, xylene 102, cyclohexanone 1190, carbon tetrachloride 1130, and ethanol 450 |

**Table 1.** *Cont.*

| Analyte; IUPAC Name; Molecular Formula, etc. | Molecular Weight (g mol$^{-1}$) | Molecular Structure | Regulatory Status: EC Regulation 1107/2009 Repealing 91/414 (Introduced/First Reported) | Soil Degradation DT 50 (Days) | Vapor Pressure (VP) at 20 °C (mPa) | Henry's Law Constant (H) at 25 °C (Pa m$^3$ /mol) | Octanol–Water Partition Coefficient, LogKow (pH 7, 20 °C) | Solubility in Water (mg L$^{-1}$)/ Solubility in Organic Solvents |
|---|---|---|---|---|---|---|---|---|
| HCH: $C_6H_6Cl_6$, CAS 608-73-1 chiral, exists as eight or more isomers: 60–70% alpha-isomer (α-HCH); 5–12% beta-isomer (β-HCH), 6–10% delta isomer (δ-HCH), 3–4% epsilon isomer (ε-HCH), 10–15% gamma isomer (γ-HCH), etc. | | | | | α-HCH = 3–6 β-HCH = 0.04–0.12 δ-HCH = 0.02–0.08 γ-HCH = 1–21.3 | | 3.80 α and β isomer, 4.14 δ-HCH, 3.5 γ-HCH | 8.52 (25 °C): 8.35 (pH 5, 25 °C). In acetone > 200, methanol 29–40, ethyl acetate < 200, and n-heptane 10–14 (all at 20 °C) |
| α-HCH: 1-alpha, 2-alpha, 3-beta, 4-alpha, 5-beta, 6-beta-benzene-transhexachloride; CAS 319-84-6 | | | | | 5.99, $4.5 \times 10^{-5}$ mmHg at 25 °C | $6.86 \times 10^{-6}$ | 3.8 high | 69.5 mg/L at 28 °C: moderately soluble |
| β-HCH: 1-alpha, 2-beta, 3-alpha, 4-beta, 5-aplha, 6-beta-exachlorocyclohexane; CAS 319-85-7 | 290.82 | | | | 0.029, $3.6 \times 10^{-7}$ mmHg at 20 °C | $4.5 \times 10^{-7}$ | 3.78 high | 2.41/- |
| δ-HCH: 1-alpha,2-alpha,3-alpha, 4-beta, 5-alpha, 6-beta-hexachlorocyclohexane; CAS 319-86-8 | | | | | $3.5 \times 10^{-5}$ mmHg at 25 °C | $2.1 \times 10^{-7}$ | 4.14 | |
| γ-HCH or LND: 1α,2α,3β,4α,5α,6β-hexachlorocyclohexane; $C_6H_6Cl_6$, CAS 58-89-9 | | | Not approved (circa 1945; first prepared in 1825) | 400–980 | 4.4, $4.2 \times 10^{-5}$ mmHg at 20 °C | $1.483 \times 10^{-6}$, $3.5 \times 10^{-6}$ | 3.50, 3.72 | 7.3 at 25 °C; 8.52 at 20 °C: slightly soluble/readily soluble in acetone, benzene, methanol, and ethyl acetate |

Table 1. *Cont.*

| Analyte; IUPAC Name; Molecular Formula, etc. | Molecular Weight (g mol$^{-1}$) | Molecular Structure | Regulatory Status: EC Regulation 1107/2009 Repealing 91/414 (Introduced/First Reported) | Soil Degradation DT 50 (Days) | Vapor Pressure (VP) at 20 °C (mPa) | Henry's Law Constant (H) at 25 °C (Pa m$^3$/mol) | Octanol–Water Partition Coefficient, LogKow (pH 7, 20 °C) | Solubility in Water (mg L$^{-1}$)/ Solubility in Organic Solvents |
|---|---|---|---|---|---|---|---|---|
| Endosulfan is an isomer mixture of alpha- and beta-endosulfan; 1,4,5,6,7,7-hexachloro-8,9,10-trinorborn-5-en-2,3-ylenebismethylene sulfite; $C_9H_6Cl_6O_3S$, CAS 115-29-7, CAS [959–98-8] α-endosulfan, CAS [891–86-1] β-endosulfan | 406.93 |  | Not approved (circa 1956) | 50, other sources: 28–50, 62–126, 68–87; 60–800 (FAO) | 0.83 mixture of α- and β-isomers 2:1), 0.28–1.47 at 20–25 °C | 1.48 (α-isomer) 0.07 (β-isomer) (22 °C) | 4.74 α-isomer (pH 5) 4.79 β-isomer (pH 5) | 0.32 at 20 °C: slightly soluble/ readily soluble in ethyl acetate, dichloro-methane, toluene 200, ethanol 65, and n-hexane 24 |
| Isodrin is an isomer of aldrin. Isomeric. The 5S,8R isomeris known as aldrin (1R,4S,5R,8S)-1,2,3,4,10,10-hexachloro-1,4,4a,5,8,8a-hexahydro-1,4:5,8-dimethanonaphthalene; $C_{12}H_8Cl_6$ CAS 4 65-73-6 | 364.91 |  | Not approved (circa 1940s) | Very persistent | 5.866, 10.35 | 39.21 | 6.75 | 0.014 at 20 °C: very slightly soluble. |
| Aldrin (ALD) is a chiral molecule; aldrin is one of the several isomers of hexachlorohexahy-drodimethanonaphthalene; (1R,4S,4aS,5S,8R,8aR)-1,2,3,4,10,10-hexachloro-1,4,4a,5,8,8α-hexahydro-1,4:5,8-dimethanonaphthalene; $C_{12}H_8Cl_6$, CAS 309-00-2 | 364.91 |  | Not approved (circa 1950) | 365 Other sources: 20–100 days (FAO) | 8.6 at 20 °C, 0.9–3.1 at 20–25 °C, $7.5 \times 10^{-5}$ mmHg at 20 °C, $1.2 \times 10^{-4}$ mmHg at 25 °C, 8.6 at 20 °C | 17.2 at 25 °C, 4.46–91.23 at 20–25 °C | 6.50, 5.17–7.4, 6.82 | 0.027 at 20 °C, 0.017 at 20–25 °C: very slightly soluble/ moderately to very soluble in most aromatic hydrocarbons, esters, ketones, and halogenated solvents: acetone, benzene, and xylene |
| MRX: dodecachloropentacyclodecane; $C_{10}Cl_{12}$, CAS 2385-85-5 | 545.54 |  | Not approved (1946, first reported) | 300 Other sources: 3000, up to 10 years | 0.11 at 20–25 °C, $3 \times 10^{-7}$ mm Hg (at 25 °C) | 839.4 at 25 °C volatile, $8.11 \times 10^{-4}$ atm-m$^3$/mole | 6.89, 5.28 (pH 7, 20 °C) | 0.085 at 25 °C, 0.0001 at 20 °C: insoluble/very soluble in dioxane, benzene chloroform, and xylene |

**Table 1.** *Cont.*

| Analyte; IUPAC Name; Molecular Formula, etc. | Molecular Weight (g mol$^{-1}$) | Molecular Structure | Regulatory Status: EC Regulation 1107/2009 Repealing 91/414 (Introduced/First Reported) | Soil Degradation DT 50 (Days) | Vapor Pressure (VP) at 20 °C (mPa) | Henry's Law Constant (H) at 25 °C (Pa m$^3$ /mol) | Octanol–Water Partition Coefficient, LogKow (pH 7, 20 °C) | Solubility in Water (mg L$^{-1}$)/ Solubility in Organic Solvents |
|---|---|---|---|---|---|---|---|---|
| HCB: perchlorobenzene; $C_6Cl_6$, CAS 118-74-1 | 284.80 |  | Not approved (circa 1947) | 2.7–7.5 years | 1.45 | 10.3 | 3.93 | 0.0047 at 20 °C: insoluble. |
| Chlordecone is a chiral molecule; perchloropentacyclodecan-5-one; $C_{10}Cl_{10}O$, CAS 143-50-0 | 490.64 |  | Not approved (first reported 1952, 1966 commercial production) | - | $3.5 \times 10^{-5}$ | $2.53 \times 10^{-3}$ | 4.5 | 3.0 at 20 °C: slightly soluble/slightly soluble in hydrocarbon solvents; soluble in alcohols, ketones, and acetic acid |
| DCF is chemically related to DDT and used as acaricide.; 2,2,2-trichloro-1,1-bis(4-chlorophenyl)ethanol; $C_{14}H_9Cl_5O$, CAS 115–32-2 | 370.5 |  | Not approved (1956 first reported, 1957 commercial production) | 40–80, other source: 95 days (FAO) | 0.25, $5.3 \times 10^{-3}$ at 25 °C | $2.45 \times 10^{-2}$, $5.7 \times 10^{-5}$ | 3.5–4.3 | 0.8 at 25 °C: insoluble/soluble in organic solvents: toluene 400, methanol 36, and isopropanol 30 |
| Toxaphene or Camphechlor a chiral molecule; reaction mixture of chlorinated camphenes containing 67–69% chlorine; CAS name:2,2,5-endo,6-exo,8,8,9,10-Octachlorobornane; $C_{10}H_{10}C_{l8}$ (approximately, CAS 8001–35-2) | 414 (average) |  | Not approved (1947, first developed) | 365, other sources (highly variable): 2 months and 14 years; US sources: 9 days; Field DT$_{50}$ 9 to 500 days | 0.67, other sources: 0.13–0.53 (at 20–25 °C) | $6.08 \times 10^{-1}$, 0.42–6382 at 20–25 °C | 3.2–5.5 | In water: 0.3–3/soluble in organic solvents: benzene, xylene, and carbon tetrachloride |

* References: [29–31,33,35–38].

## 2.2. Sampling

Soil is a complex and heterogeneous matrix that contains both inorganic and organic components in variable contents and has a wide range of physicochemical properties and structural characteristics. Hence, soil types are mainly characterized by the proportions of three materials: sand, clay, and organic matter. Soil is the recipient of chemical pollution because of intense agrochemical use. Specifically, when OCPs reach the soil, due to their physical properties, as they are not degraded nor volatilized or even leached but bound to soil organic matter (SOM), they develop strong interactions with soil compared to analogous interactions of pesticides and/with other matrices [39].

No single method applies to all monitoring and assessment needs nor standard soil-sampling guidelines imposed by law or after general agreement. Usually, less than a teaspoonful of soil is used for laboratory analysis. However, that small amount must be representative of the entire area for which the recommendation is to be made. In general, various factors need to be considered in order to develop a successful sampling strategy. The most important factors are sampling purpose, the physicochemical properties of soils, farmer practices, sampling periods, sampling methods, and sampling depth [13]. Among the various sampling methods, random and zig-zag sampling approaches are considered satisfactory for small fields, whether they are even or uneven. Additionally, there are two other basic sampling approaches: grid sampling and zone-based sampling. Grid sampling is probably the most widely used and involves sampling at points on a square grid throughout a field. The basic disadvantage of this sampling technique is that it ignores soil properties and field characteristics. On the other hand, management zones are more data driven. In this direction, the development of a sampling strategy in order to successfully characterize soil contamination with OCPs should be carefully organized. In order to achieve an accurate soil analysis with meaningful interpretation [13,39–41], further factors to be considered are the historical application of OCPs, their possible secondary emissions, as well as their relationships with SOM [42–44].

Glass jars, previously cleaned with organic solvent such as acetone or methanol and dried, are regularly used for collecting and storing soil samples. Frequently, sample collection incorporates the removal of coarse particles and sieving (varying the sieve opening size) to obtain a homogeneous sample. In many studies, soil samples were widely dried by air or oven drying, chemical desiccation, or freeze-drying to obtain a homogeneous and convenient sample. However, the maintenance of the environmental soil samples in their original state has been regarded as the most appropriate approach for preparing samples, especially for OCP analysis. Furthermore, avoiding a drying step minimizes potential contamination from the lab air or possible volatilization losses. Instead, if needed, soil samples could be mixed with a desiccant such as sodium sulfate, Celite, or Hydromatrix to bind water. It is also very important to ensure that the sample is extracted in a room that is free from significant contamination [34].

Most of the soil-sampling procedures related to the extraction of contaminants from soils are performed in forestall, in ornamental, and in agricultural soils from diverse crop fields. In this direction, in 2009, the European Commission launched a soil assessment component to the periodic Land Use/Land Cover Area Frame Survey (LUCAS) to sample and analyze the main properties of topsoil (0–20 cm) in 23 Member States of the European Union (EU). This research, following standard sampling and analytical procedures and while limited to the upper layer of soil cover, managed to create a consistent spatial database of the soil cover across Europe for the first time, with the analysis of all soil samples being carried out in a single laboratory. Sampling was based on a regular grid. More specifically, each sample was made from a mixture of five sub-samples collected from the center and at the four corners of an area of about $100 \times 100$ m$^2$. All soil sub-samples were collected at a depth of 0–20 cm using a stainless-steel shovel. Later, the soil samples were air-dried at room temperature (22–25 °C) for about 15 days, sieved to 2 mm, and stored at 4 °C in pre-cleaned glass jars until analysis [45,46].

One of the main aims of the LUCAS soil initiative was to provide information on the physicochemical status of the soil in EU countries. In two time periods, 2009–2012 and 2015, the LUCAS soil surveys targeted physicochemical properties, including pH, organic carbon, nutrient concentrations, and cation exchange capacity [46]. Following LUCAS sampling procedures and using topsoil samples from this monitoring program, 76 residues of pesticides, including the 9 OCPs banned by the Stockholm Convention, were analyzed in 317 EU agricultural topsoil (0–15/20 cm) samples in the study by Vera Silva [13]. However, the fact that information on farming systems is not available for the LUCAS soil-sampling points brought some limitations to this study. In addition, as pesticide residues often accumulate on the soil surface, further consideration of the uppermost 1 cm of the soil surface layer should take place in the future [47]. In any case, the monitoring of banned pesticides may complicate the process of obtaining representative samples. Taking this into account, it is vital that the sampling protocol for pesticide residues in soil matrices be based initially on a standard procedure but also modified solely based on a focus on pesticides, and the protocol should not be to always obtain samples in the same way [48,49].

The results encountered some inadequacies, such as variation in sampling methods, LOQs, extraction procedures, etc. There is insufficient coverage for factors influencing soil-sampling methods for the accurate monitoring of OCPs residues. However, grid sampling is the sampling method most commonly mentioned. A crucial step in choosing the best strategy must be based on the composition of each soil sample which varies from place to place. In any case, this should not contradict the need to develop a unified approach. At a general level, there is a need to harmonize soil-sampling guidelines in an effort to compare results and apply threshold values throughout Europe. This means following clear guidelines on how to take samples in the same manner for each sample, which should be supported by more research. A general diagram of soil-sampling strategies is presented in Figure 2.

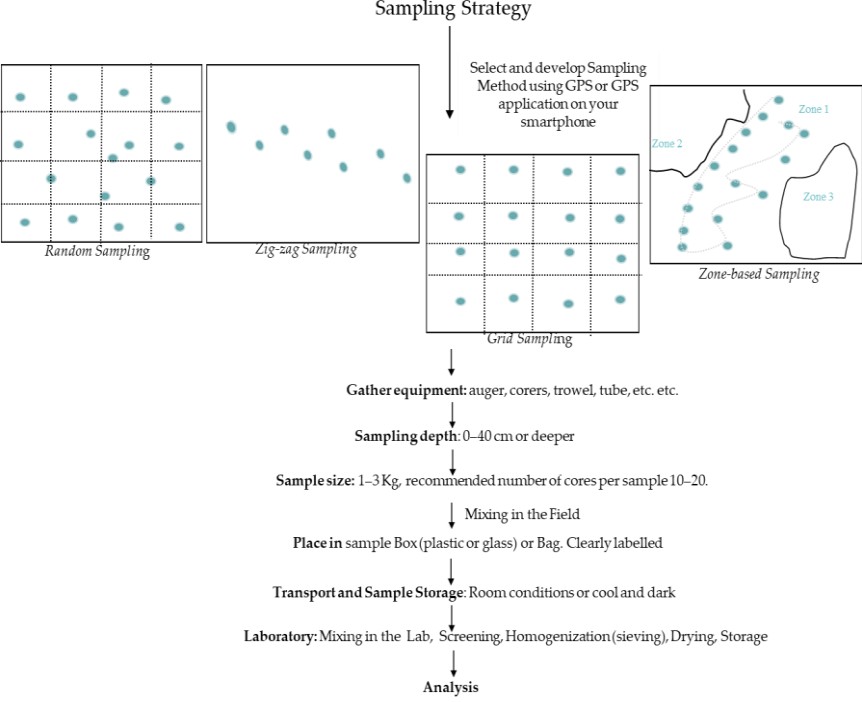

**Figure 2.** General soil-sampling strategy.

## 3. Chemical Analysis

A plethora of methods have been developed and applied for OCP analysis in environmental matrices (i.e., soil and sediment). The entire chemical analysis involves several crucial stages such as sample preparation and analyte chromatographic separation, includ-

ing detection, quantification, and data analysis. Among them, the sample preparation step is considered as the most critical point to be made prior to instrumental analysis. In particular, the development of methodologies for determining pesticides in soil is a challenging task. As a result of the complexity and the physicochemical characteristics of such types of samples, two main factors should be considered: (a) the extremely low concentration of banned OCPs in soil samples. This is the reason why extremely sensitive analytical methods (including, in many cases, enrichment techniques) are required for the detection and quantification of these analytes at such levels, and (b) the strong binding of OCPs to soil. Consequently, special extraction approaches have been developed [6,35,39].

### 3.1. Extraction Techniques Used for Solid Environmental Samples

3.1.1. Solvent Extraction Techniques

Various extraction and clean-up techniques for pesticide residues from soils have been reported in the literature since the 1990s. During the extraction step, many interfering compounds are co-extracted from soil samples together with the analytes.

Conventional Solvent Extraction Techniques

There are many studies that used the solvent extraction of organic analytes prior to solvent evaporation from solid samples such as environmental soil and sediment matrices, which is commonly known as solid–liquid extraction. Conventional sample preparation techniques include solvent extraction techniques such as mechanical agitation by shaking [50,51], Soxhlet extraction [52,53], as well as ultrasonic solvent extraction [54].

The shake flask method refers to the extraction of organic molecules using a mechanical shaker and some solvents. Almost ten years ago, in the study by Mao et al. in 2012, this extraction method was evaluated in the extraction of DDT pesticides from soil matrices, using five different organic solvents: ethanol, 1-propanol, and three fractions of petroleum ether. Various factors such as organic solvent concentration, wash time, temperature, mixing speed, and solution-to-soil ratio were studied, and it was proved that they could significantly affect the extraction procedure. In the same study, it was concluded that the extraction with 100 mL of petroleum ether (60–90 °C), a washing time of 180 min, a mixing speed of 100 r min$^{-1}$, a solution-to-soil ratio of 10:1, and a washing temperature of 50 °C were the most appropriate parameters for the extraction of DDTs from soil [55]. Since then, there have been few other studies that have referred to this technique, but in most cases, the research and assessment was conducted in conjunction with other, newer extraction techniques [56–58].

A second effective conventional extraction procedure for volatiles and semi-volatiles analytes from solid matrices is Soxhlet extraction (SE), which is still recommended by the US Environment Protection Agency (EPA), Food and Drug Administration (FDA), and Association of Official Analytical Chemists (AOAC) Standards Methods [59]. Generally, the minimum time needed for regular SE is normally ~8 h, where, after many cycles and significant solvent consumption, the desired compound is concentrated in the distillation flask. SE is performed with either a single solvent that is usually toluene or dichloromethane (DCM) or a mixture of solvents such as hexane/acetone, pentane/DCM, etc. As sulfur is present in the sediment and soil sample, clean-up usually involves sulfur removal via a reaction with Cu or tetrabutylammonium sulfite and the use of silica gel, Florisil (MgSiO$_3$), etc. [60]. Twenty-five years ago, Lopez-Aviala et al. were among the first to evaluate the SE procedure to determine several OCPs in soil and sediment samples [61]. Specifically, 250 mL of DCM:acetone (1:1) exchanged by 50 mL of hexane were selected as the extraction solvents following a 24 h extraction period prior to GC–ECD analysis with GC/MS confirmation. Silica gel (SG) was also used to clean individual OCPs from certain interferants [39]. Since then, many studies, including comparative approaches to other extraction techniques, have used the SE extraction method. Even recently, although SE is time-consuming, requires a large volume of organic solvents, and is difficult to modify and automate, perhaps due to the simplicity of the device and the fact that no specialized

training needed, it continues to be used in this direction [62,63]. For example, in 2019, Xing et al. successfully extracted several HCHs (including α-HCH, β-HCH, γ-HCH, and δ-HCH) and other DDTs (including o,p′-DDE, p,p′-DDE, o,p′-DDD, p,p′-DDD, o,p′-DDT, and p,p′-DDT) from surface soils using 150 mL of DCM as the extraction solvent in a 24 h SE extraction procedure, prior to GC–ECD analysis [64]. Alternatively, considering the previously mentioned disadvantages of SE regarding solvent waste and time consumption, it was optimized and automated firstly in 1994, providing the commercial product known as Soxtec extraction. The latter was approved by the EPA as a standard method [60,65,66]. For OCPs, the Soxtec method was compared with a newer procedure based on QuEChERS extraction by Rashid. A. et al. in 2010. Specifically, six soil samples were collected to determine 18 OCPs, yielding results with good agreement between the two extraction methods [67].

Ultrasonic extraction (USE) is another typical extraction method that could be used as an alternative to common SE and to shaking flask extraction to extract pesticides from soil samples. As the extraction temperature and pressure are lower than the common values, is recommended for the determination of thermolabile analytes from soil samples. However, USE in general requires significant amounts of solvent and an extra separation step for residue and extraction via centrifugation or filtration. In 2006, Torr et al. applied this technique for the determination of OCPs and their metabolites from real soil samples [56]. The optimization was performed in relation to the solvent type, the amount of solvent, and the sonication time. Finally, it was proposed that the soil matrices should be extracted twice using 25 mL of a mixture of petroleum ether and acetone (1/1 *v/v*) incorporating 20 min of sonication to achieve satisfactory extraction efficiency. Pesticide recoveries from fortified soil samples ranged from 88% to 92%, with relative standard deviations generally below 6%. The time consumption was reduced by approximately 75% and 82% compared to the shake flask and SE methods, respectively. Furthermore, solvent reduction was near to 67% compared to SE. Even though USE is not widely used to determine OCP extraction in soils, the results of this study strongly demonstrated that USE could be efficiently applied to extract OCPs from soils with solvent and time extraction to be scientifically reduced. Furthermore, a miniaturized ultrasonic extraction procedure was successfully developed for the determination of different OCPs in soil by Ozcan S. et al. in 2009 [68]. The parameters influencing the efficiency procedure (i.e., amount of sample, volume of extraction solvent, number of extraction steps, etc.) were optimized by using 23 experimental factorial designs. Ideally, 0.5 g of soil sample was sonicated for 5 min with 5 mL of petroleum ether and acetone mixture (1/1, *v/v*) in an ultrasonic bath. The extraction was repeated three times with satisfactory reproducibility at lower consumption levels of solvents and samples.

Modern Extraction Techniques

Conventional techniques, as already mentioned, use large amounts of solvents and time-consuming extraction procedures to extract low-content organic analytes from complex solid matrices. Furthermore, as the wasted solvents not only increase the analysis cost but may cause significant pollution by releasing solvents into the environment, constant changes in environmental regulations severely limit the amount of solvent usage in laboratories worldwide. For example, in the United States, an order has called for a 50–90% reduction in solvent usage in all federal laboratories [69]. Later, new extraction procedures were developed to reduce the extraction and amount of solvent required and to improve the accuracy and precision of analytes for the common extraction techniques. Over the past twenty years, some green solvent extraction techniques have been developed to mitigate these effects. Among them are ultrasound extraction (USE), microwave-assisted extraction (MAE) [70,71], accelerated solvent extraction (ASE) [72,73], supercritical fluid extraction (SFE) [72], solid phase extraction (SPE) [74], solid phase microextraction (SPME) [75], matrix solid phase dispersion extraction (MSPDE) [76,77], QuEChERS methods [39,56], and procedures such as focused ultrasound liquid extraction (FUSLE) [78,79] and pressurized liquid extraction (PLE) [29,35,41,73], which all require shorter extraction times and low

amounts of solvents while sometimes providing higher recovery yields of the analytes when compared with classical extraction [80].

In recent years, the USE technique has been developed to minimize its problems and become more useful and applicable to analytical chemistry. Modified focused ultrasound liquid extraction, known as FUSLE, is based on the application of high-power focused ultrasonic waves using a micro-tip immersed directly in the extraction mixture. FUSE has been successfully optimized for the simultaneous analysis of multiple pesticides including POPs in several matrices, including soils and sediments. The variables studied during the optimization process were the percentage of maximum power, extraction time, number of cycles, extraction solvent, and sample amount [81]. In the literature review, the first reference to the use of the FUSE technique for the determination of OCPs in soils was made by Flores-Ramírez et al. in 2015. This group managed to successfully validate FUSE using gas chromatography–electron impact–mass spectrometry (GC-EI-MS) for the determination of 13 OCPs. In optimized conditions, FUSE was carried out with a 1 g sample using 10 mL of hexane:DCM (75:25, *v/v*) as an extraction solvent of for 1 min in duplicate and at 60% irradiation power at 20 °C. In addition, a cleaning procedure was followed using columns packed with Florisil. The results showed that FUSE was viable and easy to use for the detection of POPs in soils [78]. Even if only a few studies for the extraction of POPs from soils are available, it can be considered as a suitable technique.

The ASE technique, also known as PLE, or enhanced solvent extraction (ESE), is a modern comprehensive technique that has become very popular and was accepted by the US EPA (Method 3545A and 6860), the US Contract Laboratory Program (SOW OLM04.2), and ASTM (Standard Practice D-7210 and D-7567). It is also standardized for in use methods, based on ASE extraction, in China (Method GB/T 19649–2005) and in Germany (Method L00.00-34). This technique is similar in principle to SE but is performed at high temperatures in the range of 40–200 °C to enhance the speed of elution and high pressures in the range of 1000–2500 psi to keep the solvents in liquid states. Among the main advantages of ASE, as highlighted in several publications over the years, is the extraction of multi-residue pesticides and the fact that ASE can be used for a wide variety of analytes, as polar and non-polar solvents can be used for the extraction process [29,82]. Many years ago, in 1999, the extraction of pesticides from soil using the ASE technique was described by Gan et al. Soil (10 g) was shaken vigorously with 20 mL of a solvent combination of methanol:water (4:1, *v/v*) for 1 h. The supernatant was decanted after the mixture was centrifuged at 10.000 rpm for 15 min. The efficiency of ASE was proven to be better than that of the classical extraction methods such as SE or shake extraction using the same solvent mixture [82]. Later, several researchers tried to apply ASE to extract some OCPs from environmental solid matrices [73,82,83]. However, in 2008, the team of Vega Moreno et al. was the first that managed to determine a wide group of OCPs ($\alpha$-, $\beta$-, $\gamma$-HCH, aldrin, endrin, dieldrin, p,p′-DDE, etc.) in agricultural soil samples, using a one-step ASE with gas chromatography with an electron-capture detector (ECD) [84,85]. In this work, particular variables of ASE were studied, and they were the type of solvent, the number of cycles, temperature, time, flush volume, and pressure. The optimized conditions were: hexane:acetone (1:1), one cycle, 50 °C, 60% flush volume, and 1500 psi. This method detected thirteen OCPs in soils at low concentrations of up to 0.3 ng·kg$^{-1}$ with satisfactory recoveries and RSD values [84]. The study by Wentao Wang et al. (2007) attempted to test the extraction efficiencies of three methods, including ASE extractions in soil samples with different total organic carbon contents. The results showed that the values obtained were comparable with the values reported by other studies and that ASE demonstrated the best extraction efficiency [81]. In the study by Lehnik-Habrink et al. (2010), the simultaneous extraction of several pesticides, including OCPs from organic forest soil, using the ASE technique with acetone/cyclohexane (2/1, *v/v*) followed by SPE with silica gel and GC–MS analysis, provided the highest extraction efficiency compared to other extraction techniques such as to SE, USE, and sonication [58].

In another study, Xingru Zhao et al. developed a novel method to simultaneously detect eight classes of POPs, including 18 OCPs, in sediment that underwent freeze-drying and biota samples. In this study, the sample extraction was performed via ASE using a mixture of dichloromethane:hexane (1:1, $v/v$). In addition, in order to remove elemental sulfur acting as an interfering substance in sediment samples, further purification of the extracts with activated copper or tetrabutylammonium sulfite mixed with the extract or mounted on the top of the silica column were applied. The proposed clean-up procedure included the fractionation of extracts into two fractions with a multi-layer silica gel column, followed by further fractionation using a basic alumina column (fraction 1) and a Florisil column (fraction 2), respectively [86].

Several studies applying ASE for the extraction of OCPs can be found in the recent literature. In 2018, Chaza et al. analyzed several OCPs, including DDTs and HCHs from dried soils via ASE followed by GC–MS analysis at high levels. The levels of DDT and HCH insecticides were as follows: 29.4 and 10.1 ng g$^{-1}$, respectively. In their work, two successive extraction processes were applied. First, the sample was extracted with hexane/acetone (1/1 $v/v$) at 100 °C and pressure at 103.45 bar. Second, the extraction was performed with DCM at the same temperature and pressure at 138 bar. Finally, the two extracts were combined. To remove the sulfur, clean-up with activated copper was applied and purified in a silica column and several combinations of organic solvents [87]. Two years later, in 2020, Aleksandra Ukalska-Jaruga et al. followed the analytical procedure according to the ISO10382 standard and successfully extracted six OCPs (α-HCH, β-HCH, γ-HCH, p,p′-DDT, p,p′-DDE, and p,p′-DDD) from arable soils in Poland using a hexane:acetone mixture (70:30 $v/v$). The pesticides residues were determined using gas chromatography with an electron-capture detector (GC–μECD) and ranged from 0.61 to 1031.64 μg kg$^{-1}$ [88].

MAE, also called microwave-assisted solvent extraction (MASE), uses microwave energy to heat solvents in contact with a sample and has been applied in many studies as an alternative green method for the extraction of organic pesticides from soil samples. Among the advantages of the method is the fact that it does not cause changes to the molecular structure, it is fast, and it allows for the simultaneous extraction of several compounds. Particularly, if a conventual extraction procedure is considered to take 15–30 min and requires small volumes of solvent in the range of 10–30 mL, MAE uses approximately 10 times smaller volumes. This technique was first introduced by Ganzer and Salgo in two publications in 1986 and 1987 using a domestic microwave. Onuska and Terry used microwave energy to extract OCPs from sediment samples with quantitative recoveries and without compound breakdown due to sample exposure to microwaves [61]. In 1994, Avila et al. evaluated this extraction procedure for the extraction of twenty OCPs from six types of soils and sediments, and in a subsequent study, the list of compounds was expanded to nearly a hundred OCPs and OPPs [47,61,89].

In the following years, several researchers investigated and optimized the factors influencing the performance of MAE extractions of OCP pesticides from soil samples, and many authors were interested in comparing known extraction methods for the publication of numerous studies [49,81,90]. In 2003, Concha-Graña proposed the solvent combination of hexane:acetone (1:1) as the MAE solvent for the extraction of OCP pesticides in soil samples, arguing that the time consumed during the evaporation phase was much shorter with this solvent mixture. The extraction time was set at 10 min, and the highest power applied was set at 800 W [91]. A simple and novel analytical method for quantifying twelve persistent organic pollutants (POPs) in marine sediments was developed in 2005 by Basheer et al., using MAE and a single-step liquid-phase microextraction (LPME), hollow fiber membrane (HFM) clean-up and enrichment procedure [92]. One year later, in a study by Herbet et al. using MAE with a small volume of the same extraction solvent, followed by a clean-up step for the redissolved extracts and GC–MS/MS analysis, the successful isolation and quantitation of eleven OCPs was demonstrated in very complex solid matrices such as landfill soil samples [69]. Over the years, there have been several other reports that have

used the MAE extraction technique in conjunction with other purification steps, further described below in a separate section.

As an alternative to the MAE extraction technique, the Microwave-Assisted Micellar Extraction (MAME) technique, using a micellar (surfactant-rich) medium to substitute organic solvents as an extractant, has been applied to achieve the satisfactory extraction of several organic molecules from solid samples, including soil matrices, with low cost and low toxicity. However, as the micelle-rich phase is viscous and cannot be injected directly into some analysis apparatus (e.g., LC-MS/MS), an additional clean-up such as solid phase extraction (SPE) or headspace (HS)-SPME was also needed, resulting in a multi-step analytical procedure [49,93]. In 2006, Moreno et al. effectively optimized the MAME technique and studied the determination of five OCPs from several kinds of real agricultural soil samples, using a surfactant as an extractant with SPME HPLC-UV. Taking into consideration the reduction in the time and extractant required, the cost was also effectively lowered [94]. In the last decade, fewer studies have focused on the optimization and application of MAE extraction for the determination of OCPs from soils. However, there are several publications in which the MAE method was compared to other extraction methods [57,81]. MAE presented the advantages of shorter extraction time, lower solvent consumption, and greater automation with the possibility of the simultaneous determination of multiple samples. The basic advantages of MAE in comparison to other conventional and alternative methods are based on the use of high temperatures during a short period of time and low solvent volume consumes, while sometimes, MAE offers higher recoveries than SE and USE extraction techniques [80,90,95].

The SFE technique appeared in the late 1980s. There was growing interest in SFE in the early 1990s due to its numerous advantages over liquid extraction. The principle of this technique involves the use of an extractant being in its supercritical state, passing it through a contaminated matrix, solubilizing the contaminant, and transporting it to a collection solvent for analysis. Supercritical $CO_2$ is commonly used in SFE since it is a good extraction medium for non-polar compounds and moderately polar ones, such as OCPs. The extraction efficiency of $CO_2$ can be improved by adding small amounts of modifiers that interact with the matrix to promote desorption into the fluid. In addition to $CO_2$, supercritical $N_2O$ could also be used both with and without modifiers. In 1996, Ling et al. evaluated the SFE extraction of 16 OCP pesticides from soil samples and studied the SFE extraction efficiency based on analyte–matrix interactions. The soil properties of TOC and pH were found to significantly influence the extraction efficiency, while moisture content (0.9% to 5.0%) did not affect the recovery results, except for the endrin aldehyde compound. Specifically, the increase in the recovery of this analyte with increasing amounts of moisture content was associated with the formation of H-bonding between the carbonyl group in the endrin aldehyde and the water molecule. In addition, the simultaneous clean-up of sulfur-containing soils using silver nitrate ($AgNO_3$) in the extraction cell proved to improve the extraction efficiency, except that the heptachlor analyte appeared to be degraded into heptachlor epoxide using $AgNO_3$. The total amount of time needed from SFE extraction to GC–ECD analysis was less than 2 h [96].

Many years later, in 2006, an experimental design approach was used to optimize the SFE extraction conditions for several pesticides, including OCPs in real soil samples from the Povoa de Varzim area, north of Portugal. A high extraction temperature was used given the high percentage of non- and semi-polar compounds, while the addition of a modifier was beneficial for the extraction of more polar pesticides. SFE has been shown to be an attractive technique for the analysis of pesticide multi-residues in soil samples, confirming a few of the previous results for persistent OCPs in the same solid environmental matrices [97]. This procedure is simple, rapid, and only requires a small number of samples and solvents. SFE was proven to be more efficient than solvent extraction, particularly for non-polar components in solid matrices such as soils, especially with high organic carbon contents, while the applicability of SFE to polar pollutants and metabolites is limited. Furthermore, SFE produces clean extracts in comparison to other extraction procedures

(e.g., Sohxlet), and no additional purification is required [72]. Despite evident advantages, it was soon found that extraction conditions are strongly dependent on both the solutes and the matrix, so that parameters need to be adjusted for every new application. For example, it was reported that despite their high solubility in $CO_2$, the solute–matrix interactions may yield lower recoveries for OCPs than expected from solubility alone. Thus, extractions of OCPs from spiked soils were unsatisfactory, especially for soils with a high organic content [97]. Therefore, SFE is not widely adopted in official methods, and only a few analytical protocols have been reported for the multi-residue analysis of pesticides in soils based on SFE [98].

### 3.1.2. Other Extraction and Clean-Up Techniques

SPE was introduced in the 1970s, and it is still dominantly used for the extraction of organic pollutants from soil samples. Many different sorbents (which were previously conditioned by an appropriate solvent or solvent mixture) are used for the isolation of the analytes depending on the properties of the analytes and the matrix samples. The most commonly reported sorbents in pesticide extraction are reverse phase octadecyl (C18), normal-phase aminopropyl (-NH2), primary–secondary amine (PSA), anion-exchange three-methyl ammonium (SAX), and adsorbents such as graphitized carbon black (GCB). In addition, normal-phase sorbents such as Florisil, aluminum oxide ($Al_2O_3$), and silica ($SiO_2$) are usually used in combination with the previously mentioned sorbents due to their extremely polar character, and they are the proper choice for the efficient extraction of non-polar analytes in soil such as OCPs [99]. There were several studies reported regarding the extraction of OCP pesticides from environmental matrices, where SPE was used as a clean-up step in combination with several extraction techniques. For example, in 2009, Hu et al. reported the use of SPE cartridges containing 1 g of Florisil after USE for the extraction of several OCP pesticides from surface soil samples and sediments [45,74].

In 2013, a study based on MAE using an additional SPE clean-up step was developed by Yu Liu. Twenty-three OCP pesticides were extracted from soil samples using MAE, followed by the evaporation of the solvent and purification with a silica and alumina combination column, prior to GC–ECD and GC–negative chemical ionization (NCI)-MS analysis [100]. Two years later, Shanshan et al. likewise used MAE–SPE for OCP analysis in six soil samples. The instrument was able to extract 40 solid samples in PTFE extraction vessels in a single step using a petroleum ether:acetone (1:1, *v/v*) solvent mixture combined with a clean-up step on a Florisil-SPE column. Satisfactory recovery, clean chromatograms, proper selectivity, and accuracy were achieved [95].

A different clean-up technique is dispersive solid phase extraction (dSPE), where the media are suspended in the sample solution to trap interferences from the solution to be able to analyze the solution for the target. More information on the use of this purification technique for the successful extraction and analysis of OCPs from soils can be found below. SPME, introduced by Pawliszyn and his coworkers as a novel and solvent-free sample pretreatment technique, is based on the redistribution of analytes between the microextraction fiber and the sample matrix [101]. SPME is an equilibrium technique where analytes are distributed between three phases, sample, gas phase, and fiber, with the advantage of simplicity, lower LOD, and reproducibility. Most of the available studies regarding the SPME technique were focused on water and to a lesser extent on soil, sediments, and air. The applications for the determination of OCPs in soil matrices are based on the preparation of soil with distilled water and the direct dipping of the SPME fiber into the produced slurry.

Researchers found that the recoveries of organic pollutants were obviously affected by the complex sample matrix. As a result, HS-SPME was utilized to determine OCP pesticides, improving selectivity and causing reductions in background adsorption and the matrix effect as the fiber is not in contact with the sample, with the advantage of a longer life for the SPME coating [69,102]. In 2006, Zhao et al. reported the SPME method for the determination of OCPs in environmental soil samples with validation results comparable

to the USE extraction procedure. A very interesting observation of this group was the fact that aging the soil samples prior to spiking them with a standard solution during method validation produced samples closer to the real soil samples compared to the not-aged soil samples [75]. In the work by Carvalho et al. (2008), MAE combined with HS-SPME allotted further clean-up and pre-concentration steps, resulting in a quick and efficient procedure of sample preparation before the GC–MS determination of several OCPs in different sediment samples [90]. This combination of MAE and HS-SPME proved to be more effective than each one of these procedures acting separately as it permits the pre-concentration of analytes at the fiber and minimizes the need of a pre-clean-up of the extract from MAE, since the microextraction is performed in the headspace. In the study by Concha-Graña et al. in 2010, pressurized hot water extraction (PHWE) followed by the SPME and GC–MS methods was used for the analysis of 28 OCPs in sediment samples [103]. Alternatively, in 2014, M. Miclean et al. reported the PHWE technique in conjunction with SPE and SPME, followed by GC–ECD analysis for the determination of 19 OCPs in real soil samples collected in Cluj County, Romania [104]. Recently, in 2018, He et al. evaluated another combined approach for the determination of eight OCPs in soil, based on microwave-assisted magnetic solid phase extraction (MAE-MSPE) prior to GC–ECD analysis. The advantage of this technique was the use of novel $Fe_3O_4$-$NH_2$ MIL-101(Cr) composites as the MSPE, which aided in the selective enrichment and purification towards the targets [105]. In any case, a drawback of the technique is that minimum detection limits were achieved, as the pesticide compounds were concentrated on the SPE cartridge or (and) on the SPME fiber, and they were rapidly delivered to the column.

A miniaturized solid–liquid extraction (MISOLEX) combined with SPME as an automized clean-up procedure and GC–MS analysis was successfully validated for the quick and simple extraction of HCH and DDT from soils. In principle, the RT of an analyte was used as an indicator of its physicochemical properties [106].

Matrix solid phase dispersion (MSPD) was developed by Barker in 1989 for the extraction of solid and semi-solid samples as a new SPE-based extraction and clean-up technique that involves mixing a solid or semi-solid sample with a sorbent material (silica, alumina, Florisil, C18, etc.) to form a dry body. Then, after blending it, the body is packed into a tube and eluted with solvent to allow target analyte to be separated from the body. The most commonly used adsorbents were C8- and C18-bonded silica materials. MSPD has been reported in the past to be used for the multi-residue analysis of several pesticide classes, including OCPs [47]. Recently, this extraction and clean-up technique was reported in 2018 in the study by Wang et al. for the determination of two typical groups of OCPs, DDTs, and HCHs in farmland and bare land soil samples [76,77]. Compared with classical methods, the MSPD procedure is simple and less labor-intensive, and it allows extraction and purification in a single step using less-toxic solvents. An important practical advantage is that MSPD does not need special instruments or costly hardware. However, as with the other solid–liquid extraction methods, the LODs are worse than those obtained, for example, using LSE or even SPME [107]. Another drawback of this method is that, in many cases, large volumes of solvents are required, and an additional evaporation step is required.

The solid–liquid extraction with low temperature purification (SLE-LTP) could be considered as a very good alternative for the extraction of organic contaminants in solid and semi-solid matrices to emphasize the clean-up of the extracts. SLE-LTP, like all the other SLE techniques, is easier, is cheaper as it entails low solvent consumption, and has been presented in the literature to have high extraction efficiency. Recently, in 2017, Mesquita et al. studied SLE-LTP as well as liquid–liquid extraction with low-temperature purification (LLE-LTP) extraction techniques combined with the GC–MS analytical method for the determination of ten OCPs in three types of samples, including a soil matrix [108]. The two extraction methods were optimized three years earlier based on a bibliography on sewage sludge using SLE-LTP [109]. Regarding the soil matrix, due to its complexity, a key parameter studied was the homogenization time of the vial containing it, which increased

from 1 min to 5 min, providing excellent recoveries around 93 to 114%. A second parameter to be considered was the volume of water for freezing the soil. While the moisture of the soil was only 2% ($w/w$) and as acetonitrile and water is a homogeneous system, it did not homogenize the vial in vortex before freezing, gaining recoveries that ranged from 93 to 125%.

Despite the new trends that have been appeared in pesticide residue analysis, conventional SE methods are still used for routine analysis. To overcome their identified disadvantages and retain their advantages, a different method with "Green Chemistry" characteristics has been developed, known as QuEChERS (Quick, Easy, Cheap, Effective, Rugged and Safe).

QuEChERS Extraction Technique

QuEChERS present a unique combination process involving initial extraction with acetonitrile followed by an extraction/partitioning step after the addition of a salt mixture. The raw extract is cleaned-up with dSPE, with $MgSO_4$ and PSA as the sorbent. Regarding the d-SPE step (not named this at that time), sorbents were also already applied for sample clean-up, though mostly in SPE cartridges [41]. Anastassiades et al. introduced QuEChERS as a simplified version of conventional extraction methods for the determination of multi-class pesticide residues in fruits and vegetables using GC–MS and TPP as IS [110]. The original unbuffered version evolved into two official methods, the European standard EN 15662, developed by Anastassiades and co-workers, which involves the use of citrate buffer, [111] and the American standard accepted by the Association of Official Analytical Chemists (AOAC) and developed by Lehotay, which involves the use of acetate buffer [112]. Both versions lead to a pH of around 5, which corresponds to a compromise to extract the analytes of interest satisfactorily.

Subsequent adjustments were developed to make the method performance even better, especially for some difficult analytes and commodities of different origins. Specifically, a wide variety of factors were studied and evaluated regarding the sample constitution, type of extraction process, extraction time, extraction solvent, sample/solvent ratio, extraction temperature, addition of non-polar co-solvents, and/or salts and clean-up. Researchers adapting appropriate modifications to the QuEChERS method achieved the extraction of a wide range of pesticides, including highly polar pesticides as well as highly acidic and basic ones [113]. This extraction process is mainly applied for pesticides in plants and foods of animal origin and less extensively applied in several complex matrices, such as biological fluids [114], non-edible plants, and environmental samples [40,115], including soils [41,113,116]. The QuEChERS method has been applied to the extraction of pesticides from soils without clean-up [41] or with clean-up, mostly using PSA and $MgSO_4$, PSA and C18, and LLE, achieving recoveries from 53 to 128% [41]. In a recent manuscript by Leesun Kim. et al. in 2019, an overview of the current information on the QuEChERS method applied to the POP analysis of various sample matrices indicated that especially OCP analyses have mainly been limited to sample matrices with lipid content such as fish and seafood [40,114,117–119]. There are only a few studies having successfully used the QuEChERS method to determine OCPs in environmental matrices (e.g., water and sediment), marine products, hair, and medical plants using the modified QuEChERS methods [40,117,118]. The QuEChERS methodology was first applied to the extraction of pesticides from soils in 2008 by Lesueur et al. [120]. The application of the QuEChERS method provided good results for the extraction of polar as well as non-polar pesticides, strengthening its diverse applicability.

Based on a Scopus survey, since the development of QuEChERS in 2003, up to November 2020, among 1826 documents relating to pesticide extraction using this method, 151 papers specifically referred to OCPs. Furthermore, 368 documents referred to QuEChERS extraction from soil matrices, while only 22 documents have been published regarding OCP extraction from soils. Figure 3 presents a simple diagram of the QuEChERS extraction trend for OCPs in previous years.

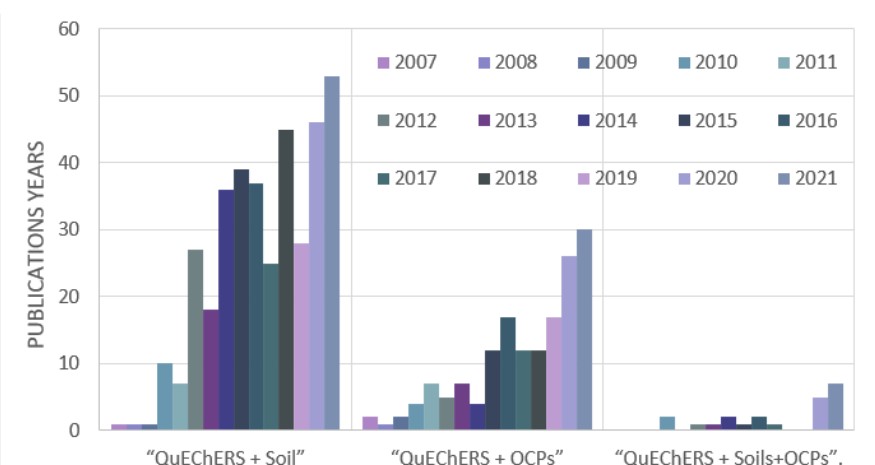

**Figure 3.** Trend chart for the application of QuEChERS extraction method. Article title, abstract, and keyword literature search was conducted with the following search terms: "QuEChERS + Soil"; "QuEChERS + OCPs"; "QuEChERS + Soils + OCPs". Source: Scopus (December 2021).

The QuEChERS methodology was first applied to the extraction of pesticides from soils in 2008 by Lesueur et al. [120]. In this study, the authors compared different extraction methods for 24 multiclass pesticides including LND that were commonly reported as soil pollutants in the literature. In addition to Lesueur et al., other authors have applied the QuEChERS methodology for the extraction of the mentioned pesticide classes, which presented high recoveries [111]. Two years later, a procedure based on QuEChERS was developed for the determinations of nineteen OCPs in soil samples with different physico-chemical properties. The proposed method was compared with an established procedure based on Soxhlet extraction, indicating recovery results of good agreement. QuEChERS, due to the cleaner nature of the extraction, presented lower detection limits [67].

The QuEChERS method has been applied to the extraction of pesticides from soils without clean-up [113,121] or with clean-up using sorbents, with the most widely used being PSA, C18, florisil, and their combinations, jointly with anhydrous $MgSO_4$ [41,120–122]. In this sense, Cvetkovic et al., to improve the overall technique performance, proposed a modification of the original method consisting of the same LLE step but using diatomaceous earth and clinoptilolite as d-SPE sorbents, which have never been used before, for the extraction of the 16 polycyclic aromatic hydrocarbons (PAHs) listed by the US EPA from soil samples. The use of the diatomaceous earth, in addition to reducing the analysis cost, in combination with acetonitrile:water (2:1) as the extraction solvent, provided the best recovery and precision results [116,123].

Nineteen OCPs in hydrated soil samples using acetonitrile and liquid–liquid partition into n-hexane achieved satisfactory recoveries [67]. In the same year, Yang et al. used this technique to determine 38 pesticides, including endosulfan I and II, in freeze-dried soil and sediment samples of a vegetable-growing area in China and achieved good reproducibility and low detection limits [124]. In 2012, Florent Rouvière et al. optimized and modified the conventional QuEChERS method for the extraction of 34 OCPs from organic peat soils. For the first time, in the extraction of the major pesticides tested, the use of a non-miscible-water solvent such as DCM was proven to be a better solvent than the commonly used acetonitrile. Moreover, clean extracts, without the need for further purification by dSPE, have been produced using PSA, which is an expensive but efficacious material. However, DCM is one of the solvents that has been avoided in recent years. At the European level, regarding the authorization of plant protection products, hazardous reagents (e.g., carcinogens, mutagens, and reproductive toxicants of category 1 and 2 according to Regulation (EC) No 1907/2006 [125]) are not permitted in risk assessment and monitoring studies [126]. At the same time, in this paper, the ASE extraction technique was also optimized and compared to QuEChERS. The latter was proved to be rapid, easy to use, and provided

adequate recoveries [72]. In the same year, the QuEChERS method was developed by Luisa Correia-Safor et al. for the determination of 14 organochlorine pesticides in 14 soil samples from different Portuguese regions with different organic carbon contents, most of which came from carrot cultivation fields. The best recoveries were obtained when OCPs were extracted from hydrated soils (5 g/sample) with 3 mL of water and 7 mL of ACN. Afterwards, an AOAC packet and a clean-up step with PSA in combination with C18 was used [121]. In another study, the organic matter content was proven to be a key factor in the process efficiency. Generally, pesticides such as OCPs are mostly absorbed in soils rich in organic matter or clay. In particular, Lopez-Avila et al. reported lower recoveries for soils with higher contents of organic matter. As a result, it was concluded that soil matrices with higher organic carbon content have a negative effect on pesticides' recoveries, probably due to a possible sequestration [81,121,127].

Later, in 2016, Bed Salem et al., based on a previous QuEChERS method followed by GC–MS, which was validated in 2010 by Yang et al. regarding the analysis of 38 pesticides in soils [A.52 116], studied the simultaneous analysis of 16 PAHs, 12 PCBs, and 9 OCPs in sediment samples. The sample preparation was evaluated regarding the selection of the extraction solvent, the extraction technique, and the amount of sediment. The mixture of dichloromethane:acetone (DA) (1:1) using ultrasonic agitation was finally chosen for the simultaneous extraction of PAHs, PCBs, and OCPs in 5 g of frozen sediment samples. As in the study by Florent Rouvière et al. [72], the DA solvent mixture was only considered as the best extraction solvent for the simultaneous determination of all pesticides [128]. In the same year, Cheng et al. investigated the QuEChERS method combined with atmospheric pressure gas chromatography quadrupole–time-of-flight mass spectrometry (APGC-QTOF-MS) as an alternative sensitive soft ionization technique for the simultaneous determination of 15 OCPs in soil and water samples. The modified QuEChERS method used a combination of hexane with acetone (9:1 $v/v$) as the extraction solvent, NaCl as a partition salt, and florisil and MgSO4 as sorbents in the clean-up step [129]. Another QuEChERS method followed by d-SPE clean-up was developed by Yu et al. for the simultaneous determination of 58 pesticides from several classes, including OCPs in soil [130]. The use of the QuEChERS method combined with GC–MS/MS for the analysis of dozens of pesticides in soils was rarely reported. In this sense, a large group of 305 organic compounds, of which 49 were POPs, including 12 OCPs, were extracted and analyzed in the study by Andrea Acosta-Dacal with an easy QuEChERS extraction method with LC-MS/MS and GC–MS/MS. Then, to verify the applicability of the validated method, 81 agricultural soil samples of the Canary Islands, classified as clay loam, were used. A second study aimed to extend the use of the QuEChERS-based method for the extraction and analysis of the already-mentioned 305 organic compounds of different natures and origins to other types of soil of the agricultural land of the Canary archipelago: sandy loam, sandy clay, clay, and loamy sand [122,124,131]. This proposed method provides an effective method for pesticide screening in soil.

A case study recently published by Karasali et al. belongs to a series of studies reported to adequately detect and quantify pesticide residues in soil. In this study, using a developed GC–MS/MS analytical method, one soil sample tested positive for dieldrin at 0.018 mg kg$^{-1}$ [132], adding to the evidence of the presence of this persistent chemical in European soil.

### 3.1.3. An Overview of the Extraction Procedures

The extraction and clean-up technique used to determine OCPs in soil matrices has been the most defining step and the weakest link in the whole analytical procedure until recently. Classical extraction techniques used to determine OCP residues in soil, although showing acceptable recovery and accuracy results, have many drawbacks. They are laborious, time-consuming (involving many steps), require large amounts of organic solvents, leading to loss of some analyte quantity, are difficult to modify and automate, etc. On account of this, upgrading the existing technique and its replacement with extraction tech-

niques of lower environmental impact, such as SFE, SPE, SPME, MAE, ASE, QuEChERS, MSPD, FULSE, etc., is increasingly used in pesticide analysis of soil matrices and may be particularly suited to developing countries.

Among them, MAE offers higher recoveries than SE and USE. Furthermore, it allows the simultaneous extraction of several compounds. However, to absorb microwaves, MAE requires polar solvents which tend to co-extract matrix interferences. SFE was proven to be more efficient than solvent extraction, particularly for apolar components in soils with high organic carbon contents, such as peat. However, due to solute–matrix interactions, only a few studies have been reported that determine OCPs in soils. ASE offers numerous advantages: it is quite a quick method which uses low amounts of solvents and allows the use of solvents with a wide range of polarities, and for this reason, it is used for the extraction of multi-residue pesticides. Moreover, the high pressure and temperature conditions enable a better penetration of the solvent into the matrix and a breaking of the intermolecular bonds [17]. As the recoveries of OCPs were affected by the complex matrix of soils, new extraction procedures have been developed or modified, with many of them followed by clean-up procedures such as SPE, SPME, etc. [95,112,121]. Among them, SLE techniques were reported in many studies with the worst LODs. On the other hand, it is remarkable that the conventional SE extraction method, also characterized by the simplicity of the device and the fact that no specialized training is needed, continues to be used in many laboratories in many studies.

QuEChERS consists of an easy alternative extraction method, able to deliver similar or better results and overcome the identified disadvantages in relation to the former methods used for the determination of OCPs in soils. QuEChERS allows for the use of different forms of LLE, SLE, SPE, and others, depending essentially on the nature of the analytes to extract and the complexity of the matrix. In the analysis of soil contaminants, Di et al. analyzed several OCPs in soils and reported that MAE and QuEChERS extraction methods generally yielded higher results compared to the USE and ASE [95].

### 3.2. Analytical Methods

Albeit multi-residue analyses have more commonly adopted liquid chromatography tandem mass spectrometry (LC–MS/MS) than gas chromatography tandem mass spectrometry (GC–MS/MS), in recent years, GC–MS/MS has increasingly been used in pesticide analysis. Various multi-residue analytical methods using GC–MS have been reported for the analysis of pesticide residues in different environmental media. In particular, OCPs as volatile or semi volatile compounds tend to be determined by GC equipped with either electron-capture (ECD) or mass spectrometry (MS) detection, unlike the analysis of other POPs, which tend to employ GC–MS/MS or LC–MS/MS [40,41,56]. In any case, the extraction of multiple pesticides including OCPs from complex environmental matrices, such as soils, remains a challenge due to their different physiochemical properties [124].

The regular use of GC–ECD for the detection of OCPs is based on its high resolution and sensitivity. Additional advantages of this detector include reduced cost of operation and the fact that it requires less technical skill to obtain reliable results. However, due to the already-mentioned complex matrices and soil interferences, the use of GC–ECD often leads to false positive results. In this sense, more selective methods such as GC–MS or even GC–MS/MS have been developed for the determination of OCPs in complex matrices in order to confirm and further determine pesticides' identity [53,133]. For example, in the study by Yu Liu, two analytical methods, GC–ECD and GC–MS with negative chemical ionization (NCI), were evaluated and compared for the measurement of 26 organochlorine pesticides. Although GC–ECD sensitivities for most of the compounds investigated were higher than those observed with the GC–NCI-MS method, the matrix interferences were obvious with GC–ECD. Consequently, GC–ECD should be used with caution in real environmental sample analysis, as the possibility of false positives associated with the GC–ECD methods may result in an overestimation of the concentrations of these banned compounds in the environment [100].

Atmospheric pressure gas chromatography (APGC) is an alternative soft ionization technique used to overcome sensitivity limitations of GC–MS methods. In particular, the APGC technique was coupled to a quadrupole-time-of-flight (QTOF) instrument by Cheng et al. for 15 OCPs in soil and water samples to identify their ionization behavior under atmospheric pressure conditions, as well as to compare the difference between EI and APGC ionization sources. Although the method was successfully applied to the analysis of those real environmental samples, there is limited research available on the application of APGC-QTOF-MS for the determination of OCPs [129].

Although methods based on liquid chromatography (LC) seem most appropriate when pesticides are thermally instable, over the years, studies have rarely examined the determination of OCP pesticides in environmental and other matrices using LC. However, several OCPs such as chlordecone, due to their physicochemical properties, do not separate well from other OCPs using conventional GC [134]. For this reason, LC-(ESI-)-MS/MS was used as an alternative to quantify chlordecone in animal tissues, drinking water, and human biological fluids [134]. Recently, in the report by Naidu. et al., the use of HPLC-UV in detecting LND and its derivatives in biological samples assisted by confirmation with matrix-assisted laser desorption/ionization (MALDI)-TOF was described [135]. Though HPLC combined with a traditional UV detector is less selective and sensitive compared to GC instruments, residues of DCF and dichlorobenzophenone (DBP) were quantified using HPLC-UV in wastewaters by Oliveira, J. L et al. [136]. In particular, the determination of OCP residues in soil matrices using the LC technique is more rarely noticed. Individually, a report is available from 2006 by Moreno et al., where MAME was optimized in conjunction to SPME-HPLC-UV for the determination of OCPs in agricultural soil samples [94].

Referring to alternative methods to avoid or reduce the required clean-up steps, expensive instrumentation, and the use of large amounts of solvents, rapid bioanalytic methods such as enzyme-linked immunosorbent assays (ELISA) have been tested individually in order to quantify few OCPs [34]. This method was used in 2002 in the study by Shivaramaiah et al. to monitor DDT/DDE residues in soil samples. The ELISA result obtained was compared against results from the GC instrumental method. Recovery values calculated using the ELISA method were comparable to that of GC, and the results indicate that the developed assay could be an alternative analytical tool for monitoring DDE [137,138].

Table 2 summarizes analytical methods in combination with different extraction techniques used mainly in selective studies, and a few individual techniques were also found to determine the most studied OCPs in soils. As shown in the table below, among the various methodologies that have been tested over the years, no significant differences were identified in terms of accuracy, precision, LODs/LOQs, etc.

**Table 2.** Analytical methods used to analyze the most studied OCPs in solid environmental samples.

| Analytes | Sample Treatment | Technique | LODs, LOQs | Recovery % (RSD %) | Reference/Year |
|---|---|---|---|---|---|
| α-, β-, γ-HCH, aldrin, endrin, dieldrin, p,p′-DDE, p,p′-DDD, p,p′-DDT, α-, β-endosulfan, heptachlorobenzene (HPT) | ASE Extraction solvent: hexane-acetone (1:1–45 mL); conditions: 50 °C; 1500 psi; preheating period, 5 min; static extraction, 5 min | GC–ECD HP-5MS (cross-linked 5%-phenyl-methylpolysiloxane) of 30 m × 0.25 mm with a 0.25 μm, carrier gas; helium, was 2.8 mL min$^{-1}$ | 0.0003–0.2 ng/g (LOD) highest for p,p′-DDT and lowest for p,p′-DDD and β-endosulfan | 64–103 (1.8–11) | [84]/2008 |
| MRX, α- and γ- Chlordane, p,p′-DDT, heptachlor, heptachlor epoxies isomer A, γ-HCH, dieldrin, endrin, aldrin, and HCB | MAE 20 mL (n-hexane–acetone 1:1)-HS SPME HS-SPME conditions: headspace sampling of 40 mL of sample (1.8% of ethanol) at 65 °C for 60 min, with 100 μm PDMS-coated fiber | GC–MS/MS (EI at 70 eV) with multiple reaction monitoring (MRM)), Varian 60 m × 0.25 mm CP-Sil 8 CB lowbleed/MS (0.24 μm film thickness). Helium (0.9 mL/min, constant flow) as carrier gas | 0.02 to 3.6 ng/g (LOD) | 35–51 with four exceptions (aldrin, the chlordanes, and MRX) (2–13) for soil with high content of organics (8.4%) 72–115 (16–31) for soil with low content of organics (2.2%) | [69]/2006 |
| MRX, DDT, chlordane, HPT, LND, HCB, heptachlor, aldrin, trans- and cis-chlordane, endrin | SLE-LTP 4.00 g of soil with 4 mL of water and 8 mL of acetonitrile, vortex for 5 min, freezer at −20 °C for 1 h for phase separation. Organic phase to 375 mg of anhydrous sodium sulphate, vortex for 30 s and centrifuged at 4000 rpm for 10 min | GC–MS (EI at 70 Ev) and selective ion monitoring mode (SIM), DB-5 MS capillary column, with 5% phenyl stationary phase and 95% methylpolysiloxane (30 m × 0.32 mm i.d. × 0.25 μm, helium as carrier gas at a flow rate of 1.0 mL min$^{-1}$ | 1 and 6 ng/g (LOQ) | 78–115 (1–13) | [108]/2017 |
| p,p′-DDD, p,p′DDT, o,p′-DDT, p,p′-DDE, dieldrin | MAME-SPME 2 g sample, no solvent MAME: 8 mL of surfactant: Polyoxyethlylene 10 Lauryl Ether with concentration is 5% (*v/v*) for 2 g of soils; 1000 W for microwave power during 14 min SPME: PDMS/DVB fiber; 40 min for absorption time, 4 min for desorption time, room temperature, and non-addition of salt | HPLC-UV Varian Microsorb-MV 100 C18 column, 250 mm × 4.6 mm, 8 μm, methanol–water (85:15%, *v/v*) isocratic with a flow-rate of 1 mL min$^{-1}$ 238 nm (Diledrin 220 nm) | 56–96 ng/g (LOD) | 76–98 (5.5–8.4) soil (in the range of concentrations studied (120–2000 ng g$^{-1}$) with organic matter (3.9–6.2%), except dieldrin for garden uses soil.) | [94]/2006 |
| α-, β-, γ-, δ- HCH, heptachlor, aldrin, dieldrin, o,p′-DDE, p,p′-DDE, p,p′-DDT, methoxychlor, MRX, heptachlor | USE Extraction: 10 g soil, twice petroleum ether and acetone (1/1 *v/v*) for 20 min, volume: 25 mL clean-up column activated Alumina, 1 mL extract with 100 mL of n-hexane:ethylacetate (7/3 *v/v*) | GC–ECD DB- 1701 (30 m length, 0.25 mm i.d. and 0.25 μm) and HP-5 (for confirmation) column, nitrogen was used as carrier (flow rate 1.23 mL/min) | 4.8–10.3 ng/g (LOQ) | >88 (<6) for 15–200 ng g$^{-1}$, 92–101 (1–3) using clean-up chromatography (note: soil was 56.1% sand, 22.9% silt, 21.0% clay, organic matter: 1.68%, pH (0.01 M CaCl$_2$): 7.1 and maximum water capacity: 21.2%.) | [56]/2006 |

Table 2. *Cont.*

| Analytes | Sample Treatment | Technique | LODs, LOQs | Recovery % (RSD %) | Reference/Year |
|---|---|---|---|---|---|
| Simultaneous extraction of PAHs and OCPs including DDTs and HCHs (α-, β-, γ-, δ-HCH, o,p′-DDE, o,p′-DDD, o,p′-DDT, p,p′-DDE, p,p′-DDD, and p,p′-DDT) | **Soxhlet** Extraction: 5 g soil, 100 mL hexane/acetone (1:1, *v/v*) for 15 h. | Clean-up: dryness under reduced pressure in a 35 °C water bath using a rotary evaporator, n-hexane, chromatography column (30 cm × 10 mm i.d.) filled with 10 g silica gel (100–200 mesh) to separate the PAHs, DDTs | GC–ECD HP-5 column (0.25 mm i.d., 0.25 μm film thickness), N2 as carrier gas at a rate of 23 mL min$^{-1}$. | 0.47–1.67 ng/g (LOQ) | 86.8–105.1 (0.61–13.12) at levels of 20 ng g$^{-1}$ of each compound in soil samples | [81]/2007 |
| | **MAE** Extraction: 5 g soil, 25 mL n-hexane and acetone (1:1, *v/v*); 1200 W, 110 °C for 20 min, | | | 053–1.40 ng/g (LOQ) | 85.0–104.0 (0.5–9.3) at levels of 20 ng g$^{-1}$ of each compound in soil samples | |
| | **ASE** Extraction: 10 g soil, DCM/ acetone (1:1) at 140 °C and 1500 psi, 11 min | | | 0.53–1.87 ng/g (LOQ) | 82.9–105.4 (0.5–1.9) at levels of 20 ng g$^{-1}$ of each compound in soil samples | |
| 19 OCPs: α-HCH, γ-HCH, heptachlor, trans- and cis-chlordane, p,p′-DDT, o,p′-DDT, p,p′-DDD, p,p′-DDE, aldrin, dieldrin, α-endosulfan and β- endosulfan, endosulfan sulfate, heptachlor, trans-heptachlor epoxide, etc. | QuEChERS with liquid–liquid partition and hydration: 5 g air-dried, 5-day-aged soil with 10 mL water or 1.0 M aqueous Na$_2$-EDTA solution for 30 min. Extraction: An aliquot (10 mL) of acetonitrile + acetic acid mixture (99:1, *v/v*) was added to the centrifuge tube containing the hydrated sample. After 30 s vortex, 4 g anhydrous MgSO$_4$ (4.0 g) and NaAc·3H$_2$O (1.7 g) were added. The contents were shaken vigorously and then centrifuged at 5000× *g* for 5 min. Simple clean-up/concentration step: aliquot (8 mL) of the upper acetonitrile layer in hexane | GC–MS/MS triple quadrupole MS/MS in EI mode and MRM, fused silica capillary column (Zebron ZB-50, 50% phenyl 50% methylpolysiloxane, 30 m × 0.25 mm i.d., and 0.25 μm film thickness; helium as carrier gas at 1 mL min$^{-1}$ | 1 ng/g (LOQ) | 70–100 (<20) at 1–200 g kg$^{-1}$ (exception of HCB) Notes: no significant difference in extraction efficiency for the two hydration methods—no need to use EDTA; 5 soils with organic matter: 1.0–9.8%, etc.—not adversely affected regardless of the type of 5 samples | [67]/2010 |
| Multi-residue method: 38 pesticides include 13 organophosphorus pesticides, 4 herbicides, 8 fungicides, 2 carbamates, 4 pyrethroids, and 7 other class pesticides including endosulfan, DCF | QuEChERS 1.10 g of freeze-dried soil/sediment samples, acetonitrile and buffer salt consisting of 8 g of magnesium sulfate anhydrous grit, 2 g of sodium chloride, 1 g di-sodium hydrogen citrate sesquihydrate, and 2 g of trisodium citrate dehydrate 2.10 mL acetonitrile phase, 1.5 g of MgSO$_4$, and 250 mg PSA and internal stds: pentachloronitrobenzene and triphenylphosphate in methanol | GC–MS with electron impact ionization mode, 5MSi column (30 m × 0.25 mm, 0.25 μm film thickness); Helium carrier | 6–20 ng/g (LOQ) 6.4 ng/g (LOQ for dicofol), 22.4 ng/g (LOQ for endosulfan I); 10.3 ng/g (LOQ for endosulfan II) | 94–110 (3–11) | [124]/2010 |

**Table 2.** *Cont.*

| Analytes | Sample Treatment | Technique | LODs, LOQs | Recovery % (RSD %) | Reference/Year |
|---|---|---|---|---|---|
| 14 OCPs: α-,β-,γ-,δ–HCH, HCB, o,p′-DDT, p,p′-DDE, p, p′-DDD, aldrindieldrin, endrin, α-,β- endosulfan, etc. Note: two types of soils 1. HS (organic carbon 2.3%) 2. LS (organic carbon <2.3% | QuEChERS 5 g of dried sieved soil Two types of extraction: (i) adding 3 mL of water and 7 mL of ACN (ii) 20 mL of ACN. 6 g magnesium sulphate (MgSO$_4$), 1.5 g sodium chloride (NaCl), 0.750 g disodium citrate sesquihydrate (Na$_2$HCit 1.5H$_2$O), and 1.5 g sodium citrate dihydrate (Na$_3$Cit 2H$_2$O) Clean-up: 1.5 mL of the supernatant, 50 mg of PSA, 150 mg of MgSO$_4$ and 50 mg of C18 | GC–ECD ZB-XLB column (0.25 µm i.d., 0.25 µm film thickness, Zebron-Phenomenex), helium as carrier gas at flow rate of 1.3 mL/min GC–MS (SIM) and MS/MS (confirmation); SLB-5MS column (30 m 0.25 mm, 0.25 µm film thickness) | For HS 0.0114–0.079 ng g$^{-1}$, for LS 0.0204–0.0493 ng g$^{-1}$. | 77–130 (≤16) with the addition of H$_2$O 20–46 without H$_2$O addition Only dieldrin could be quantified attending the concentration level of 45.36 g kg$^{-1}$ 0.01–0.14 ng g$^{-1}$ for HS soils and 0.005–0.100 ng g$^{-1}$ for LS soils. | [112]/2012 |
| 15 OCPs: α-CHLOR, β-CHLOR, o,p′-DDE, p,p′ -DDE, o,p′-DDT, p,p′ -DDT, o,p′-DDD, p,p′-DDD, α-, β-chlordane, α-endosulfan, LND, aldrin, endrin, MRX, etc. | QuEChERS 5 g soil with hydration step (water) Solvent: hexane:acetone (9:1 *v/v*) 1 g /NaCl; d-SPE: 1.5 mL upper layer solvent with 40 mg florisil and 150 mg MgSO$_4$ | APGC-QTOF-MS HP-5MS with positive polarity (API), analytical column of 30 m × 0.250 mm inner diameter and 0.25 µm film thickness, helium as carrier gas at 2.0 mL/min Compared with EI: more sensitive | <9.99 µg/L (LOQ) | 70.3–118.9 (0.4–18.3 intra-day and 1.0–15.6 inter-day) 10–500 µg/L | [129]/2016 |
| 26 OCPs including aldrin, α-, β-, γ-, δ- HCHs, α-, γ-chlordane, o, p′-DDD, p, p′-DDD, o, p -DDE, p, p′ -DDE, o,p′-DDT, p, p′-DDT, dieldrin, endosulfan I, endosulfan II, endrin, heptachlor, HCB, isodrin, methoxychlor, MRX, oxychlordane, etc., and 2 hexabromobiphenyls (HBBs) | MAE Extraction: hexane: acetone (1:1, *v/v*)) Clean-up: SPE (10 mm i.d. 350 mm length) with 10 g alumina, 10 g silica gel, and 1 cm anhydrous sodium sulfate. | GC–GC–MS with negative chemical ionization (NCI) and SIM, HP-5MS column HP-5MS column (30 m 0.25 mm i.d., 0.25 µm film thickness), helium as the carrier gas with a constant flow rate of 1.0 mL min$^{-1}$ | 0.2–5 ng/mL (LOD) | 93.0–128.3 (3.0–10.8) 0.1–450 ng/mL Notes: Higher sensitivity with GC–ECD but false positives and overestimation were found by the GC–ECD | [100]/2013 |
| | | GC–ECD (30 m 0.32 mm i.d., 0.25 µm film thickness) | 0.05–0.2 ng/mL (LOD) | | |
| 10 OCPs: α-, β- and γ-HCH o,p′-DDT, o,p′-DDD, o,p′-DDE, cis-chlordane, trans- chlordane, α-endosulfan, and β-endosulfan. Several soils with organic matter 3.8–42.0 g/100 g$_{dwt}$ | MAE-SPE Soil samples (5 g), solvent: petroleum ether–acetone (1:1, *v/v*). 100 °C over 5 min and held for 10 min, cooled down for 15 min. The microwave power was 800 W—extracts were filtered through 5 g of anhydrous sodium sulfate for dehydration—purification on Florisil-SPE column | GC–ECD HP-5 column (30 m, 0.32 mm i.d., 0.25 µm film), N$_2$ as the carrier gas and the flow rate was 1.0 mL/min; (confirmation with GC–MS\MS; TR-35MS column 30 m, 0.25 mm i.d., 0.25 µm film thickness). Helium as carrier gas at a constant flow rate of 1.0 mL/min) | 1.9–4.9 ng/g | 95–5-114.9 (1.9–4.9) 60.6–119.0 (2.7–12) for 0.01 µg/g 55.3–109.0 (2.3–4.6) for 0.1 µg/g Notes: Higher extraction efficiency compared to the other 3 extraction methods | [95]/2015 |

Table 2. *Cont.*

| Analytes | Sample Treatment | Technique | LODs, LOQs | Recovery % (RSD %) | Reference/Year |
|---|---|---|---|---|---|
| | ASE<br>Soil samples (5 g), solvent: acetone: petroleum ether (1:1, *v/v*) at 110 °C, 1500 psi with 6 min for heating, 22 min | | | 47.5–117.9 (1.4–6.0) | |
| | USE<br>30 min using 20 mL of acetone: petroleum ether (1:1, *v/v*) as the solvent, at 3500 rpm for 3 min | | | 58–9–127.9<br>(3.0–7.4) | |
| | QuEChERS<br>Soil samples (5 g), extraction solvent: 3 mL of water and 7 mL of acetonitrile; 6 g magnesium sulphate (MgSO$_4$), 1.5 g sodium chloride (NaCl), 0.75 g disodium citrate sesquihydrate (Na$_2$HCit1.5H$_2$O), and 1.5 g sodium citrate dihydrate (Na$_3$Cit$_2$H$_2$O); d-SPE: 50 mg of PSA (primary secondary amine), 150 mg of MgSO$_4$, and 50 mg of C18. | | 1–7–12.1 ng/g | 80–123.4 lowest recovery (56.8%) for o,p′-DDD<br>(1.7–12.1) | |
| 17 OCPs: p,p′-DDE, p,p′-DDD, p,p′-DDT, α-HCH, γ-HCH, β-HCH, δ-HCH, aldrin, dieldrin, endrin, endrin aldehyde, α-, β-endosulfan, endosulfan sulphate, heptachlor, heptachlor epoxide, and methoxychlor | QuEChERS<br>Air-dried soil samples (10 g), extraction: ACN; 4 g MgSO$_4$ (activated), 1 g NaCl<br>clean-up: 10 mL supernatant in 1.5 g MgSO$_4$ and d-SPE: 500 mg PSA<br>Dry residue reconstituted in 1 mL of n-hexane | GC–ECD T<br>G-5MS column (30 m and 0.25 mm i.d. with a 0.25 µm film thickness), nitrogen as the carrier gas at 1 mL/min | 3.74–11.33 ng/g (LOQ)<br>Highest LOQ was observed for β-endosulfan and lowest for aldrin 4.78 to 11.33 ng/g | 52.1–110.9<br>(0.19–3.57)<br>The concentration of ∑OCP ranged from 6.35 ng/g to 118.29 ng/g, with most the frequently found being β-endosulfan. | [4]/2020 |
| 19 OCPs: α-, β-, γ-, δ-HCHs, p,p′-DDE, o,p′-DDE, p,p′-TDE, o,p′-TDE, o,p′-DDT, p,p′-DDT, aldrin, dieldrin, heptachlor, heptachlor epoxide (isomer A), heptachlor epoxide (isomer B), α-endosulfan, β-endosulfan, and HCB | PHWE-SPE-SPME<br>Extraction PHWE: 10 g clean soil µg/kg extracted in 1 L ultrapure water at 1500 psi and 150 °C<br>SPE: concentrated at Lichrolut C18 cartridge<br>SPME:Fiber type: PDMS/DVB | GC–ECD<br>DB-XLB column (60 and 0.25 mm, 0.25 µm film thickness), helium as carrier gas | 24 and 22 ng/g, (LOQ) lowest HCB and highest p,p′-DDT | 80–115 and satisfactory precisions were obtained. Between LOQs and 500 lg/kg$^{-1}$ for most of the studied pesticides | [103]/2010 |

**Table 2.** *Cont.*

| Analytes | Sample Treatment | Technique | LODs, LOQs | Recovery % (RSD %) | Reference/Year |
|---|---|---|---|---|---|
| 19 OCPs: α-, β-, γ-, δ-HCHs, p,p′–DDE, o,p′-DDE, p,p′TDE, o,p′-TDE, o,p′-DDT, p,p′-DDT, aldrin, dieldrin, heptachlor, heptachlor epoxide (isomer A), heptachlor epoxide (isomer B), α-endosulfan, β-endosulfan, and HCB | PHWE-SPE-SPME Extraction PHWE: 10 g clean soil μg/kg extracted in 1 L ultrapure water at 1500 psi and 150 °C SPE: concentrated at Lichrolut C18 cartridge SPME:Fiber type: PDMS/DVB | GC–ECD DB608 column (30 m × 0.32 mm ID × 0.50 μm), helium as carrier gas | 1.69–50.4 ng/kg (LOQ) lowest HCB and highest p,p′-DDT | 68–90.1 (9.87–20.9) Note: HCH and DDT isomers: 1.99 to 7.85 μg/kg g. The most predominant compounds | [104]/2014 |
| Multi-residue analysis: 34 organochlorines including α-, β-,γ-, and δ HCH, LND, HCB, etc. | QuEChERS Extraction: 5 g of wet peat soil with 15 mL ACN (or dichloromethane; 4 g MgSO4, 1 g NaCl, 1 g sodium citrate dehydrate, and 0.5 g di-sodium hydrogen citrate sesquihydrate. d-SPE: 150 mg PSA and 950 mg MgSO4. | GC–MS ionization was performed in the electron impact mode and the quadrupole analyzer operated in the SIM mode (selected ion monitoring); DB-VRX 60 m × 0.32 mm ID × 1.80 m column, with helium as carrier gas at a constant flow of 2 mL min$^{-1}$ | 170.6–384.6 μg/Kg (LOQ) | 82.3–93.9 (4.0–8.5) | [72]/2012 |
| | ASE 4 g anhydrous Na$_2$SO$_4$, extraction: dichloromethane at 40 °C and 10 MPa with a pre-heat time of 5 min, followed by a 10 min static extraction and a 100% flush volume | | 1151–1808 μg/Kg | 64 (-δ HCH)-85 (10–13) | |
| 14 OCPS: α-, β-,γ-, and δ- HCHs, HCB, o,p′-DDT, p,p′-DDE, p,p′-DDD, aldrindieldrin, endrin, α- and β-endosulfan, and methoxychlor | QuEChERS Extraction: 5 g portion of dried sieved soil and 3 mL of water (hydration step) and 7 mL of ACN and or adding only 20 mL of CAN. 6 g magnesium sulphate (MgSO$_4$), 1.5 g sodium chloride (NaCl), 0.750 g disodium citrate sesquihydrate (Na$_2$HCit 1.5H$_2$O), and 1.5 g sodium citrate dihydrate (Na$_3$Cit 2H$_2$O) Clean-up: 50 mg of PSA, 150 mg of MgSO$_4$, and 50 mg of C18 | GC–ECD, column of 30 m, ZB-XLB (0.25 mm i.d., 0.25 μm), helium as carrier gas at constant flow rate of 1.3 mL/min Confirmatory by GC–MS/MS, SLB-5MS (30 m 0.25 mm, 0.25 m film thickness) column | 11.41 to 79.23 mg/g$^{-1}$ (LOQs) | 77 to 130% (with hydration step) 20 to 46% (without hydration step), (≤16%) | [121]/2012 |

Table 2. *Cont*.

| Analytes | Sample Treatment | Technique | LODs, LOQs | Recovery % (RSD %) | Reference/Year |
|---|---|---|---|---|---|
| α-, β-, γ-, δ-HCHs, p,p′-DDE, p,p′-DDD, o,p′-DDT, and p,p′-DDT | USE-SPE Extraction: hexane/dichloromethane (1:1, *v/v*) for 60 min SPE cartridges 1 g of florisil; 1 g silica gel and 1 g anhydrous sodium sulfate. | GC–ECD; HP-5 column | 0.05 to 0.20 ng/g. (LOD) | 73.3–96.2 (<10) | [45]/2009 [74]/2010 |
| α-HCH, β-HCH, γ-HCH, δ-HCH, p,p′-DDE, p,p′-DDD, o,p′-DDT, and p,p′-DDT | MAE-MSPE Extraction: ACN; MSPE sorbent: Fe$_3$O$_4$-NH2@MIL-101(Cr) | GC–ECD | 0.15–0.28 ng/g (LOD) | 71.2–102.4 (<10) | [105]/2019 |
| DDT, DDE | 10 g of soil with 25 mL of 90% methanol in water; supernatant diluted to 1:20 in 0.1% of fish gelatin in PBS (FG-PBS) | ELISA | - | 85–95 (0 to 10 ppm) | [137]/2002 |
| 49 POPs, between OCPs: DCF, α- and β-endosulfan, p,p′ DDD, p,p′ DDE, dieldrin, endrin, heptachlor, HCB, ALD, α-HCH, β-HCH, γ-HCH, δ-HCH, etc. | QuEChERS Extraction: 10 g of dried and sieved soil and 10 mL of acetonitrile—2.5% FA, 6 g of MgSO$_4$, and 1.5 g of CH$_3$COONa were shaken and sonicated in an ultrasonic bath. The supernatant was filtered (PET filters). | GC–MS/MS, J&WHP-5MS (Crosslinked 5% phenyl-methyl-polysiloxane, 15 m length, 0.25 mm i.d., and 0.25 μm film thickness, helium as carrier | 0.5–20 ng/g | 70–120 at 0.5–5 ng/g (below 20) | [131]/2002 [116]/2021 |
| α-HCH, β-HCH, γ-HCH, δ-HCH, p,p′-DDD; DDE; DDT, ALD, heptachlor; heptachlor epoxide, α- and β-endosulfan and sulfate endosulfan, etc. | FUSE 1 g of soil with 10 mL of hexane: dichloromethane (75:25) for 1 min in duplicate, and at 60% irradiation power. | GC–MS (EI), HP 5 MS (60 m, 0.25 mm, 0.25 μm) column, helium as carrier gas at flow of 1 mL/min | 2.5 to 15 ng/g | 75.8 (1.63)–101 (2.81) at 25 ng/g and 82.3 (0.18)–109 at the 75 ng/g | |

## 4. OCPs' Occurrence in Agricultural Soil

### 4.1. European Countries

Research on soil monitoring regarding OCPs residues does not have an extensive record in Europe. The most recent and complete study within Europe with regard to the analysis of currently used and banned pesticides including OCPs has been carried out recently by Silva et al. [13]. A total of 317 soil samples from 11 different European countries with the greatest portion of agricultural areas had been examined. In total, 21 OCPs were considered in this study, including aldrin, α- chlordane, γ- chlordane, chlordecone, o,p′-DDD, p,p′-DDD, o,p′-DDE, p,p′-DDE, o,p′-DDT, p,p′-DDT, dieldrin, α- and β-endosulfan, endosulfan sulfate, endrin, α-, β-, and γ-HCH, heptachlor, and heptachlor epoxide along with pentachlorobenzene. Based on the results of this study, the most frequently detected OCP and with the highest maximum content was p,p′-DDE (a metabolite of DDT). Its detection frequency was 23% (72 out of 317 soil samples) with a median content of 20 μg kg$^{-1}$ and a maximum content of 310 μg kg$^{-1}$. o,p′-DDD, p,p′-DDT, and dieldrin were also detected with the highest concentrations of 40 μg kg$^{-1}$, 10 μg kg$^{-1}$, and 60 μg kg$^{-1}$, respectively, with detection frequencies of 7%, 7%, and 5%, respectively. The detection frequencies of p,p′-DDD and alpha chlordane were 3% and 1%, respectively. The detection frequencies of γ-chlordane, o,p′-DDE, o,p′-DDT, heptachlor, and HCB were below 1% in all examined soils.

Italy carried out large-scale studies for the determination of OCPs in soil. One study inclusively interpreted the contamination level of 24 organochlorine pesticides (OCPs), including α-, β-, γ-, and δ-HCH, p,p′-DDE, p,p′-DDD, o,p′-DDT, o,p′-DDE, o,p′-DDD, p,p′-DDT, HCB, α-endosulfan, β-endosulfan, endosulfan sulfate, cis-chlordane, trans-chlordane, cis-nonachlor, trans-nonachlor, aldrin, dieldrin, endrin, heptachlor, MRX, and methoxychlor, in the soils was performed in Benevento, located in southern Italy [139]. Sixty-four surface soil samples (0–5 cm) were collected from the whole region of Benevento Province between December 2014 and February 2015. Quantifiable amounts of OCPs were detected in all examined soil samples, while HCBs and DDTs were the most frequently detected compounds, contributing up to 73.5% to the total concentration of OCPs. The concentrations of DDTs varied from not detected to 16.4 μg kg$^{-1}$, while p,p′-DDT and p,p′-DDE were the predominant compounds. The p,p′-DDD/p,p′-DDE ratio was found to be only > 1 in three soils, indicating the aerobic degradation of p,p′-DDT. The p,p′-DDT/p,p′-DDE ratio varied from 0.0010 to 9.6, indicating the occurrence of aged DDT. Taking into account the o,p′-DDT/p,p′-DDT ratio of all examined soil samples, the authors rejected the possibility of the 'dicofol-type DDT'. With regard to HCHs, their concentration levels ranged from not detected to 0.72 μg kg$^{-1}$, while β-HCH was the amplest isomer, contributing to 35.2% of the total HCHs, followed by α-HCH (32.1%), γ-HCH (26.1%), and δ-HCH (6.59%) [139]. The α-HCH /γ-HCH ratio ranged from 0.076 to 1.91, suggesting that HCHs come from both technical HCH and LND. However, the α-HCH /β-HCH ratio in all soils was lower than that of technical-grade HCHs, excluding the risk of the fresh input of technical HCH. HCB was noted as the most frequently detected OCP, with a detection frequency up to 96.9%, while the concentration level varied between not detected to 4.15 μg kg$^{-1}$. The authors proposed that the possible source of HCB emissions to the environment include its use as a pesticide, byproducts of chemical manufacturing processes, and waste incineration [139]. Endosulfan concentration levels ranged from not detected to 1.49 μg kg$^{-1}$. The detection frequencies of α-endosulfan, β-endosulfan, and endosulfan sulfate were 6.25%, 14.1%, and 45.3%, respectively, indicating that endosulfan residues in soil are mainly derived from historical uses. The chlordan concentration in the studied soils varied from not detected to 0.096 μg kg$^{-1}$, while its detection frequency followed the order: cis-Nonachlor (35.9%), cis-chlordane (9.38%), trans-chlordane (6.25%), trans-nonachlor (4.69%), and heptachlor (0%) [139]. Based on the results, the authors suggested that the recent application of technical-grade heptachlor and chlordane seems to be not possible. However, as heptachlor is rapidly transformed into heptachlor epoxide and other metabolites, additional studies are needed for the examination of heptachlor degradation products. Finally, the concentration

of drin compounds varied from not-detected to 0.37 µg kg$^{-1}$. Among them, dieldrin was the most frequently detected compound, followed by aldrin and endrin.

Another study was performed in the Campanian Plain in southern Italy for the determination of OCPs, including HCH isomers (α-β-γ- and δ-HCH) and DDTs (p,p′-DDE, p,p′-DDD, o,p′-DDT, and p,p′-DDT), in 119 soil samples collected from April to May 2011 [140]. The residual levels of HCHs varied between 0.03 and 17.3 µg kg$^{-1}$, while, for DDTs, they varied from 0.08 to 1231 µg kg$^{-1}$, indicating that DDT residues in soil are significantly higher than HCHs. Regarding HCHs, the analysis results revealed that β-HCH is the main isomer conducing to 57.7% of the total HCHs, followed by γ-HCH, δ-HCH, and α-HCH. In the case of DDTs, p,p′ -DDE and p,p′-DDT were noticed to be the dominant compounds with maximum concentrations of 589 µg kg$^{-1}$ and 962 µg kg$^{-1}$, accounting for 44.1% and 48.0% of total DDTs, respectively. The ratio of α-HCH/β-HCH and p,p′-DDT/p,p′-DDE in the area indicates the historical past uses of both OCPs.

An additional study was performed in an area located in central Italy that included four administrative zones (Latium, Marches, Tuscany, and Umbria), as well as in southern Italy, which consisted of seven zones (Abruzzo, Apulia, Basilicata, Calabria, Campania, Molise, and Sicily) for the determination of the status, regional sources, and pollution levels of 24 OCPs, including HCHs (α -HCH, β-HCH, γ-HCH, and δ-HCH), DDTs (o,p′-DDT, p,p′-DDT, o,p′-DDD, p,p′-DDD, o,p′-DDE, and p,p′-DDE), chlordanes (cis- and trans-chlordanes), heptachlor, heptachlor epoxide, aldrin, endrin, dieldrin, endrin aldehyde, endrin ketone, α-endosulfan, β- endosulfan, endosulfan sulphate, HCB, and methoxychlor, in urban and agricultural areas [141]. One hundred and forty-eight topsoil samples were collected between April and September 2016. The OCP concentrations varied between not detected and 1043.98 µg kg$^{-1}$ in urban soils and from not detected to 1914.1 µg kg$^{-1}$ in rural soils. The predominant OCP was endosulfan, with a 44.42% detection frequency, followed by DDTs, with a detection frequency of 17.6%, drins (15.75%), methoxychlor (12.17%), HCHs (6.08%), chlordane-related compounds (3.53%), and HCB (0.55%) in urban areas. In the case of agricultural soils, detection frequencies followed the order: drins (39.46%) > DDTs (29.94%)> methoxychlor (18.22%) > endosulfan (5.12%) > HCHs (5.06%) > chlordanes (1.40) > HCB (0.79%). Regarding the DDT concentration levels, it was observed that their total concentration fluctuated from not detected to 56.97 µg kg$^{-1}$ in soils from urban areas and from not detected to 632.95 µg kg$^{-1}$ in agricultural soils. In urban areas, the most predominant DDT isomer was p,p′-DDT, followed by p,p′-DDE, o,p′-DDD, o,p′-DDT, o,p′-DDE, and o,p′-DDD. In agricultural areas, the predominant compound was p,p′-DDT, with a detection frequency of 49.43%, followed by p,p′-DDE (29.96%), o,p′-DDT (9.32%), p,p′-DDD (6.17%), o,p′-DDD (3.64%), and o,p′-DDE (1.49%). The ratio of o,p′-DDT/p,p′-DDT demonstrated a wide range of values ranging from 0.0002 to 214 in urban soils and from 0.008 to 16.06 in rural soils. Consequently, the outcomes point towards a dominance of the historical application of technical DDT with the exclusion of some potential recent use of DCF in some urban areas. The ratio of p,p′-DDT/(p,p′-DDE + p,p′-DDD) was used to distinguish historical and recent DDT applications. In this case, this ratio presented a considerable range from 0.0014 to 55.02 for urban soils and from 0.006 to 40.42 for agricultural soils. It can be seen that residues of DDT for this survey could be related to a combined impact from historical and recent (illegal) applications. The illegal applications of DDT or DCF are most significant in urban areas. HCH (sum of α-HCH, β-HCH, γ-HCH, and δ-HCH) concentrations varied from not detected to 25.08 µg kg$^{-1}$ in urban areas and from not detected to 47.27µg kg$^{-1}$ in agricultural areas. β-HCH was the predominant isomer in all cases, which accounted for 60.25% in urban soils and 48.31% in agricultural soils. The indicative ratio of α-HCH/γ-HCH was used to distinguish the application of LND or technical HCH. In this case, the ratio of α-HCH/γ-HCH varied between 0.06 and 568 in urban soils and from 0.09 to 78.19 in rural soils. A total of 22% of the soils demonstrated a ratio of α-HCH/γ-HCH over 4.64; thus, the residues could probably be linked to applications of technical DDT. The α-HCH/β-HCH ratio was used to identify the HCH source. This ratio varied from 0.002 to 822 in urban samples

and from 0.005 to 180 in agricultural soils, indicating both historical and recent (illegal) applications of technical HCH in the study area. The total concentration of drins (aldrin, dieldrin, endrin, endrin aldehyde, and endrin ketone) fluctuated from not detected to 82.5 µg kg$^{-1}$ in urban soils and from not detected to 1212 µg kg$^{-1}$ in agricultural soils. Among drins, endrin ketone was the principal compound, accounting for 80.28% in urban areas and 93.71% in rural areas. Endrin ketone is the last photodegradation product of both endrin and endrin aldehyde, which is difficult to degrade further. The existence of drins may reveal that their residues in soils are primarily due to historical applications. The total concentrations of chlordanes (sum of cis-chlordane, trans-chlordane, heptachlor, and heptachlor epoxide) varied from not detected to 12.46 µg kg$^{-1}$ in urban areas and from not detected to 12.46 µg kg$^{-1}$ in agricultural areas. Heptachlor epoxide was the predominant compound with detection frequencies of 58.73% in urban areas and 67.56% in agricultural areas. The heptachlor/heptachlor epoxide ratio in both urban and rural soils indicated the historical application of the commercial chlordane. The total endosulfan (sum of α-endosulfan, β-endosulfan and endosulfan sulfate) concentration varied from not detected to 904.21 µg kg$^{-1}$, accounting for 44.32% of the total OCPs in urban areas, and from not detected to 92.99 ng g$^{-1}$, accounting for 5.12% of total OCPs in agricultural areas. The α-endosulfan/β-endosulfan ratio was used to decide the age of their residues in soil. In this case, this ratio ranged from 0.05 to 312.9 in urban areas and from not detected to 40 in agricultural areas. However, endosulfan sulfate was the dominant compound in rural regions. The authors attributed their findings to the recent (illegal) use of technical endosulfan in urban areas, while historical applications were the source of residues in rural areas. HCB concentration varied from 0.01 to 2.39 µg kg$^{-1}$ in urban soils and from not detected to 13.37 µg kg$^{-1}$ in rural soils. The authors recommended that HCB could be partly linked to the input of technical HCH or LND in the study area. The concentrations of methoxychlor oscillated from not detected to 53 µg kg$^{-1}$ in urban areas and from not detected to 521 µg kg$^{-1}$ in rural areas.

Romanian research groups carried out numerous monitoring studies for the determination of OCPs. Soil contamination by OCPs of 17 soil samples from southeastern Romania in the lower Danube–Black Sea basin was investigated for 15 OCPs, including HCHs (α-, β-, and γ-HCH), DDTs (p,p′-DDT, o,p′-DDT, p,p′-DDD, o,p′-DDD, p,p′-DDE, and o,p-DDE), heptachlor, chlordane, aldrin, dieldrin, endrin, and MRX [142]. Sampling was performed at two different depths of (0–5 cm) and (5–20 cm); however, the overall landscape features and different land use patterns were taken into consideration. The cyclodiene organochlorine pesticides aldrin, endrin, dieldrin, and MRX were not detected in any of the samples and depths. The total measured OCP concentrations fluctuated widely, depending on the sample location and depths. The total OCP concentration varied between 58 µg kg$^{-1}$ dry weight (dw) and 1662 µg kg$^{-1}$ dw for a 0–5 cm soil depth and between 6 µg kg$^{-1}$ dw and 12,644 µg kg$^{-1}$ dw for a soil depth of 5 to 20 cm. The most frequently detected OCPs were HCHs (α-HCH was the main compound), DDTs (p,p′-DDE was the dominant compound), and heptachlor, with concentrations ranging from 6 to 6818 µg kg$^{-1}$ dw, 27 to 5826 µg kg$^{-1}$ dw, and 108 to 873 µg kg$^{-1}$ dw, respectively. The authors concluded that the detection frequency of 65% for HCHs and 62.5% for DDTs indicates the extensive usage of these pesticides in the past [142]. Another study was performed in northeastern Romania for the determination of OCPs in soil, mosses, and tree bark samples [143]. Fifteen soil samples from forests in Moldavia were collected between August and September 2005, from a 0–5 cm soil depth for the determination of 15 OCPs including HCH isomers (α-, β-, γ-, and δ-HCH), DDT and its metabolites (p,p′-DDT, o,p′-DDT, p,p′-DDD, o,p′-DDD, p,p′-DDE, and o,p-DDE), HCB, oxychlordane (OxC), trans-nonachlor (TN), trans-chlordane (TC), and cis-chlordane (CC). The total concentration of HCHs ranged between 1.1 and 9.8 µg kg$^{-1}$ dw, while γ-HCH was the predominant isomer (detection frequency 64%) followed by β-, α-, and δ-HCH, with detection frequencies of 18%, 15%, and 4%, respectively [143]. The authors attributed the determined concentrations of HCHs to long-air transport deposition volatilized from agricultural nearby fields. The sum concentration of DDTs varied between

4.4 and 79 µg kg$^{-1}$ dw, while p,p'-DDE was the predominant compound indicating either an aerobic degradation of p,p'-DDT in soil or a long-range transport of p,p'-DDE after the conversion of p,p'-DDT following its release in the environment. The main contributors to the DDT total were p,p'-DDE and p,p'-DDT (54% and 38%, respectively) followed by p,p'-DDD, o,p'-DDT, and o,p'-DDD (5%, 2%, and 1%, respectively). The DDE/DDT ratio in the examined samples was <1, indicating a recent illegal application of DDT in the area. Soil samples from 20 agricultural fields from Iassy, located in northeastern Romania, and 27 sites including rural (near Timisoara, Arad, Ploiesti, and Cernavoda), urban (Bucharest, Timisoara, Arad, Baia Mare, Ploiesti, and Calimanesti), industrial (Copsa Mica, and Ramnicu Valcea), and waste incineration (Timisoara and Arad) sites from all over Romania were investigated for DDT and its metabolites (DDTs), HCB, and HCHs (α-, β-, and γ-HCH) [144]. The highest concentration for HCB (337.4 µg kg$^{-1}$) and the sum of HCHs (α-, β-, and γ-, HCH) (2585 µg kg$^{-1}$) were reported for the industrial site of Ramnicu Valcea located near a factory. For all other examined sites, excluding the Iassy agricultural sites, the concentration of HCB ranged between 0.3 µg kg$^{-1}$ and 1.9 µg kg$^{-1}$, while the sum concentration of HCHs varied from 6.5 µg kg$^{-1}$ to 29.2 µg kg$^{-1}$. With regard to the Iassy agricultural areas, the concentration for HCB was in the range from not detected to 0.18 µg kg$^{-1}$ in all examined soil samples, while the DDTs ranged from 3.5 µg kg$^{-1}$ to 119.5 µg kg$^{-1}$ in 18 out of 20 samples. There were two agricultural sites in the Iassy area with higher DDT concentrations, which were 492.2 µg kg$^{-1}$ and 1331 µg kg$^{-1}$. The authors concluded that Iassy agricultural soil samples were less contaminated with OCPs than other Romanian sites [144].

A study was conducted in tilled agricultural fields of the southern part of Romania (Dorobantu, Calarasi farm), in either burned (the burning treatment of soils is used for cleansing) or unburned soil samples collected from the surface horizon at a depth of 25 cm for the determination of 11 OCPs (α, β and δ-HCH, heptachlor epoxide, α- and γ-chlordane, endosulfan α-, and β-, dieldrin, p,p'-DDE, and p,p'-DDD) among other soil parameters. Based on the results of this study, the authors concluded that burned soils adsorb higher quantities of organochlorine pesticides compared with unburned soils [145]. From central Romania (Mures country), 20 soil samples were collected from 50 cm depth, between November 2004 and April 2005, from agricultural fields including apple orchards, vineyards fields, arable lands (maize, soybean, wheat, and potato fields), and greenhouses for the examination of various pesticides including OCPs. In 12 of the total 20 soil samples, OCPs were detected. DDTs were the major OCP found, while HCHs were not detected at concentrations above the limit of quantification (LOQ). The DDT and DDE concentrations in soil varied between 20 µg kg$^{-1}$ dw (arable agricultural fields) and 50 µg kg$^{-1}$ dw (apple orchards). Since soil samples were collected from 50 cm depth, the authors considered that higher concentrations of the detected OCPs could be found in the upper soil layers and that DDT may still have been being applied in fields at the time of the study. Dieldrin was also detected, with its concentrations ranging from 27 µg kg$^{-1}$ to 46 µg kg$^{-1}$. The authors attributed its presence to its past use [146].

A study was carried out during May 1990 for the determination of 12 OCPs, including α-HCH, β-HCH, γ-HCH (LND), δ-HCH, heptachlor, heptachlor epoxide, aldrin, dieldrin, p,p'-DDE, p,p'-DDD, p,p'-DDT, and dichlorobenzophenone, in water, soils, and earthworms along the Guadalquivir River, which runs across an agricultural area with the extensive use of pesticides in Spain [17]. The Guadalquivir basin is an economically significant area of the south of the Iberian Peninsula due to its closeness to a main metropolitan area (Cordova, Seville), which signifies the existence of several urban, commercial, and industrial locations in the district of the sampling stations. Ten soil samples were collected from the surface horizon (0–5 cm) of the studied area. The analysis results indicated the existence of HCHs and DDTs; however, aldrin, dieldrin, heptachlor, and heptachlor epoxide were not detected in any of the samples. In the analyzed soil samples, the concentration of HCHs (α-, β-, γ-, and δ-HCH) ranged from 0.66 to 2.49 µg kg$^{-1}$, and the detection frequencies were 50% for α-HCH, 80% for β-HCH, 70% for γ-HCH (LND), and 90% for

δ-HCH. The total concentration of DDTs (p,p′-DDE, p,p′-DDD, p,p′-DDT, and dichloroben-zophenone) varied from 3.49 to 46.30 µg kg$^{-1}$. The analysis results reveal that p,p′-DDE and p,p′-DDD were present in all examined samples, while dichlorobenzophenone and p,p′-DDT existed in 90% and 70% of the soil samples, respectively. The reported results for OCPs in soil revealed not only significant variation but also reflect their extensive use before their ban.

Thirty-two agricultural soils were sampled during the springs of 2007 and 2008 with regard to their contamination by DDT in southwestern Spain [147]. Samples were taken from areas surrounding Doñana National Park, which contains the region of Comarca de Doñana. Soils were taken from the surface horizon (depth 10 cm), and the characteristic crops of the area were considered as follows: strawberry (*n* = 11), citrus (*n* = 5), rice (*n* = 4), cotton (*n* = 3), vineyard (*n* = 3), and olive grove (*n* = 4). In some cases, DCF treatment was allowed. Two soil samples were gathered from the park and were intended as DDT contamination background soils. DDTs were identified in all soils except for the two samples collected from the park, which were used as blank soils.

A large-scale study was conducted in Germany, covering the whole country, with regard to the determination of POPs including 8 OCPs (HCB, dieldrin, p,p′-DDT, o,p′-DDT, p,p′-DDD, o,p′-DDD, p,p′-DDE, and o,p′-DDE) in 447 soils collected from forest sites [148]. DDT residue levels were up to 4000 µg kg$^{-1}$ dw in forest soils from eastern Germany; in contrast, their residue levels were lower than 100 µg kg$^{-1}$ dw in soils from western Germany. The low DDT levels in western Germany could be explained as omnipresent background contamination. The authors attributed DDT levels in German forest soils to historical applications based on their spatial distribution. Concerning HCB, its spatial distribution varied from below the Limit of Detection (LOD) to 24 µg kg$^{-1}$ dw; however, the authors concluded that its spatial distribution is difficult to be interpreted due to its multiple potential sources, such as its application as a fungicide, as a secondary byproduct in numerous chemical synthesis processes, and as a byproduct of combustion processes if chlorine compounds are involved. Generally, the spatial distribution of low concentrations possibly reveals background contamination, while high residue levels could be attributed to the abovementioned cases. Dieldrin was detected at very low residue levels (median of 2.7 µg kg$^{-1}$ dw), and its existence in soils could be attributed to background contamination levels. Aldrin was analyzed; however, its levels were below the LOQ in all examined samples.

In the UK, there is inadequate information regarding the historical usage of OCP and existing residues in soils and sediments. Archived background soils and sludge-amended soils gathered from long-term agricultural experiments in the UK were examined for the determination of OCPs including HCB, endosulfan, α-HCH, β-HCH, γ-HCH, heptachlor, cis-heptachlor epoxide, trans-heptachlor epoxide, aldrin, dieldrin, endrin, trans-chlordane, cis-chlordane, cis-nonachlor, trans-nonachlor, o,p′-DDE, p,p′-DDE, o,p′-DDD, p,p′-DDD, o,p′-DDT, and p,p′-DDT to determine trends over time. Thus, surface soil samples (0–23 cm) from a semirural area located 42 km north of London and soils from an untreated area (which did not receive any direct application) were collected [149]. The total OCP levels ranged from 0.1 to 10 µg kg$^{-1}$ dw, while γ-HCH, dieldrin, and p,p′-DDE constantly had the highest concentrations. HCB had the lowest concentration of all OCPs identified in this survey, and there was little variation among the years. A possible reason for the lack of an explicit HCB time trend could be the impact of other sources, as HCB also appears as a waste product in the production of several chlorinated products and some pesticides. Additionally, HCB emissions have been related to combustion and metallurgical processes involving the use of chlorine [150]. The detectable DDT isomers were o,p′-DDT, p,p′-DDT, and p,p′-DDE, while the residue levels of p,p′-DDE were higher than the other DDTs. The authors explained that their findings are inconsistent with the fact that DDE is a metabolite of DDT and that it is more persistent than the parent compound. With regard to their trend, it was concluded that all the DDT compounds exhibited the same time trend, with concentrations reaching their peak in the 1960s and then falling in the 1980s.

γ-HCH is present both in technical HCH and in its pure form as LND (>99% γ-HCH). Even though information on γ-HCH use in the UK is inadequate, its application has decreased since the the mid-1980s, which is reflected in the soil concentration trends. Cis- and trans-chlordane exhibited very low concentrations, with a peak in the decade of 1980. There is no documentation on its use in the UK. Among drin compounds, only dieldrin was detected. Dieldrin has been used in the past as a soil insecticide; however, it is also a metabolite of aldrin. In both the control and sludge-amended soil, o,p′-DDT, α-HCH, γ-HCH, and trans-chlordane along with dieldrin presented considerable decreases in concentrations from 1968 to 1990, and this is consistent with patterns of their use.

Twenty-four surface soil samples (0–5 cm) were collected between September and November 1994 from the southern part of Poland, specifically from the cities Kraków, Katowice, and Chorzów, which are highly industrialized and populated, for the determination of polychlorinated biphenyls and OCPs including HCB, HCHs, DDTs, and chlordanes [151]. Samples were collected from areas close to a mine machines factory, transformer station, coal mine, along with a nearby park, garden, barren soil, city center, and near a chloro-alkali plant. In Katowice, DDTs were the predominant compounds found in the range of 23–260 µg kg$^{-1}$, with an average value of 110 µg kg$^{-1}$, followed by HCB detected in the range of 0.46–30 µg kg$^{-1}$; HCHs ranged from 1.1 to 11 µg kg$^{-1}$ and chlordanes from 1.0 to 5.8 µg kg$^{-1}$. DDTs were the predominant compounds in the examined soils from Kraków, ranging from 4.3 to 2400 µg kg$^{-1}$, followed by HCHs ranging from 0.36 to 110 µg kg$^{-1}$; HCB levels fluctuated from 0.19 to 9.9 µg kg$^{-1}$, and chlordanes varied from 0.07 to 1.9 µg kg$^{-1}$. The soils from the city of Katowice, the more industrialized sampling site, were more contaminated, since the DDT residue levels were one order of magnitude higher than those in Katowice, while chlordane levels were comparable with soils from Katowice. DDTs were found to be in the order of p,p′-DDT, p,p′-DDE, and p,p′-DDD, while HCHs were in the order γ-HCH> β-HCH ≥ α-HCH. The α-HCH/γ-HCH ratios indicated the use of LND in much higher rates than the technical HCH in Poland.

Soil samples from the surface horizon (0–10 cm) were collected for the determination of OCPs, including HCHs (α-, β-, γ-, δ-HCH), p,p′-DDD, p,p′-DDE, p,p′-DDT, o,p′-DDT, o,p′-DDD, o,p′-DDE, methoxychlor, heptachlor, heptachlor epoxide, endosulfan (α-, β-endosulfan, endosulfan sulfate), aldrin, HCB, cis- and trans-chlordane, dieldrin, and endrin from the northeastern part of Poland (89 samples) and Almaty-region farms (32 samples) in southeastern Kazakhstan [152]. The average total concentrations od DDT in Polish and Kazakh soil samples were 104 and 97 µg kg$^{-1}$, respectively. DDTs and HCHs were still the major OCPs detected in the examined soils. The p,p′-DDT/(p,p′-DDE +p,p′-DDD) ratio was used to specify whether p,p′-DDT in soils was from historical or recent input, and it varied from 0.23 to 11.41 for both countries. Specifically, this ratio fluctuated from 0.48 and 11.41 for Polish samples, indicating both historical and recent applications, and between 0.23 and 0.75 for Kazakh soils, indicating only past applications. The ratio of o,p′-DDT/ p,p′-DDT was used to discriminate whether DDT contamination was instigated by the usage of technical DDT or DCF. In this case, the ratio of o,p′-DDT/p,p′-DDT ranged between 0.11 and 0.50 for both countries; however, it was higher for Kazakh samples, showing that the fresh application of DDT was mostly introduced by DCF, while, in the case of Polish samples, it showed that the recent application of DDT was mainly introduced by the use of technical DDT. DDT levels found in the soil from northeastern Poland and southeastern Kazakhstan were normally low compared with values around the world. Despite the fact that OCPs have been restricted worldwide and for long periods of time in many countries, it remains an abundant contaminant whose environmental levels are stated not to have declined in some areas.

A study on the determination of OCPs including HCB, α-HCH, β-HCH, γ-HCH, δ-HCH, and DDTs in soils (collected from the lake catchment areas) and sediments from two high-altitude European mountain lakes, Redon in the Pyrenees (in Spain) and Ladove in the Tatra mountains (between Slovakia and Poland), was performed [12]. DDT (p,p′-DDE and p,p′-DDT) residue levels were in ranges between 1.7 and 3.4 µg kg$^{-1}$ and 4.5

and 13 µg kg$^{-1}$ in Redon and Ladove soils, respectively. HCH concentrations fluctuated between 0.08 and 0.19 µg g$^{-1}$ and 0.28 and 0.49 µg kg$^{-1}$ in Redon and Ladove soils, respectively. The difference between the two lakes possibly reveals the historic use of this pesticide in its pure form or as a technical mixture including high proportions of α-HCH. The total concentration of HCB varied from 0.15 to 0.91 µg kg$^{-1}$ and 0.23 to 0.33 µg kg$^{-1}$ in Redon and Ladove soils, respectively. The low concentrations in these mountain areas suggested a lack of pollution sources. The authors concluded that the residues of OCPs are higher in the soils from the lakes in the Tatra mountains than in Redon, and this indicates the impact of the higher use and production of these compounds near the Tatra region. It is obvious that DDTs were the predominant OCPs in the area followed by HCB.

Topsoil samples (0–15 cm) were collected from the Plomin Power Plant (PPP) (25 samples) in Croatia and from Varaždin, an urban-industrialized area in north Croatia (16 samples), in March 2014, as well as in June and July 2013, for the determination of 13 OCPs including HCB, HCHs (α-HCH, β-HCH, and γ-HCH), DDTs (p,p′-DDE, p,p′-DDD, o,p′-DDT, and p,p′-DDT), aldrin, isodrin, heptachlor, heptachlor epoxide, and α-endosulfan [153]. HCB and p,p′-DDE were the only identified OCPs in soil samples from the Plomin Power Plant, while HCB was detected in more than 90% of the examined soils, and traces of p,p′-DDE were detected in nearly 70% of PPP soil samples. The occurrence of p,p′-DDE and the absence of the parent compound DDT revealed that there was no recent input of p,p′-DDT in soil in the vicinity of the PPP. The author attributed the presence of both OCPs to atmospheric transport. In Varaždin soil samples, the profile of OCPs was different than that in the soils from PPP. Varaždin city is surrounded by arable land, where pesticides are frequently used. HCB, p,p′-DDE, and p,p′-DDT were detected in more than 70% of the samples. Additionally, α-endosulfan, and γ-HCH were also identified. The total concentration of OCPs ranged from 0.86 to 21.0 µg kg$^{-1}$, while the total concentration of DDTs fluctuated from 0.74 to 19.9 µg kg$^{-1}$. The ratio of p,p′-DDE/p,p′-DDT was used as an indicator of recent or historical uses. The mean p,p′-DDE/p,p′- DDT ratio revealed a recent input of low concentrations of p,p′-DDT from the city atmosphere, probably from distant areas. HCHs were detected in more than 50% of the samples, and the γ-HCH was the only detected isomer, indicating its recent input probably through atmospheric transport. Traces of HCB were also identified, which varied from 0.05 to 1.82 µg kg$^{-1}$, with an average concentration of 0.08 µg kg$^{-1}$. The authors attributed its existence in all analyzed samples to atmospheric transport. The authors concluded that HCHs and DDTs were the predominant OCPs in both cases.

Various soil samples from different categories from central and southern Europe were investigated for the determination of OCPs [154]. Sampling sites were thoroughly chosen to represent a selection of background, rural, urban, and industrial areas. Samples were collected from 47 sites over a period of 5 months. Total HCH concentrations were below 1 µg kg$^{-1}$ in all samples, while total DDTs were from 1 to 60 µg kg$^{-1}$. The authors concluded that soil samples from the examined sites were found to be a sink for DDT and for γ-HCH.

A study was conducted for the determination of OCPs, including HCB, α-HCH, β-HCH, γ-HCH, o,p′-DDE, p,p′-DDE, o,p′-DDD, p,p′-DDD, o,p′-DDT, and p,p′-DDT in soils from Belgium (16 soil samples), Italy (6 soil samples), Greece (2 soil samples), and Romania (46 soil samples) [14]. Soil samples were collected from rural, urban, industrial, and waste-incineration areas. HCB residue levels were below 5.5 µg kg$^{-1}$ in samples from all countries; however, there were exceptions for some Romanian rural samples with concentrations up to 89.5 µg kg$^{-1}$. The γ-HCH/HCH ratios were similar in all countries and in urban and rural sites; however, some exceptions for Romania were observed. With regard to DDTs, their concentrations in soils varied from 0.6 to 22.4 µg kg$^{-1}$ in the case of Belgium, from 1.8 to 60.4 µg kg$^{-1}$ for Italy, 24.1 µg kg$^{-1}$ for Greece (two samples), and from 3.6 to 561.4 µg kg$^{-1}$ for Romania. The concentrations of DDT were higher in the rural sites in all countries except for Greece. The ratio of p,p′-DDT/DDTs was below 0.35 for

Belgian and Italian soils, indicating the historical use of DDT. In the case of Greece and Romanian soils, this ratio was higher than 0.66, indicating more recent applications.

### 4.2. African Countries

A few studies in African countries have been dedicated to the soil monitoring of OCPs. Recent research regarding the determination of 15 OCPs, including α-, β-, and δ-HCH, HCB, LND, aldrin, dieldrin, endrin, heptachlor, heptachlor epoxide, α- and β- endosulfan, p,p′-DDE, p,p′-DDT, and chlorothalonil, was performed in Nigeria [155]. Twelve samples were collected from the plain of the Onuku River from different fields and from the surface soil horizon of 0–20 cm using a hand auger for sampling. The authors observed high spatial distribution in most OCPs, while their individual concentrations ranged from not detected to 5250 μg kg$^{-1}$. The highest concentration was observed for chlorothalonil (4510 μg kg$^{-1}$) and p,p′-DDT(5250 μg kg$^{-1}$). However, the total concentration of OCPs in the examined soil samples ranged from 13,870 to 21,100 μg kg$^{-1}$. The authors ascribed the low levels of most OCPs to possible run-off to the Onuku River.

Sixty-six soil samples were collected during a sampling campaign performed between February and August 2003 in rice fields in the Rufiji River Delta, a worldwide known wetland, located in the east coast of Africa in Tanzania [156]. One of the objectives of this study was the determination of 16 OCPs, including HCB, HCHs, DDT, and its degradation products, cyclodienes (aldrin, dieldrin, heptachlor, heptachlor epoxide, γ -chlordane, endrin, and keto-endrin), in soil samples collected during the application period (February 2003 and April 2003) and post-application (June 2003 and August 2003). For HCB, there was no noted significant concentration variation in the examined samples during the different sampling events, and the mean concentration was reported to be 0.4 μg kg$^{-1}$ dw. Among HCHs (α-HCH, β-HCH, γ-HCH, δ-HCH), γ-HCH (LND) was more frequently detected than the other isomers, and its mean determined concentration was 2.4 μg kg$^{-1}$ dw in samples taken during the application period and 1.3 μg kg$^{-1}$ dw for post-soil-application samples. Nevertheless, in soil samples, the sum concentration of HCHs revealed a substantial reduction in concentrations between surveys (4.6 to 2.7 μg kg$^{-1}$ dw). Residues of heptachlor, heptachlor epoxide, aldrin, keto-endrin, γ-chlordane, and dieldrin were identified in all soil samples, but concentrations ranged between 0.1 and 3.0 ng g$^{-1}$ dw. DDTs were identified in all soil samples, while p,p′-DDE was the predominant compound, contributing to 70% of the total DDTs.

A study was conducted in former storage sites in Tanzania after 5–14 years of their clean-up for the determination of the level, composition, and allocation of OCPs in soil, including DDT, DDD, DDE, HCHs, aldrin, dieldrin, endrin, endosulfans, chlordanes, and heptachlor. Soil samples were gathered between January and April 2009 from six contaminated sites [157]. For soils with high clay content, samples were taken from two different depths: 5–10 cm and 30 cm; for sandy soils, samples were taken from 10–30 cm and at deeper depths from 50 cm and 3 m. The analysis results revealed that 27 compounds were detected in the examined samples, while the DDTs were the most frequently detected compounds, followed by HCHs, aldrin, dieldrin, endrin, endosulfans, chlordanes, and heptachlor. DDT concentrations ranged from 0.01 to 250,000 μg kg$^{-1}$, while p,p′-DDT was the predominant compound, and its concentration was higher than o,p′-DDT in all cases, indicating that technical DDT was the main source of contamination. HCHs (α-HCH, β-HCH, γ-HCH, and δ-HCH) were detected in most of the examined cases, with concentrations ranging from 40 to 140,000 μg kg$^{-1}$, while, for the rest of detected OCPs, their concentrations varied from 2900 to 3300 μg kg$^{-1}$. The results suggested that there were no considerable degradations/transformations in the OCPs for most of the sites. The maximum concentrations of the compounds were primarily observed in surface soil samples, and there were differences in the distribution among the sampling depths. The results reveal risks and concerns for both public health and the environment.

Soil samples from the surface horizon (0–20 cm) were collected and analyzed for the occurrence of OCPs, including β-HCH, α-HCH, δ-HCH, γ-HCH, p,p′-DDT, p,p′-DDE,

p,p′-DDD, and o,p′-DDT, in three different counties in Kenya as a picture of different sections of the country, providing the chance for comparison between sites [158]. Sampling focused on areas affected by anthropogenic activities in the adjacent urban centers and agricultural areas. Fifty-two soil samples were gathered, and twenty of them were obtained from Kapsabet town, nine from Voi town, and twenty-three were taken from Nyeri town. The sampling sites from Nyeri, a city located in the central Kenya, which is well-recognized for the farming of both coffee and tea and where several manufacturing industries are set, including a soft drink bottling plant, leather products processing plant, and tea and coffee manufacturing industries, were locations affected by anthropogenic activities from nearby urban and agricultural areas. Samples were also taken from Voi, a town located on the Nairobi–Mombasa highway close to a National Park with intense farming activities. The third sampling area was in Kapsabet, the capital of Nandi County, with maize and tea industries. The total concentration of OCPs were in the range of 0.03–52.7, 0.24–24.3, and 0.06–22.4 µg kg$^{-1}$ for Kapsabet, Nyeri, and Voi, respectively. The predominant OCPs in the three study areas were from the HCH families. HCH (α-HCH, β-HCH, γ-HCH, and δ-HCH) concentrations varied from 0.03 to 48.1, 0.06 to 6.86, and 0.24 to 4.72 mg kg$^{-1}$ for Kapsabet, Voi, and Nyeri, respectively, while those for DDTs fluctuated from not detected to 19.6 µg kg$^{-1}$, not detected to 15.5 µg kg$^{-1}$, and not detected to 4.68 µg kg$^{-1}$ in Nyeri, Voi, and Kapsabet, respectively. The residues of HCH isomers in Kapsabet were in the sequence β-HCH > α-HCH > δ-HCH > γ-HCH; however, in Nyeri and Voi, HCH isomers were in the sequence β-HCH > γ-HCH > δ-HCH > α-HCH. DDTs were present in the Nyeri soil samples, and this could be attributed to the use of pesticides containing DDT in the tea and coffee farms. In Kapsabet town, trace levels of p,p′-DDE and o,p′-DDT were detected. The identified amounts were attributed to the application of pesticides in agriculture. The concentrations of p,p′-DDE and p,p′-DDD in Voi were related to the agricultural activity. Among the DDTs, o,p′-DDT was the predominant compound in all cases. o,p′-DDT could imply that the use of technical DDT pesticides, which could have led to an increase in DDTs in the soil in Kenya. The highest concentration of HCHs was in Kapsabet (0.03–48.1 µg kg$^{-1}$), whereas the highest DDT concentration was in Voi (not-detected–15.5 µg kg$^{-1}$). Source identification revealed OCP pollution originated from the present usage of DDTs to control insect-borne diseases and from the use of LND in agriculture. The enantiomeric ratios of α-HCH/γ-HCH were <3, indicating the use of LND, while the ratios of DDE/DDT were <1, suggesting recent input of DDT.

One study was performed for the first evaluation of the occurrence, residue levels, spatial distribution, and sources of POPs and OCPs, including aldrin, dieldrin, DDTs, endosulfan, endrin, HCHs, heptachlor, heptachlor epoxide, HCB, isodrin, methoxychlor, and MRX, in soils in the lower Nyabarongo catchment in central Rwanda [159]. One hundred and eight soil samples were collected from topsoil (5–10 cm) in fields situated along the main rivers, Nyabarongo and Nyabugogo, and their tributaries. DDTs were the most frequently detected compounds with a detection frequency of 89%, while the isomers' detection frequency followed the order p,p′-DDE > p,p′-DDT > p,p′-DDD > o,p′-DDT > o,p′-DDD, and o,p′-DDE. Dieldrin was found in 7% of samples, aldrin in 3%, and heptachlor and heptachlor epoxide (endo-) were detected in one sample each. DDT concentrations varied from 100 to 120,000 µg kg$^{-1}$ dw, dieldrin concentrations ranged from 0.53 to 18 µg kg$^{-1}$ dw, aldrin from 0.38 to 0.59 µg kg$^{-1}$ dw, and heptachlor from 0.14 to 0.19 µg kg$^{-1}$ dw. The higher DDE/ DDT rate (0.22) compared to the DDD/DDT ratio (0.08) indicated the aerobic degradation of DDT as the predominant pathway. The (DDE + DDD)/DDT ratio in the examined samples varied considerably at various sampling points, amplifying different ages and sources of DDT residues in the area. Aldrin and dieldrin detection frequencies were 3% and 7%, respectively. The concentration levels for aldrin ranged from 0.38 to 0.59 µg kg$^{-1}$ dw and for dieldrin from 0.53 to 18 µg kg$^{-1}$ dw. Since dieldrin was demonstrated to have higher concentrations compared to aldrin, this suggests a current illegal application of both compounds in random places in the area and surroundings.

*4.3. Asian Countries*

China undoubtedly has the most abundant data on monitoring the OCPs in soils in Asia. One study was conducted for the determination of the concentration levels, sources, and stocks of polycyclic aromatic hydrocarbons (PAHs) and OCPs in 55 surface vegetable soils in the watershed of the Pearl River Delta, located in southern China. The area is of great economic interest as it is developing fast in industrial and agricultural activities. Soil samples were gathered during August 2005 from the surface horizon (0–20 cm) for the determination of 18 OCPs including DDTs (p,p′-isomers), HCHs, heptachlor, aldrin, heptachlor epoxide, α-endosulfan, β-endosulfan, dieldrin, endrin, endrin aldehyde, endosulfan sulfate, endrin ketone, and methoxychlor [160]. The total concentrations of HCHs varied from 0.19 to 42.3 µg kg$^{-1}$ in all soil samples, while the authors attributed their existence to the agricultural activities of the area, as it was widely used in the past. Among the four HCH isomers (α-, β-, γ-, and δ-HCH), β-HCH was the predominant compound in all soil samples, and the authors thought that this finding was attributed to its thermodynamically stable structure, with all chlorine atoms being in equatorial positions, and in addition, it is the least reactive and maximally persistent isomer, while its biodegradation is slower. The total DDT content ranged from 3.58 to 831 µg kg$^{-1}$, while p,p′-DDT and p,p′-DDE were the isomers detected at higher concentrations. As DDT could be transformed to DDE by soil microorganisms under aerobic conditions and DDD under anaerobic conditions, it was expected that DDE residue levels should be higher in surface soil samples, and it was confirmed by the findings in that case. Additionally, the DDT/(DDD + DDE) ratio was reported as being lower than 1 in some of the examined samples, indicating historical DDT use, while in the majority of the samples, the ratio was much higher than 1, suggesting fresh DDT application in the studied soils.

Thirty-two topsoil samples (0–15 cm) were gathered and analyzed for the determination of OCPs including HCHs (β-, γ-, δ-, and ε-HCH), HCB, heptachlor, heptachlor epoxide, trans-chlordane, cis-chlordane, α-endosulfan, β-endosulfan, o,p′-DDE, p,p′-DDE, o,p′-DDD, p,p′-DDD, o,p′-DDT, p,p′-DDT, and MRX of arid and semiarid areas of eight prefectures of four provinces in northwest China in 2011 [161]. Soil samples were collected from urban (15 samples) and rural (14 samples) areas, as well as background sites (3 samples). The detection frequencies of total HCHs, DDTs, endosulfans, and heptachlors in the soils were up to 100%, suggesting their widespread existence in the arid and semiarid areas of northwest China, while their total concentration varied from 0.90 to 133.44 µg kg$^{-1}$. DDTs were demonstrated to exhibit the highest concentration in the examined soils, followed by HCHs, HCB, heptachlors, endosulfans, and chlordanes; thus, DDTs were the dominant pollutant in arid and semiarid areas of northwest China. DDT concentrations varied from 0.1 to 120.49 µg kg$^{-1}$, with a mean value of 12.52 µg kg$^{-1}$, while HCH concentrations varied from 0.17 to 9.39 µg kg$^{-1}$ in soils from arid and semiarid areas. The soil concentrations of HCB fluctuated from not detected to 11.71 µg kg$^{-1}$ with an average concentration of 1.21 µg kg$^{-1}$. The authors noticed that soil concentrations for heptachlors, endosulfans, and chlordanes in arid and semiarid areas of northwest China were much lower than DDTs, HCHs, and HCB and more were uniform amongst all sites. The soil concentrations varied from 0.1 to 2.19 ng g$^{-1}$ with an average concentration of 0.61 µg kg$^{-1}$ for heptachlors, from 0.01 to 0.84 µg kg$^{-1}$ with a mean of 0.09 µg kg$^{-1}$ for endosulfans, and from not detected to 0.28 µg kg$^{-1}$ with a mean level of 0.04 µg kg$^{-1}$ for chlordanes, respectively. The composition of DDTs in the examined soils followed the order: p,p′-DDT (68.5%) > o,p′-DDT (18.9%) > o,p′-DDE (7.2%) > p,p′-DDE (6.8%) > p,p′-DDD (4.0%) > p,p′-DDD (1.0%), demonstrating the dominance of p,p′-DDT. The (DDE + DDD)/DDT ratio in arid and semiarid areas of northwest China ranged from 0.06 to 0.41, indicating the recent illegal input of technical DDT. The o,p′-DDT/p,p′-DDT ratios at nearly all sampling sites were lower than 1.3, indicating that DDTs in arid and semiarid areas of northwest China were mainly from the application of DCF, a pesticide containing on average 11.4% o,p′-DDT and 1.7% p,p′-DDT. The major sources of HCHs in soil samples are mainly from the direct application of technical HCH and LND; thus, the authors conducted composition

analysis among HCH isomers to assess the potential source of HCHs using the $\alpha$-HCH/$\gamma$-HCH ratio [161]. The $\alpha$-HCH/$\gamma$-HCH ratios in the rural areas were between 0.16 and 15.5, indicating the historical use of HCH in two provinces. However, the $\alpha$-/$\gamma$-HCH ratio in most of the areas was less than 3, indicating the recent input of LND. Investigating HCH isomers suggests that $\beta$-HCH is the highest isomer among HCHs, accounting for 59.1% of total HCHs, followed by $\alpha$-HCH (24.3%), $\delta$-HCH (8.3%), and $\gamma$-HCH (8.2%). Higher $\beta$-HCH can be ascribed to its persistence in soil. HCB was never used in China after its ban in 1982; thus, its residues indicate activities of the chemical industry in this area. The $\alpha$-endosulfan/$\beta$-endosulfan ratio presented significant variation since it fluctuated from 0.18 to 10.85, indicating both historical uses and recent input. Technical chlordane is still being used in China against termites. The cis-chlordane/trans-chlordane ratio in the soils in the present study was less than 1, indicating that chlordane in arid and semiarid areas of northwest China was very probable due to long-range atmospheric transport. With regard to heptachlors, it was noticed that heptachlor epoxide had higher concentrations than those of heptachlor in most of soil samples, indicating historical uses of this pesticide [161].

An intensive soil-sampling program was implemented in the outskirts of Beijing, China, for the determination of HCHs ($\alpha$-HCH, $\beta$-HCH, $\gamma$-HCH, and $\delta$-HCH) and DDTs (p,p'-DDD, p,p'-DDT, p,p'-DDE, and o,p'-DDT) in shallow subsurfaces (5–30 cm depth) and deep soil layers (150–180 cm depth). Forty-seven shallow subsurface soil and forty-six deep-layer soil samples were collected and analyzed [162]. The residue levels of HCHs in the shallow subsurface soils varied between 1.36 and 56.61 $\mu$g kg$^{-1}$ dw with an average concentration of 5.25 $\mu$g kg$^{-1}$, while the DDTs residue levels fluctuated from 0.77 to 2180 $\mu$g kg$^{-1}$ dw with an average concentration of 38.66 $\mu$g kg$^{-1}$. The concentrations of HCHs and DDTs in the deep-layer soils varied from 0.40 to 5.36 $\mu$g kg$^{-1}$ (average value 0.99 $\mu$g kg$^{-1}$) and from 0.13 to 66.98 $\mu$g kg$^{-1}$ (average value 0.82 $\mu$g kg$^{-1}$), respectively. The spatial distribution of HCHs and DDTs for both the shallow subsurface soils and deep-layer soils was comparable, with higher amounts found in the east, west, and northwest sites for both OCPs, indicating the known historical usage in the area. The percentages of HCHs in the shallow subsurface soils were as follows: $\beta$-HCH > $\gamma$-HCH > $\alpha$-HCH > $\delta$-HCH, which proved that the technical HCH had not been in use in the Beijing area for quite a long time; meanwhile, those in the deep soil layers were: $\gamma$ > $\alpha$~$\beta$ > $\delta$, demonstrating that $\gamma$-HCH and $\alpha$-HCH had higher leaching abilities than $\beta$-HCH and $\delta$-HCH [162]. The percentages of individual compounds in both shallow subsurface soils and deep-layer soils followed the order: p,p'-DDE > p,p'-DDT > p,p'-DDD > o,p'-DDT. The percentage of p,p'-DDE in the deep-layer soils was lower than that in the shallow subsurface soils; however, that of p,p'-DDD was higher, since DDT is likely to be dechlorinated to DDD in the anaerobic condition in the deep soil layer. Most soils included more DDE than DDT, suggesting DDT residues are from historical sources. Extremely high DDT/DDE ratios were identified in the deep layer soil, possibly because of the long-term leaching of p,p'-DDT and the retarded degradation of DDT to DDE in the deep-layer soils [162].

Another study was performed the Pearl River of China for the determination of POPs including OCPs in air, water, and soil [163]. OCPs, such as HCHs, DDT, HCB, chlordane, dieldrin, and aldrin, were extensively employed in agricultural activities in China from 1950 until 1983. Additionally, the manufacture of HCHs and DDTs in China accounted for 33% and 20% of the overall globe production, respectively. In the study area, it was assessed that the use of OCPs was about 76,000–100,000 tons annually. Sixty-three cultivated soils, including those from vegetable, banana, sugar cane, and fruit plantations, and non-cultivated soil samples were analyzed for the determination of OCPs. The analysis results indicated that DDTs and HCHs ranged from 15 to 125 $\mu$g kg$^{-1}$ in 70% and 2 to 30 $\mu$g kg$^{-1}$ in 80% of the examined soil samples, respectively, while the highest concentrations were detected in the cultivated soils. Because DDE was the predominant DDT compound in most of the samples and considering the DDE/DDT ratio, it could be suggested that an on-land weathering process took place. The higher HCH/DDT ratio in non-cultivated

soils compared to cultivated soils indicates that these OCPs were mainly input from air precipitation.

Another study was performed in 2009 on 61 surface soils (0–20 cm) from paddy (33 soils), upland (22 soils), and wetland fields (6 soils) in the province around the Hongze Lake in China for the determination of organochlorine pesticides, including HCHs (α-HCH, β-HCH, γ-HCH, and δ- HCH) and DDTs (p,p′-DDD, p,p′-DDE, o, p′-DDT, and p,p′-DDT) [164]. The levels of OCPs were higher in the cultivated soils than in wetland soils, while DDTs and HCHs were the prevalent contaminants with detection frequencies 94.55% and 85.4%, respectively, though β-HCH and p, p′-DDE were the two major compounds of HCHs and DDTs, respectively. HCB was detected in 54.5% of the examined soils. The total concentration of OCPs varied from 4.80 to 219.10 μg kg$^{-1}$. DDT and DCF are considered the major sources of the existence of DDTs in the environment; thus, the authors, based on the o, p′-DDT/p, p′-DDT ratio (which ranged from 0.31 to 0.45) in agricultural soil samples, indicated the possibility of the historical use of DCF in this region. The DDT/(DDD + DDE) and α-HCH/γ-HCH ratios indicated that residues of HCHs and DDTs in soil resulted from historical past uses.

Ninety three surface soils (0–20 cm) from a vital agricultural area in Zhangzhou City, located south of the Fujian Province in China were gathered from paddy fields, vegetable lands, orchards, and tea plantations for the determination of eight OPCs, including α-HCH, β-HCH, γ-HCH, δ-HCH, p,p′-DDE, p, p′-DDD, o,p′-DDT, and p,p′-DDT [8]. DDTs and HCHs presented the highest detection frequencies, as they were detected in all examined samples, while their DDT concentrations (sum of p,p′-DDE, p,p′-DDD, o,p′-DDT, and p, p′-DDT) fluctuated from 0.64 to 78.07 μg kg$^{-1}$ with an average of 3.86 μg kg$^{-1}$; HCHs (sum of α-HCH, β-HCH, γ-HCH, and δ-HCH) ranged from 0.72 to 30.16 μg kg$^{-1}$ with an average of 9.79 μg kg$^{-1}$. Amongst HCHs, β-HCH, and δ-HCH isomers had the highest residue levels, indicating the historical usage of HCH. The total concentrations of HCHs and DDTs varied between soils from different land uses as follows: paddy fields > vegetable lands > tea plantations > orchards, and tea plantations > orchards > paddy fields > vegetable lands, respectively. The α-HCH/γ-HCH ratio was used for the identification of contamination source (LND or technical DDT), and it was concluded that LND was widely used in the past. The (p,p′-DDE + p,p′-DDD)/p,p′-DDT ratio was used to identify the pollution source (DCF or DDT), and it was concluded that technical DDT was widely used in the past. However, the exact contribution of DCF in the total DDTs was found by the o,p′-DDT/p,p′-DDT ratio, and the contribution of dicofol-type DDT was 23% to the paddy fields, 26% to the vegetable lands, 82% to the orchards, and 66% to the tea plantations [8].

A pertinent study was conducted in college school yards in Beijing, China for the determination of 15 OCPs including HCH (α-HCH, β-HCH, γ-HCH, and δ-HCH), heptachlor, heptachlor epoxide, chlordanes (cis and trans chlordane), endosulfans (a and b-endosulfan), p,p′-DDE, p,p′-DDD, p,p′-DDT, o,p′-DDT, HCB, and 2,4,5,6-tetrachloro-mxylene [165]. Soil samples were gathered from the surface horizon (0–20 cm) in 2006. In that study, DDTs were found to be the major soil contaminant, accounting for 93.7% of the total OCPs, followed by HCHs (2.25%) and HCBs (1.82%). Other contaminants such as chlordanes, heptachlors, and endosulfans comprised 0.51%, 1.05% and 0.79% of the 15 OCPs, respectively. The total OCP concentration varied from 21.25 ng g$^{-1}$ to 276.45 μg kg$^{-1}$, while the total concentration of HCH (α- HCH, β- HCH, γ- HCH, and δ-HCH) varied from 0.40 to 3.72 μg kg$^{-1}$ with the average value of 2.25 μg kg$^{-1}$. The authors concluded that since Beijing is in the temperate zone where pollutants cannot be easily evaporated from soil, the HCH levels in soils could be considered as unimportant pollution. Among four HCH isomers, β-HCH was the predominant compound, indicating the recent application of HCH. Based on the α-HCH/γ-HCH ratio, it was concluded that both technical HCH and LND were used. DDTs presented the highest concentration among the other OCPs and were in the range of 0.42–76.77 μg kg$^{-1}$ for p,p′-DDE, 0.67–26.59 μg kg$^{-1}$ for p,p′-DDD, 0.80–163.90 μg kg$^{-1}$ for o,p′-DDT, and 0.31–104.58 μg kg$^{-1}$ for p,p-DDT. Taking into account that o,p′-DDT had the highest concentration followed by p, p′-DDE, p,p′-DDT, and p,p′-DDD, as well

as the o,p′-DDT/p,p′-DDT ratio, the authors concluded recent DCF application had occurred. The (p,p′-DDE + p,p′-DDD)/p,p′-DDT ratio was quite variable, ranging from 0.12 to 294.37 μg kg$^{-1}$, indicating both historical uses (70% of the samples) of DDT and recent applications. HCB ranged from 0.13 to 5.13 μg kg$^{-1}$ with the average concentration of 1.78 μg kg$^{-1}$. Regardless of the fact that HCB is banned, a possible source could be waste from chlorine-related industries. Chlordanes, heptachlors, and endosulfans were detected in the ranges of 0.03–0.88 μg kg$^{-1}$, 0.16–4.84 μg kg$^{-1}$, and 0.14–2.41 μg kg$^{-1}$, respectively. Based on the detected concentrations, the authors attributed their existence to past uses.

One study was performed with plastic shed soils and open-field agricultural soils planted primarily with corn and wheat from five provinces and one municipality of northern China, namely, Henan, Shandong, Liaoning, Jilin, Heilongjiang, and Tianjin, to examine the pollution status of soils from OCPs [166]. Twenty OCPs, including HCHs consisting of α-HCH, β-HCH, γ-HCH, and δ-HCH; DDTs, comprising o,p′-DDT, p,p′-DDT, p,p′-DDD, and p,p′-DDE; chlordanes (CDs), containing cis-chlordane and trans-chlordane; endosulfans, comprising α-endosulfan, β-endosulfan, and endosulfan sulfate; heptachlors, including heptachlor and heptachlor epoxide; and drins, containing aldrin, dieldrin, endrin, endrin ketone, and endrin aldehyde, were analyzed in 52 soil samples from plastic shed soils and 52 soil samples from the surface horizon (0–20 cm) of open-field agricultural soils collected from April to November 2018. The concentration of OCPs in the plastic shed and open-field soils were in the range of 40.1–2555 μg kg$^{-1}$ and 19.1–746 μg kg$^{-1}$, respectively. It is obvious that the plastic shed soils concealed considerably greater amounts of total OCPs than the adjacent open-field soils. The analysis results indicate that endosulfans, chlordanes, HCHs, and DDTs were the principal OCPs in the plastic shed soils, while only chlordanes and DDTs had substantially greater residual levels than other OCPs in the open-field soils. The detection frequencies of HCHs were more than 83% in the plastic shed soils, while β-HCH was the predominant compound with an average concentration of 27.8 μg kg$^{-1}$. The detection frequencies of β-HCH and γ-HCH were below 15% in the open-field soils. Considering the ratio of α-HCH/β-HCH, the authors concluded thatm with regard to plastic shed soils, there was no likelihood of the fresh application of either HCH or LND (γ-HCH) and that the concentration of HCH isomers in open-field soils indicated historical applications. p,p′-DDT was the major constituent of DDT residues both in open-field and plastic shed soils, accounting for 65.7% and 64.0% of the total DDT residues, respectively, indicating the fresh input of DDT in agricultural soils. Cis-chlordane was the main component of chlordanes, signifying that the new application of chlordane in this area was doubtful in recent years. The α-endosulfan/β-endosulfan ratio and the existence of endosulfan sulfate (a degradation product of endosulfan) in most of the soil samples suggests that the recent application of technical endosulfan was dubious in most sampling sites. However, there were rare cases indicating illegal recent use. Heptachlors and drins were found in low amounts and detection rates; thus, no further consideration was given.

Yu et al. summarized the results of nearly 120 studies performed between 2004 and 2018 in more than 2000 agricultural and urban soil samples, collected from 29 provinces and municipalities of China from various soil depths (<50 cm), for the determination of OCPs and PCBs with regard to their spatial and temporal distribution besides their pollution sources. Among the OCPs, the authors was focused on DDTs (p,p′-DDE, p,p′-DDD, o,p′-DDT, and p,p′-DDT) and HCHs (α-, β-, γ-, and δ-HCH) [167]. It was noticed that the entire OCP concentration varied from 7600 to 37,331 μg kg$^{-1}$, while the total concentrations of DDTs and HCHs varied from 2.9 to 26,723 μg kg$^{-1}$ and from 0.4 to 2943 μg kg$^{-1}$, respectively. The reported concentrations for DDTs isomers varied from: 0.05 to 139 μg kg$^{-1}$ for p,p′-DDE, 0.1 to 57 μg kg$^{-1}$ for p,p′-DDD, 0.05 to 505 μg kg$^{-1}$ for o,p′-DDT, and 0.8 to 488 μg kg$^{-1}$ for p,p′-DDT, indicating that o,p′-DDT was the predominant DDT compound. The HCH isomer concentration ranged from 0.08 to 420 μg kg$^{-1}$ for α-HCH, 0.2 to 155.5 μg kg$^{-1}$ for β-HCH, 0.05 to 119.3 μg kg$^{-1}$ for γ-HCH, and 0.06 to 53.5 μg kg$^{-1}$ for δ-HCH, indicating that α-HCH was the main compound. The authors observed that

urban soils presented higher pollution levels by OCPs compared to agricultural soils and ascribed this finding to rapid economic growth and the development of China and that the pollution of urban soils is strongly related to industrial activities. As anticipated, the urban soils from highly industrialized Beijing had the greatest pollution level, while urban soils from remote Tibet had the lowest pollution concentration, which verifies that the levels of economic development, urbanization, and industrialization have a terrific influence on urban soil contamination. HCH levels in soils from urban areas varied from 50 to 420,200 $\mu g\,kg^{-1}$, while β-HCH was the major component of the four isomers in most cases. The geographical allocation reveals that soils in eastern China were more polluted than those in western China. The ratio of α-HCH/γ-HCH in urban soils varied from 0.05 to 3.5, indicating not only historical applications but also recent uses. The ratio of α-HCH/γ-HCH is usually used to discover the source of HCHs in the environment. The α-HCH/γ-HCH ratio varied between 3 and 7, indicating that HCHs were derived from past applications, while α-HCH/γ-HCH < 3 indicated recent use in the province.

It was demonstrated that forests play a significant role in the accumulation of POPs (including OCPs) in the southeast Tibetan Plateau (TP) due to the 'forest filter effect' [168]. For that reason, a study was performed for the determination of the distribution and transfer of organochlorine pesticides (OCPs) in soils of different forest types (quercous, birch, fir, and spruce dominated forests) in Mt. Shergyla, southeast Tibetan Plateau (TP), under comparable environmental and climatic conditions. HCHs, DDTs, and HCBs in the examined soils varied from below the LOD to 2.25 $\mu g\,kg^{-1}$ dw, from below the LOD to 10.2 $\mu g\,kg^{-1}$ dw, and from below the LOD to 0.95 $\mu g\,kg^{-1}$ dw, respectively. The authors observed that the total concentrations of OCPs in the humus layers were considerably higher than those in the mineral layers in the four forest types. Broadleaved birch forests were demonstrated to exhibit higher DDTs concentrations, while HCHs and HCB were considerably higher in coniferous fir forests. The ratio of p,p′-DDE/p,p′-DDT was below 1 in surface soil, indicating the fresh input of DDT; however, this ratio usually rose in deeper layers, indicating the potential degradation of DDT to DDE. The authors attributed this finding to the current use of DDTs in neighboring countries. Regarding the α-HCH/γ-HCH ratio, it was proven to be between 3 and 7, suggesting that HCHs were mainly derived from technical HCH use [168].

One more study was conducted in Zhejiang province in eastern China for the determination of DDTs in 58 agricultural soils collected in 2006 from the surface horizon (0–20 cm) [169]. The concentration levels of DDTs varied significantly within the samples of the province, ranging from 4.0 to 530 $\mu g\,kg^{-1}$ dw. Among DDTs, p,p′-DDE was the prevalent compound, followed by p,p′-DDD. The low p,p′-DDT/p,p′-DDE ratios and the high o,p′-DDT/p,p′-DDT ratios indicated that there were no recent applications of DDTs; however, the fresh application of DCF, which contains DDT (o,p′-DDT in specific) was identified.

Twenty-six surface soil samples (0–20 cm) from the nature reserve of the Yellow River Delta and specifically from its entrance, the nearby coast, the roadside, and wetland were investigated for the determination of 22 OCPs, including DDTs (o,p′-DDE, o,p′-DDD, o,p′-DDT, p,p′-DDE, p,p′-DDD, and p,p′-DDT), HCHs (α-HCH, β-HCH, γ-HCH, and δ-HCH), heptachlor, heptachlor epoxide, chlordanes (cis- and trans-chlordane), endosulfans (endosulfan α- and β-), HCB, drins (aldrin, dieldrin and endrin), methoxychlor, and MRX [170]. The results indicate a significant variation in the concentration of total OCPs, as they varied from 0.01 to 10.5 $\mu g\,kg^{-1}$; however, DDTs (including o,p′-DDE, o,p′-DDD, o′,p-DDT, p,p′-DDE, p,p′-DDD, and p,p′-DDT) were the predominant compounds with concentrations that varied between 0.17 and 10.46 $\mu g\,kg^{-1}$ in the examined samples. High detection frequencies were also observed for HCHs, and their concentrations ranged between 0.28 and 1.32 $\mu g\,kg^{-1}$. The concentration levels of both HCHs and DDTs suggested that they were extensively used in the past. Heptachlor epoxide was the predominant compound among chlordanes, and its concentration varied between not detected and

3.53 µg kg$^{-1}$, with their detection being in the following order: DDTs > HCHs > chlordanes > endosulfans.

One hundred and fifty-three soils from the surface horizon (0–10 cm) around the Yellow and Bohai Seas of China, including twenty-one cities and five provinces, were collected in September 2013 for the determination of seven OCPs, including α-HCH, β-HCH, γ-HCH, δ-HCH, p,p′-DDE, p,p′- DDD, and p,p′-DDT [171]. The detection frequencies of α-HCH, β-HCH, γ-HCH, δ-HCH, p,p′-DDE, p,p′- DDD, and p,p′-DDT were 100%, 99%, 99%, 100%, 100%, 91%, and 99%, respectively. OCP concentrations in soils fluctuated from 5.89 to 179.96 µg kg$^{-1}$ dw with a mean value of 25.39 µg kg$^{-1}$ dw. Concentrations for individual compounds ranged from 0.94 to 16.89 µg kg$^{-1}$ dw for α-HCH, not detected to 156.12 µg kg$^{-1}$ dw for β-HCH, not detected to 16.17 µg kg$^{-1}$ dw for γ-HCH, 0.9 to 56.24 µg kg$^{-1}$ dw for δ-HCH, 0.43 to 91.23 µg kg$^{-1}$ dw for p,p′-DDE, not detected to 116.2 µg kg$^{-1}$ dw for p,p′-DDD, and not detected to 10.42 µg kg$^{-1}$ dw for p,p′-DDT. The authors determined that the detected OCP residues had a significant relationship with orchard land-use types.

One hundred and fifty-nine soil samples from thirty forested mountain sites across China were collected from May 2012 to March 2013 from O-horizon and A-horizon and analyzed for the determination of 13 OCPs, including α-endosulfan, β-endosulfan, HCB, MRX, o,p′-DDT, pp′-DDT, o,p′-DDD, pp′-DDD, o,p′-DDE, pp′-DDE, cis-chlordane, trans-chlordane, and heptachlor [172]. DDTs (sum of all isomers) demonstrated the greatest concentrations, while p,p′-DDT was the predominant compound in most of the samples, followed by p,p′-DDE in both the O- and A-horizons. Total DDT concentrations in the O- and A-horizons varied between 0.197 and 207 µg kg$^{-1}$. The HCB concentration in the O- and A-horizons ranged from 0.047 to 6.12 µg kg$^{-1}$ and from 0.022 to 0.748 µg kg$^{-1}$, respectively. Heptachlor and cis- and trans-chlordane are the main components of technical chlordane, while trans- and cis-chlordane are the most plentiful isomers in the examined samples. The concentrations of trans- and cis-chlordanes ranged from 0.008 to 0.215 µg kg$^{-1}$ and from 0.017 to 0.333 µg kg$^{-1}$ in the O-horizon and from 0.012 to 0.153 µg kg$^{-1}$ as well as from non-detected to 0.239 µg kg$^{-1}$ in the A-horizon, respectively. A-endosulfan and β-endosulfan concentrations fluctuated from not detected to 0.160 µg kg$^{-1}$ and not-detected to 0.097 µg kg$^{-1}$ in the O-horizon, respectively. Regarding the A-horizon, their concentrations ranged from not detected to 0.039 µg kg$^{-1}$ and from not detected to 0.185 µg kg$^{-1}$ in the A-horizon, respectively. MRX in the O- and A-horizon samples was detected at concentrations ranging between 0.001 and 0.029 µg kg$^{-1}$ and from not detected to 0.019 µg kg$^{-1}$, respectively. Generally, the highest concentrations appeared near agricultural zones or high consumption areas. The chiral compounds were usually non-racemic in the soils and revealed the preferential degradation of o,p′-DDT, trans and cis-chlordane in both O and A-horizons. The authors concluded that recent and historical applications of DDT and historical uses of chlordane and endosulfan may be major sources of OCP accumulation in Chinese forest soils.

Another study was performed in Wuhan, the largest city in central China, regarding the determination of 21 OCPs, including α-HCH, β-HCH, γ-HCH, δ-HCH, p,p′-DDD, p,p′-DDE, p,p′-DDT, o,p′-DDT, o,p′-DDD, o,p′-DDE, methoxychlor, heptachlor, heptachlor epoxide, α-endosulfan, β-endosulfan, aldrin, HCB, cis-chlordane, trans-chlordane, dieldrin, and endrin, in agricultural soils due to a lack of information in this area [173]. In total, 44 soil samples from rice, wheat, corn, bean, cotton, and vegetable soil were gathered from the Wuhan agricultural region in June 2009. DDTs were the predominant OCPs detected in Wuhan, and their concentrations varied from not detected to 1198 µg kg$^{-1}$, accounting for 77.10% of total OCPs. Among DDTs, p,p′-DDE was the predominant compound with residue levels that varied between not detected and 807.82 µg kg$^{-1}$, with the mean concentration of 73.38 µg kg$^{-1}$, followed by p,p′-DDT (mean concentration 52.74 µg kg$^{-1}$), o,p′-DDT (mean concentration 10.64 µg kg$^{-1}$), p,p′-DDD (mean concentration 9.26 µg kg$^{-1}$), o,p′-DDD (mean concentration 2.94 µg kg$^{-1}$), and o,p′-DDE (mean concentration 2.62 µg kg$^{-1}$) [173]. The p,p′-DDT/(p,p′-DDE + p,p′-DDD) ratio varied

from 0 to 46.72; however, in 70.5% of the examined samples, it was below 1, indicating historical uses in most of the examined samples, but recent applications could not be excluded. The o,p′-DDT/p,p′-DDT ratio was used to discern whether DDT pollution was due to the usage of technical DDT or DCF, and the authors concluded that recent DDT was primarily introduced by technical DDT. On the other hand, HCHs (sum of α-, β-, γ-, and δ-HCH) accounted for 7.83% of OCPs, and their concentration varied from not detected to 100.58 μg kg$^{-1}$ with the mean concentration of 15.39 μg kg$^{-1}$. The mean concentrations of β-HCH, δ-HCH, α-HCH, and γ-HCH were found to be 6.41, 5.14, 2.64, and 1.20 μg kg$^{-1}$, respectively. It is obvious that among HCHs, β-HCH was found to be the predominant isomer, indicating a lack of new HCH applications in most cases. Moreover, the α-HCH/γ-HCH ratio ranged from 0 to 2.83, indicating historical uses of LND. Regarding chlordanes (cis and trans isomers), their residue levels varied between not detected and 7.17 μg kg$^{-1}$, with a mean concentration of 0.48 μg kg$^{-1}$. The residue levels of chlordanes (sum of cis and trans isomers) ranged from not detected to 7.17 μg kg$^{-1}$, with a mean concentration of 0.48 μg kg$^{-1}$, with cis-chlordane being the predominant compound. The concentration of heptachlors (heptachlor and heptachlor epoxide) ranged from not detected to 29.04 μg kg$^{-1}$ with a mean of 5.55 μg kg$^{-1}$, with heptachlor being the predominant compound with a concentration that varied from not detected to 28.71 μg kg$^{-1}$, and heptachlor epoxide being in the range of not detected to 3.65 μg kg$^{-1}$. Although heptachlor is metabolized in soils to heptachlor epoxide, the concentration of heptachlor was found to be higher than that of heptachlor epoxide, indicating fresh applications in most agricultural soils of Wuhan. HCB was identified in 86% of the examined soils, with a concentration that varied from not-detected to 17.77 μg kg$^{-1}$ with a mean concentration of 3.01 μg kg$^{-1}$. Since HCB exists as an impurity in some pesticides, no conclusion could be reached about its use. Endosulfan (α- and β-isomers) is still being used on cotton and other crops in China and thus was identified in the examined soils with a concentration that varied between not detected and 23.04 μg kg$^{-1}$, with a mean concentration of 1.28 μg kg$^{-1}$, while both α- and β-isomers were observed in 9% of soil samples. Metoxychlor was identified in 14% of soil samples with a concentration that varied from not detected to 169.03 μg kg$^{-1}$ and a mean concentration of 10.47 μg kg$^{-1}$. The detection frequencies for aldrin, dieldrin, and endrin were 73%, 9%, and 5%, respectively, and their concentrations ranged from not detected to 21.56 μg kg$^{-1}$, not detected to 16.69, and not detected to 6.12 μg kg$^{-1}$, respectively. The authors attributed their existence to atmospheric depositions [173].

Allocations, sources, environmental risks, as well as environmental behaviors of 20 OCPs, including HCB, aldrin, HCHs (α-, β-, γ-, and δ-HCH), DDTs (p,p′-DDT, p,p′-DDE, p,p′-DDD, o,p′-DDT, o,p′-DDE, and o,p′-DDD), heptachlors (heptachlor, heptachlor epoxide A, and heptachlor epoxide B), chlordanes (cis-chlordane, trans-chlordane, and oxy-chlordane), and endosulfans (endosulfan-α and endosulfan-β), in riparian soils and sediments of the middle reach of the Huaihe River, a traditional agricultural area of China, were examined. The Huaihe River Basin is a conventional farming area, as well as an essential grain production center in China [174]. Riparian topsoil samples (0–5 cm) were gathered from 28 sampling sites. All the target OCPs were identified in riparian soils, except heptachlor epoxide A, while their total concentration ranged from 1.8 to 63 μg kg$^{-1}$, and the average concentration was 19 μg kg$^{-1}$. HCHs were the predominant compounds, while α-HCH was the most prevalent HCH isomer in riparian soils, and its concentration ranged from 1.2 to 31 μg kg$^{-1}$ with an average concentration of 13 μg kg$^{-1}$, accounting for 69% of the total OCPs. Consequently, the isomeric ratio of α-HCH/γ-HCH was used to distinguish the sources of HCH, and this ratio was in the range between 1.7 and 14 with an average value of 5.9, indicating that both technical HCH and LND were used in the area. The DDT average concentration was found to be 0.4 μg kg$^{-1}$, accounting for 2% of the total OCPs in soils, and p,p′-DDE was the main isomer, followed by p,p′-DDD and p,p′-DDT. The ratio of (p, p′-DDE + p,p′-DDD)/p,p′-DDT was used to distinguish between fresh DDT application and historical uses. In this monitoring study, the abovementioned ratio was below 1 for only one sample, while the mean ratio for the rest of the samples was 4.6,

indicating mainly historical uses. Heptachlor and heptachlor epoxide B were identified in most of the soils, and their total concentration was found to be 3.6 $\mu$g kg$^{-1}$, accounting for 19% of the total OCPs. Chlordanes were also identified in soils, and their total concentration was found to be 0.83 $\mu$g kg$^{-1}$, and oxy-chlordane was the main compound, accounting for 96% of total chlordanes. The authors concluded that chlordanes were derived mainly from weathered chlordane. Aldrin was detected in 96% of soils, with the mean concentration of 0.53 $\mu$g kg$^{-1}$.

Fifty-five soil samples were gathered during May 2015 from six stations along the agricultural (twenty-one samples), backwaters (seven samples), and coastal (twenty-seven samples) transects of the southwest coast of India for the determination of 17 OCPs including $\alpha$-HCH, $\beta$-HCH, $\gamma$-HCH, $\delta$-HCH, p,p$'$-DDT, p,p$'$-DDE, p,p$'$-DDD, $\alpha$-endosulfan, $\beta$-endosulfan, endosulfan sulfate, aldrin, dieldrin, endrin, endrin ketone, methoxychlor, heptachlor, and heptachlor epoxide. HCH, dieldrin, endrin, and endrin ketone were the most frequently identified compounds, with detection frequencies of more than 95% [175]. Generally, a declining trend in DDT, HCH, and endosulfan was noted as the study moved down south to the Indian Ocean. It should be stated that $\alpha$-endosulfan was not identified in any of the samples; however, $\beta$-endosulfan was identified in some samples, and the omnipresence of endosulfan sulfate was noted. The total OCP level was at the maximum in the agricultural transect (48%), followed by the coastal (40%) and backwater transect (11%). For DDTs, endosulfans, and HCHs, the highest concentrations were observed in the agricultural transect, possibly due to historical and/or continuing usage of OCPs. HCHs (sum of $\alpha$-HCH, $\beta$-HCH, $\gamma$-HCH, and $\delta$-HCH) contributed to 15% of the total OCPs, and their concentration ranged from not detected to 123 $\mu$g kg$^{-1}$, with an average of 30 $\mu$g kg$^{-1}$. $\alpha$-HCH was the predominant compound, followed by $\delta$-HCH, $\gamma$-HCH, and $\beta$-HCH [175]. Based on the $\alpha$-HCH/ $\gamma$-HCH ratio, the authors concluded that its major source was from technical HCH. The $\beta$-HCH/($\alpha$-HCH + $\gamma$-HCH) ratio was used to reach the conclusion of ongoing HCH usage. DDTs (sum of p,p$'$-DDT, p,p$'$-DDE, and p,p$'$-DDD) contributed to around 5% of the total OCPs, and their concentration fluctuated between not detected to 148 $\mu$g kg$^{-1}$, while the highest DDT concentration was observed in the agricultural transect. The major DDT was p,p$'$- DDE, contributing almost 60% of the total DDT concentration, followed by p,p$'$-DDE (30%) and p,p$'$-DDD (10%). The total DDT concentration followed the order: agricultural > coastal > backwater transects. The prevalence of p,p$'$-DDT was noted in a few sites in agricultural and coastal transects, possibly due to its usage in vector control programs. Endosulfan (sum of $\alpha$-endosulfan, $\beta$-endosulfan, and endosulfan sulfate) residues varied from not detected to 21 $\mu$g kg$^{-1}$, with the mean concentration of 2 $\mu$g kg$^{-1}$. The detection frequency of $\beta$-endosulfan and endosulfan sulfate was 13% and 35%, respectively. The contamination of endosulfan can be ascribed to historical applications in cashew plantations in the southwest coast. Endrins (sum of endrin and endrin ketone) contributed more than 50% of the total OCPs, and their concentrations ranged from not detected to 491 $\mu$g kg$^{-1}$, and the mean concentration was 104 $\mu$g kg$^{-1}$. Higher endrin ketone levels in the surface soil could be attributed to historical applications of endrin. Heptachlors (sum of heptachlor and heptachlor epoxide) contributed totally to 16% of the OCPs, while their residues fluctuated between not detected and 197 $\mu$g kg$^{-1}$, with an average concentration of 33 $\mu$g kg$^{-1}$. Heptachlor epoxide was the most frequently detected compound (>90% detection frequency) and contributed nearly 70% to heptachlors' concentration. The authors proposed that the higher detection of heptachlor epoxide could be associated with its strong adsorption in the soil and its resistance to biodegradation. Drins (aldrin and dieldrin) along with methoxychlor contributed less than 10% of the total OCPs concentration.

One recent monitoring study was performed in Cardamom Hill Reserve, located in the southwestern Ghats of the Idukki District of Kerala, India, for the determination of 17 OCPs, including p, p$'$-DDE, p,p$'$-DDD, p,p$'$-DDT, $\alpha$-HCH, $\gamma$-HCH, $\beta$-HCH, $\delta$-HCH, aldrin, dieldrin, endrin, endrin aldehyde, $\alpha$-endosulfan, $\beta$-endosulfan, endosulfan sulphate, heptachlor, heptachlor epoxide, and methoxychlor [4]. Twenty-two samples were randomly collected between May 2017 and 2018 from the surface horizon (0–15 cm depth) from

cultivated cardamom fields. The study results indicated that HCHs, DDTs, endrin, dieldrin, and endosulfan were identified in practically all soil samples, while the most frequently detected OCP was β-endosulfan, followed by endrin, p,p′-DDD, and δ-HCH. The integral concentration of OCPs in the examined soil samples varied between 6.35 μg kg$^{-1}$ and 118.29 μg kg$^{-1}$. Regarding DDT and its derivatives, p,p′-DDD was the most predominant compound ranging from below the LOQ to 43.62 μg kg$^{-1}$. The authors attributed the existence of DDTs to the extensive historical application of DDT as the concentration of its metabolites was higher than the parent compound ((DDE + DDD)/DDT < 1). The allocation of p,p′-DDD in the study area suggests the active anaerobic degradation of DDT in soil [176]. Approximately 33% of the studied samples contained endosulfan isomers (α and β); among them, β-endosulfan was the predominant compound and its determined concentrations ranged between below the LOQ and 49.92 μg kg$^{-1}$, while α-endosulfan and endosulfan sulfate concentrations ranged from below the LOQ to 8.499 μg kg$^{-1}$ and below the LOQ to 5.2024 μg kg$^{-1}$, respectively. α-Endosulfan was detected in 4.5% of the examined samples, and the authors ascribed the existence of α-endosulfan to historical uses. The authors attributed the existence of endosulfan sulfate to the conversion of α- and β-endosulfan into endosulfan sulfate, which is more persistent in soil than the parent compounds. The cyclodiene drin-related compounds (aldrin, dieldrin, and endrin) were detected in approximately 30% of the examined samples, while endrin was the main compound, and its concentration ranged from below the LOQ and 35.59 μg kg$^{-1}$. Endrin aldehyde was also observed in the soil samples, ranging from below the LOQ to 32.144 μg kg$^{-1}$. The authors determined that endrin aldehyde exists in soils due to photochemical reactions and the biodegradation of endrin, as endrin is converted to endrin ketone and endrin aldehyde [177]. Aldrin and dieldrin were also detected in soil samples, indicating that agriculture is the core source of these compounds. HCHs were additionally found in 15.39% of the examined samples. The isomers that were identified were α-HCH, γ-HCH, and δ-HCH at 7.1%, 0.96, and 7.33% of the sum of OCPs, respectively. The authors concluded that the contamination of the examined area is due to technical HCH because of the lower concentration of γ-HCH compared to α-HCH. The prevalence of δ-HCH isomer in the study area indicated historical applications and not current use. Aldrin, p,p′-DDE, β-HCH, heptachlor, heptachlor epoxide, and methoxychlor were not detected at concentrations above the detection limits in all samples.

Eighty-one surface soil samples (0–20 cm depth) of urban, suburban, and rural transects from the seven most important Indian cities, namely, New Delhi and Agra in the north, Kolkata in the east, Mumbai and Goa in the west, and Chennai and Bangalore in the southern part of India, were gathered and analyzed for the determination of OCPs, including o,p′-DDT, p,p′-DDE, p,p′-DDT, α-, β-, γ-, and δ-HCH, HCB, chlordanes, and endosulfans [178]. HCH concentrations were found to be between 0.01 and 60 μg kg$^{-1}$ dw, with the greatest concentration presented in the rural site of Bangalore, followed by a rural site of Goa. β-HCH was the predominant compound followed by γ-HCH, while the β-HCH/(α-HCH + γ-HCH) ratio was higher than 1 in 65% of the examined samples, which indicated historical uses of technical HCH. However, the use of LND could not be excluded, as in the rest of the cases, as this ratio fluctuated between 0.1 and 0.5. The DDT concentration varied between 0.4 and 124 μg kg$^{-1}$ dw. Since 60% of DDTs consist of o,p′-DDT and p,p′-DDT, the authors concluded that the recent use of technical DDT had occurred. However, there were cases where DDTs consisted of p,p′-DDE and p,p′-DDD, indicating historical DDT applications. The dominating p,p′-DDE compound indicates the degradation of p,p′-DDT under the tropical climate of the country. Chlordane residue levels in the examined soils varied from 0.01 to 30 μg kg$^{-1}$ dw, while the highest concentration was detected in the rural site of Goa. Trans and cis-chlordanes contributed about 35% and 25% of total chlordanes, respectively. Endosulfans ranged from 0.01 and 237 μg kg$^{-1}$ dw, with the highest concentration in a rural site of Goa followed by Bangalore. Endosulfan sulfate was the predominant endosulfan compound in Indian soils. In 72% of the examined soil samples, the ratio of α-endosulfan/β-endosulfan was 2.33, indicating the recent usage

of endosulfan. HCB was omnipresent in all the soil samples gathered from all seven of the major Indian cities. HCB levels could be ascribed to its usage for industrial and agricultural purposes, together with emissions from the partial incineration of waste, coal, fuel, and biomass. Total HCB concentration in Indian soil varied between 0.01 and 8 µg kg$^{-1}$ dw, and the highest concentration was detected in Bangalore, followed by Goa. High levels of HCB were also found in soil samples, especially from the site of New Delhi near to a large-scale manufacturing unit, while in the rural site of Bangalore and Goa, it may be due to the inadequate incineration of anthropogenic waste [178].

A study was performed in March 2014 in an agricultural area in Kumluca, a region of Antalya on the Mediterranean coast of Turkey, for the determination of 22 OCPs including: α-HCH, β-HCH, γ-HCH, δ-HCH, heptachlor, heptachlor epoxide, α-chlordane, γ-chlordane, trans-nonachlor, cis-nonachlor, endosulfan isomers (α-, β-endosulfan, and endosulfan sulfate), p,p′-DDE, p,p′-DDD, p,p′-DDT, endrin, aldrin, dieldrin, endrin aldehyde, endrin ketone, and methoxychlor, as well as PCBs in air and soil [179]. The climatic feature of that area is a usual Mediterranean climate, which is characterized by hot and humid summer and rainy and wet winters. Thirty-four soil samples were gathered from three different soil depths: 0–15 cm, 15–30 cm, and 30–45 cm. The total concentration of OCPs in soil samples varied between not detected and 28.1 µg kg$^{-1}$ dw. Eighteen OCPs were detected in the studied soil samples, while their detection frequencies ranged from 2.94% to 91.2%. DDTs were the most frequently detected OCPs; among them, p,p′-DDE was the most prevalent compound. Based on the DDT/DDE ratio, the authors attributed the existence of p,p′-DDE to historical past uses in most (27 cases) of the examined soils. However, there were three cases with DDT/DDE ratios of 1.33, 1.14, and 2.27, indicating recent uses of DDT. Endosulfans distinguished in concentrations ranged from not detected to 13.8 µg kg$^{-1}$ dw, while endosulfan sulfate was the most predominant compound in most sampling sites. The dominance of endosulfan sulfate in most of the soil samples presents a high degree of degradation of parent isomers, which might stem from older usage. The ratio of α-/β-endosulfan in all studied soil samples varied from 0.327 to 1.20. The lower α-/β-endosulfan ratios might be an indication of fresh endosulfan applications. The levels of HCHs in the examined soil samples ranged between not detected to 0.041 µg kg$^{-1}$ dw, whilst γ-HCH was the principal compound. The isomer β-HCH was only detected in one sample. The ratios of α-HCH/γ-HCH of soil samples in this study ranged between 0.250 and 1.57; thus, the authors concluded that in the examined sites, normally, LND was used in the past. The total concentrations of chlordanes (α- and γ-chlordane, trans- and cis-nonachlor) in soil ranged from not detected to 0.121 µg kg$^{-1}$ dw [179]. Their relative contribution was trans-nonachlor, γ-chlordane, α-chlordane, and cis-nonachlor. The ratio of γ-chlordane/α-chlordane ranged from 0.500 to 3.75; thus, the author attributed the existence of chlordanes to historical uses of technical chlordane. The total concentration of heptachlor and heptachlor epoxide varied from not detected to 0.095 µg kg$^{-1}$ dw, indicating primarily past uses with some fresh contributions of technical chlordane. With regard to the drin derivatives, aldrin and endrin ketone were not detected in any of the samples, while dieldrin and endrin aldehyde were rarely detected. Endrin was distinguished in 38.2% of soil samples, while the total drin concentration ranged from not detected to 0.604 µg kg$^{-1}$ dw. Due to the absence of endrin's degradation products, the authors attributed the existence of endrin to recent applications [179].

There was a shortage of research on OCP residues in the soils of Hong Kong until 2000. However, 66 soil samples were gathered during December 2000 from the New Territories, Kowloon, Hong Kong Island, and Lantau Island for the determination of 16 OCPs including DDTs, HCHs, HCB, heptachlor, aldrin, endrin, dieldrin, and α- and β-endosulfan [10]. Forty-six soil samples were collected form the surface horizon (0–10 cm) encompassing five different land use patterns such as grassland, woodland, wetland, and arable as well as reclamation land, while the rest of the twenty agricultural soil samples were gathered from four sampling sites from different soil horizons and from nine soil profiles to demonstrate the depth distributions of OCPs. The analysis results demonstrate

that heptachlor, aldrin, dieldrin, and β-endosulfan were not detected in any of the examined samples. HCHs (100%) and DDTs (93.47%) were the most frequently detected OCPs. Endrin, HCB, and α-endosulfan were detected in 32.6%, 11%, and 2.2% of the examined samples, respectively [10]. Endrin was detected in the examined soils at concentrations that ranged from 0.007 to 0.093 μg kg$^{-1}$ dw. The HCB detection concentration varied from 0.007 to 0.31 μg kg$^{-1}$ dw. but was not detected above the LOQ in arable soils. The authors ascribed the presence of HCB in soils to waste from chlorine-related industries and not the use of HCB as a pesticide. The average concentrations of DDTs and HCHs in the examined soils were 0.52 μg kg$^{-1}$ dw and 6.19 μg kg$^{-1}$ dw accordingly. The total concentrations of DDTs in arable soils were substantially greater than those in woodland and grassland soils, indicating that DDT was primarily used for agricultural activities in Hong Kong. The authors observed that the total concentration of HCHs in all examined soils presented insignificant variations; therefore, they attributed this finding to the comparatively higher vapor pressure of HCH, indicating that HCH is much easier to volatilize than DDT from soil to atmosphere and return to soil through dry or wet deposition after atmospheric transportation, leading to the further homogenous distribution of HCH in soils [162]. DDT (o,p′ and p,p′), p,p′-DDE, and p,p′-DDD were the constituents of DDTs, with p,p′-DDE being its predominant metabolite, accounting for 78.8% of the total DDTs. Therefore, the authors concluded that DDT's main degradation route is to DDE. The ratio of DDT/ (DDE + DDD) implied the existence of aged DDT in most soils of Hong Kong; however, fresh applications were assessed in very rare cases. HCH (α-, β-, and γ-HCH) was detected in soil, with β-HCH being the predominant compound accounting for 95.8–100% of the total HCH concentration. Additionally, the ratio of α-HCH to γ-HCH in the examined soil was like the ratio of technical HCH, which meant it could be assumed that historical uses of technical HCH was its main source; however, the historical usage of LND (γ-HCH) could not be excluded. Regarding the depth distribution of HCHs and DDTs, the authors observed that for HCH, the concentration increased with soil depth, while the DDT concentration acted oppositely. The authors' findings are in line with other research, indicating higher vertical mobility in soil profiles for HCH than for DDTs due to differences in their solubility. The authors mentioned that p,p′-DDE and α-HCH and β-HCH were the main constituents of DDT and HCH, respectively, in the whole soil profile, indicating that the DDT and HCH of the various soil levels were completely undergoing the aged phase [10].

Another study was conducted in three provinces in Pakistan for the determination of OCPs, including HCHss, heptachlors, dieldrin, endrin, and DDTs, in soil and water from the vicinity of selected obsolete pesticide stores in Pakistan [180]. Between 1960 and 1970, a large amount of OCPs was imported in Pakistan for malaria suppression, locust management, and the control of crop pest infestation during the green revolution in the country. Additionally, DDT and HCH were manufactured locally. Soil samples were collected from obsolete pesticide storehouses and courtyards. Thirty-one soil samples were generally gathered from extremely contaminated locations from the vicinity of obsolete pesticide storehouses and courtyards and specifically from Northwest Frontier Province (5 samples), from Punjab (14 samples), and from Sindh (12 samples). DDTs, LND, and heptachlor were the most frequently identified compounds in all examined soil samples, while DDT and its metabolites were detected in almost 100% of soil samples. p,p′-DDT was the predominant DDT compound, followed by o,p′-DDT, p,p′-DDE, and p,p′-DDD. Dieldrin and endrin were distinguished in 29% and 16% soil samples, with an average concentration of 3000 μg kg$^{-1}$ and 5000 μg kg$^{-1}$, respectively. The total OCP concentration ranged from: 247,000 to 9,157,000 μg kg$^{-1}$ in the soils from Northwest Frontier Province, 214,000–10,892,000 μg kg$^{-1}$ in soils from Punjab province, and 86,000–113,800 μg kg$^{-1}$ in soils from Sindh province. The authors observed that obsolete pesticide stores were in bad conditions and posed a threat to human health and the environment; thus, further research into the decontamination of these sites is required [180].

Another study was conducted at an obsolete pesticide dumping ground and the associated areas in Hyderabad City of Pakistan for the determination of 13 OCPs, including DDTs (p,p′-DDE, o,p′-DDE, p,p′-DDD, o,p′-DDD, p,p′-DDT, and o,p′-DDT), chlordanes (cis and trans chlordane), HCB, heptachlor, and HCHs (α-HCH, β-HCH, and γ-HCH), in surface soils [181]. Twenty soil samples from the surface horizon (0–5 cm) were collected from different land use types including pesticide burial ground (seven samples), industrial (four samples), residential (four samples), and background soils (five samples). With regard to pesticide burial ground soils, DDTs were the most frequently detected chemicals and in higher residue levels (77–21,200 μg kg$^{-1}$), while p,p′-DDE (40.2%) and p,p′-DDT (29.5%) were the predominant compounds, followed by HCHs (43–4090 μg kg$^{-1}$), with α-HCH being the dominant component, then chlordanes (0.5–577 μg kg$^{-1}$), HCB (1.3–100 μg kg$^{-1}$), and heptachlor (0.1–28 μg kg$^{-1}$). The most frequently detected OCPs were p,p′-DDE, p,p′-DDEm, and α-HCH with the mean residue levels for p,p′-DDE, p,p′-DDT, and α-HCH being 2212 μg kg$^{-1}$, 615 μg kg$^{-1}$, and 1960 μg kg$^{-1}$, respectively. Generally, the distribution of OCPs revealed significant variations in all types of sampling sites. The authors concluded that land use plays an essential role in controlling the distribution pattern of OCPs in soil, as various land use types have different physiochemical properties and OCP levels also vary in agreement with it [181].

Twenty-seven soil samples were gathered and analyzed for the determination of OCPs, including HCHs (α-HCH, β-HCH, γ-HCH, and δ-HCH), DDTs (o,p′, p,p′-DDE, o,p′, p,p′-DDD, and o,p′, p,p′-DDT), endosulfans (α-endosulfan, β-endosulfan, and endosulfan sulfate), HCB, heptachlor, and heptachlor epoxide, from nine dumping sites in Pakistan within 200–500 m or 1–2 km distances from waste dumping sites [182]. The total concentration of OCPs in surface soil of waste dumping sites of Pakistan fluctuated between 4.2 and 30.44 μg kg$^{-1}$, and the mean concentration was 13.79 μg kg$^{-1}$, following the order: DDTs > HCHs > endosulfans > HCB > heptachlors. The average concentration of DDTs, HCHs, endosulfans, HCB, and heptachlors were 6.49 μg kg$^{-1}$, 3.5 μg kg$^{-1}$, 2.65 μg kg$^{-1}$, 1.12 μg kg$^{-1}$, and 0.93 μg kg$^{-1}$, respectively. The total DDT concentration varied between 0.16 and 25.66 μg kg$^{-1}$, while the occurrence of DDTs was in the order: p,p-DDD > o,p′-DDT > p,p′-DDT > o,p′-DDD and p,p′-DDE > o,p′-DDE. In most of the sampling sites, p,p′-DDD was identified with a mean value of 18.8 μg kg$^{-1}$. The detection frequency of DDT isomers was 100%, except for p, p′-DDE, which had a detection frequency of 35.8% [182]. The authors attributed the existence of the higher concentration of DDT metabolites in the examined soils to the subtropical weather in Pakistan that increased the rate of the transformation of the parent DDT into its metabolites. For differentiating the contribution of DCF from technical DDT, the o, p′-DDT/ p, p′-DDT ratio was employed, indicating the use of DCF in most cases, the use of both DCF and DDT in few cases, and the use of technical DDT in some other cases. The p, p′-DDT/(p,p′-DDE + p,p′-DDD) ratio suggested both recent and historical use. The DDE/DDD ratio was used as an indicator for assessing the degradation pathway of DDT. In the examined soils, both anaerobic and aerobic degradation was identified. The total concentration of HCHs (sum of isomers α-HCH + β-HCH + γ–HCH + δ-HCH) varied from 0.81 to 14.94 μg kg$^{-1}$, while the order of HCHs isomers was as follows: α-HCH > β-HCH > γ-HCH > δ-HCH. The detection frequency of HCHs was 100%, except for δ-HCH, which had a detection frequency of 62%; thus, its contribution to the total HCH pollution was lower than the other isomers. The α-HCH/γ-HCH and β-HCH/ (α-HCH + γ-HCH) ratios were used to distinguish the contamination source and input history. Based on the abovementioned ratios, both LND and technical HCH were used, while historical uses and recent applications of HCH were identified. The total endosulfan concentration fluctuated between 0.10 and 9.62 μg kg$^{-1}$, while the occurrence of its isomers followed the order: endosulfan sulfate > α-endosulfan > β-endosulfan. It was noticed that β-endosulfan had a 100% detection frequency, whereas endosulfan sulfate had the lowest detection frequency of 30.8% among endosulfan isomers. The α-endosulfan/β-endosulfan ratio was used for assessing historical or fresh applications. Both cases were identified. The heptachlor concentration varied between 0.11 and

3.69 µg kg$^{-1}$. The heptachlor epoxide/heptachlor ratio was used to estimate the age of heptachlor; thus, recent application was identified [182].

HCB was identified in the range of 0.01 to 5.87 µg kg$^{-1}$, and its existence was ascribed as a by-product during manufacturing processes or as an impurity in various chlorinated pesticides, including LND.

A study was performed for the determination of OCPs, including HCHs (α-HCH, β-HCH, γ-HCH, and δ-HCH), DDTs (o,p'- and p,p'-DDE, -DDD, and -DDT), chlordanes (cis and trans-chlordane), HCB, heptachlor, and β-endosulfan, in the Indus River of Pakistan, which plays a critical role in the agricultural sector of the country and additionally has unique ecological significance [183]. Soil and air samples were taken from the selected sites; specifically, 38 soil samples (0–15 cm) were gathered. The total OCP residue concentration fluctuated between 0.70 and 13.47 µg kg$^{-1}$, while p,p'-DDE had the highest concentration of 0.71 µg kg$^{-1}$. The tendency of OCPs was in the following order: DDTs > HCHs > chlordanes > HCB > heptachlo r> β-endosulfan. Amongst DDTs, p,p'-DDE was demonstrated to have the highest concentration (not detected to 4.76 µg kg$^{-1}$), which suggested that DDT in soil was exposed to microbial degradation and transformation takes place into its more stable and toxic metabolite, p,p'-DDE. HCHs isomers were detected in low concentrations, and β-HCH and δ-HCH were the predominant components. Chlordanes fluctuated between not detected to 0.33 µg kg$^{-1}$ for cis-chlordane and between not detected 0.96 µg kg$^{-1}$ for trans-chlordane. [183].

There is limited information about the contamination of agricultural soil by OCPs in Iran. One study was conducted between March and April 2016 in agricultural areas in southern Iran and specifically from the agricultural areas of the Dalaki and Shabankare plains of Iran. Twenty-eight soil samples were gathered form the surface horizon (0–10 cm) for the determination of OCPs, including HCH, heptachlor, DDT, chlordane, and their isomers [184]. Residues of DDTs, chlordanes, and HCHs were identified in all soil samples from both plains, while their mean concentration was determined in the following order: DDTs > chlordanes > HCH. In the Dalaki plain, the mean concentrations of the identified compounds were as follows: 0.411 µg kg$^{-1}$ dw for HCHs (α-, β-, γ-, and δ-HCH), 4.37 µg kg$^{-1}$ dw for DDTs (o,p'-DDE, o,p'-DDD, o,p'-DDT, p,p'-DDE, p,p'-DDD, and p,p'-DDT), and 2.04 µg kg$^{-1}$ dw for chlordanes (trans-chlordane, cis-chlordane, heptachlor exo-epoxide, and heptachlor). In the Shabankare plain, the mean value of HCHs, DDTs, and chlordanes was measured to be 1.38 µg kg$^{-1}$ dw, 11.99 µg kg$^{-1}$ dw, and 1.62 µg kg$^{-1}$ dw, respectively. The concentration tendency of OCP residues in both plains was as follows: DDT > CHL > HCH. The ratio of p, p'-DDT/(p,p'-DDE + p,p'-DDD) can be an indicator of the use of DDT. Based on the results, this ratio in Dalaki for 64.3% of the samples was less than 1 and was greater than 1 for 35.7% of the samples, indicating the historical use of DDT in most of the examined soils and recent use in some of them. In the Shabankare plain, 14.3% of the soil samples revealed the recent use of DDT, while 85.7% of the measured samples showed historical use. Taking into account the ratio of o,p'-DDT/p,p'-DDT, the authors concluded that in both plains, 7.1% of the soil samples demonstrated the use of DDT, and 57.1% of the evaluated samples indicated that DDT was used as DCF [184]. Additionally, the ratio of p, p'-DDE/p, p'-DDD can be utilized to ascertain the relative aerobic or anaerobic conditions governing the soil environment. The authors concluded that in Dalaki soil samples, 78.6% and 21.4% of the soil samples had aerobic and anaerobic conditions, respectively; however, in Shabankare, in all soil samples, the aerobic conditions were dominant. With regard to chlordanes, it is known that trans-chlordane decomposes faster than cis-chlordane in the environment; thus, cis-chlordane/trans-chlordane is used for the determination of the recent or historical use of this OCP. In the Shabankare and Dalaki plains, this ratio was greater than 1 in 64.3% and 21.4% of the soil, respectively, indicating historical use in most cases [184].

Ten soil samples from the surface horizon (0–20 cm) were collected twice during September 2017 and February 2018 from fields in paddy plantations situated in five locations of Machang, Kelantan of Peninsular Malaysia, for the determination of ten organochlo-

rine pesticides, including HCHs (α-HCH, β-HCH, γ-HCH, and δ-HCH), p,p′-DDT, p,p′-DDE, p,p′-DDD, α-endosulfan, β-endosulfan, and endosulfan sulfate [185]. The presented results indicated that all HCHs isomers were identified, along with p,p′-DDT and endosulfan sulfate. HCHs concentrations varied from below the LOD to 7340 μg kg$^{-1}$, while α-HCH was the predominant compound. Regarding p,p′-DDT, its concentration varied from 90 to 5240 μg kg$^{-1}$ and endosulfan sulfate from below the LOD to 30 μg kg$^{-1}$.

The central Asian Republic of Tajikistan has been an area of widespread historical agricultural pesticide use as well as large-scale burials of banned OCPs [186]. Soil samples from the surface horizon (0–10 cm) and from four rural areas of Tajikistan were gathered during a four-year study from pesticide burial sites and family farms for the determination of OCPs including DDTs (DDT, DDD, and DDE), LND isomers (α-HCH, β-HCH, γ-HCH, and δ-HCH), endosulfan isomers (α-, β-endosulfan, and endosulfan sulfate), other cyclodienes (aldrin, α- and γ-chlordanes, dieldrin, endrin, endrin aldehyde and ketone, heptachlor, and heptachlor epoxide), and methoxychlor. The sampling sites were selected to represent a variety of pesticide disposal histories and to allow for the consideration of local pesticide pollution in Tajikistan. DDT was regularly the highest measured pesticide in all four sampling areas, along with HCH isomers and β-endosulfan. Concentrations of DDD and DDE were substantially lower than the levels of DDT at each site. Heptachlor, heptachlor epoxide, methoxychlor, endrin ketone, aldrin, endrin, and endosulfan sulfate were not detected at levels above the LOD.

*4.4. American Countries*

A study to assess the contamination levels of the banned OCPs in the physical environment of Costa Rica, a country in Central America, was performed across the whole country in 2004, with sampling of air and soil samples at 23 stations. The soil-sampling sites (0–25 cm) were in protected areas, such as National Parks, Biological Reserves, and research stations, where OCPs were not used in the past and additionally reflecting not only the diverse topography (sites on the Caribbean and Pacific coast and locations with 3400 m in elevation) but also climate and soil properties. In the studied soil samples, the major DDT compounds were p,p′-DDE and p,p′-DDD with concentrations that ranged from below the LOQ up to 1 μg kg$^{-1}$, which could be attributed to non-agricultural background soils in protected areas [187]. Dieldrin had the highest concentration reaching up to 2.0 μg kg$^{-1}$; however, it was detected in less than 50% of the samples. The residue levels of HCHs were normally quite low; however, γ-HCH was comparatively abundant among the other HCHs. The authors attributed the higher γ-HCH concentrations noticed in some sampling sites to the recent use of LND in the country. DDE and DDD were only identified in 5 of the 20 samples; however, the high DDT levels clearly indicate hotspots of DDTs which were used for the elimination of malaria. Heptachlor epoxide was only detected at three sampling sites and may be related to the differential ability of soil to transform heptachlor [187].

Amongst the Latin American countries, Mexico was a major consumer of OCPs for both sanitary and agricultural purposes. For that reason, a study was performed in Mexico for the examination of the spatial distribution of OCPs including HCHs (α-HCH, β-HCH, γ-HCH, and δ-HCH), cis- and trans-chlordane, trans-nonachlor, aldrin, dieldrin, heptachlor and heptachlor epoxide, α-endosulfan, β-endosulfan and endosulfan sulfate, DDTs (p,p′-DDE, o,p′-DDE, p,p′-DDD, o,p′-DDD, p,p′-DDT, and o,p′-DDT), and toxaphenes in rural (with no agricultural activities and away from urban sites), urban, and agricultural soils. Eighteen soil samples from the surface horizon (0–5 cm) were collected from eighteen sites across nine states of Mexico during 2005 [188]. The most frequently detected OCPs were DDTs (100%), followed by toxaphenes (97%), endosulfans and chlordanes (93%), HCHs (55%), drins (21%), and heptachlors (3%). Aldrin, β-HCH, and δ-HCH were not detected in any of the samples. The DDT residue levels varied from below the LOD to 360 μg kg$^{-1}$, while their highest concentration identified in urban soils, followed by agricultural and rural soils. It was clarified that the highest residue DDTs levels found in urban areas was

due to samples taken from the endemic malaria's regions, where the highest amount of DDT was consumed during 1989 to 1999. Rural soils had a maximum DDTs residue level of 1.7 μg kg$^{-1}$, while, in most of them, the concentration was below 0.04 μg kg$^{-1}$. The most detectable DDT isomer was the p,p′-DDE, indicating historical uses; however, in 5 of the examined soils, p,p′-DDT concentration was higher than p,p′-DDE, indicating either recent DDT use or slower degradation rates in these soils. With regard to the DDTs, it was found that 59% of the samples contained p,p′-DDE, 48% o,p′-DDT, and 38% o,p′-DDE. Toxaphenes were considered as the sum of hepta-, octa-, and non-achlorobarnates; thus, toxaphene residue levels varied from below the LOD to 334 μg kg$^{-1}$ in all sampling points. Higher concentrations were found in agricultural soils, followed by urban soils, while rural soils contained the lowest residue levels [188]. Endosulfan residues (α-endosulfan, β-endosulfan, and endosulfan sulfate) ranged from below the LOQ to 909 μg kg$^{-1}$ and are the only currently used OCPs, while endosulfan sulfate was the most frequently detected compound. The highest residue levels were identified in agricultural soils, followed by urban and rural soils. Chlordanes (cis- and trans-chlordane and trans-nonachlor) residue levels varied between 0.0033 μg kg$^{-1}$ and 2.7 μg kg$^{-1}$, while their highest concentration was found in agricultural soils. HCHs were only detected above the LOD in some agricultural and urban soils [188].

A survey was organized in the southeastern region of Buenos Aires province in Argentina to evaluate agricultural soils as a potential source of OCPs for the aquatic biota of an adjacent pond [189]. Ten soil samples were taken between July and October 1998 at an altitude of 80 m above sea level and at two horizons (0–30 cm and 45–60 cm) for the same sampling point for the determination of OCPs including p,p′-DDT, p,p′-DDE, p,p′-DDD, γ-HCH (LND), heptachlor, heptachlor epoxide, aldrin, dieldrin, and endrin. The analysis results revealed that the total OCP concentration was higher in the lower horizon than in the upper horizon. The allocation pattern of the concentration of the OCPs in the upper horizon was: LND > heptachlor > heptachlor epoxide and DDT. In the lower horizon, the pattern was: LND > DDT > DDE > DDD > aldrin > heptachlor and heptachlor epoxide. Thus, heptachlor was limited to the upper horizon, while heptachlor epoxide, LND, DDTs, and aldrin were concentrated in the lower horizon. LND (γ-HCH) was the only OCP being used at the time of the study, and its concentration was more than 40% of the total OCPs detected in both horizons. The heptachlor epoxide average concentration found in soil was 6.7 μg kg$^{-1}$ dw, while DDTs' highest concentration was 116.8 μg kg$^{-1}$ dw, and it was only detected in the lower horizon of the studied soil. This was explained by its extended residence time in soils that would permit it to reach the deeper soil layer by translocation with colloids [189].

Ninety soil samples were collected randomly (grid sampling) at root level, due to fears of the possible uptake of OCPs and PCBs, from three areas (A, B, and C) in Fort Albany (on the mainland), subarctic Ontario, Canada. Samples were collected from agricultural, residential/parkland, commercial, and industrial areas and analyzed among other pollutants for OCPs including DDTs (p,p′-DDT, o,p′-DDT, p,p′-DDD, o,p′-DDD, p,p′-DDE, and o,p′-DDE), HCB (β-HCB, γ-HCB, and δ-HCB), heptachlor, heptachlor epoxide isomer B, drins (aldrin, dieldrin, endrin, and endrin aldehyde), endosulfans (α-endosulfan, β-endosulfan, and endosulfan sulfate), and methoxychlor [23]. The concentration of DDTs in soil samples presented a heterogeneous distribution with concentrations that ranged from below the LOD to 4190 μg kg$^{-1}$.

Twenty surface soil samples (0–10 cm) collected in February 2005 from James Ross Island located in the southeast coast of the Antarctic Peninsula were collected for the determination of persistent organic pollutants including OCPs (α-HCH, β-HCH, γ-HCH, and δ-HCH), p,p′-DDE, p,p′-DDD, p,p′-DDT, HCB, and PeCB (pentachlorobenzene) [190]. The HCH concentration ranged between 0.51 and 3.68 μg kg$^{-1}$, DDT between 2.41 and 7.75 μg kg$^{-1}$, HCB from 0.59 to 2.24 μg kg$^{-1}$, and PeCB from 34.9 to 171 μg kg$^{-1}$. The occurrence of HCB in the soil designates that the long-range atmospheric transport is the most probable source of pollution in James Ross Island.

A study was conducted in 36 Alabama agricultural soils (experimental stations and private farms) to evaluate residues of formerly used OCPs including $\alpha$-HCH, $\gamma$-HCH, heptachlor, heptachlor epoxide, trans and cis-chlordane, dieldrin, p,p'-DDE, o,p'-DDE, p,p'-DDD, o,p'-DDD, p,p'-DDT, o,p'-DDT, trans-nonachlor, and toxaphenes [191]. The determined concentrations fluctuated by several orders of magnitude between farms and seemed to be log-normally distributed. Toxaphene and DDTs were demonstrated to have the highest average concentrations among other OCPs; thus, the toxaphene mean concentration was 285 µg kg$^{-1}$ dw, p,p'-DDE was 22.7 µg kg$^{-1}$, p,p'-DDT was 24.6 µg kg$^{-1}$, o,p'-DDT was 4.0 µg kg$^{-1}$, and p,p'-DDD was 2.4 µg kg$^{-1}$. The authors concluded that the determined residues were not proportionate to soil organic carbon content, suggesting that residues were an indication of historical pesticide applications. The DDT/DDE ratios in six regions of the state ranged from 0.39 to 1.5, and compound ratios for chlordanes and toxaphene were different from those in the technical mixtures.

*4.5. Overall Integration of the Studies Outcomes*

In most of the studies, statistical data such as detection frequencies, arithmetic and geometric mean concentration, and concentration ranges (minimum and maximum concentrations) were available for the detected OCPs. However, the compounds studied in each monitoring survey deviated significantly. Based on the results of the monitoring studies, HCHs, DDTs, and heptachlors were the most frequently detected OCPs in soils worldwide, more than 40 years after their use was banned. The outcomes of this research proved that OCPs' residues were present everywhere in the soil throughout the last several decades. This soil pollution can be due to direct exposure by their application on agricultural fields (illegal routes) or even from indirect routes via drift or runoff. Furthermore, their long half-lives have caused them to accumulate in soil historically. Examining the pesticide residues in the soil frequently is crucial to realize pesticides' fate and their occurrence in different compartments. What is evident in the examination of the OCPs in the soils is that there have not been adequate studies and programs dedicated to this issue, while the remediation of soil has not been attempted. European nations and Asian ones, to some extent, were more involved in soil pollution with OCPs, but in other parts of the world, one can easily observe an immense need for routine monitoring programs. Certainly, the future potential of soil monitoring research for OCPs is emphasized to fill the current differences in many countries.

## 5. World-Wide Data Combination

To the best of our knowledge, there is no complete study devoted to the global examination of OCPs' distribution and existence, principally in soils. In this study, all available arithmetic data from the reviewed 57 soil-monitoring studies from 28 countries around the world are presented in the section entitled 'OCPs' occurrence in agricultural soil'. The objective was to provide the reader with a broader feature on OCPs' occurrence in world soils. From individual studies, the data were considered at the level of individual substances and gathered over samples. Data such as the number of OCPs determined, the minimum and maximum concentration of total OCPs along with the total concentration of individual substances/metabolites, the arithmetic mean concentration, the detection frequency, the number of soil samples, and sampling profile were reported. Data analysis and processing are presented in Figure 4. The presence of each OCP alone does not designate soil pollution. To recognize the impairment of the situation, limits must be parleyed and authorized by experts who can help to designate the results from the aspect of environment contamination. Except for the existing Dutch limits for individual OCPs and the total OCPs of 0.4 mg kg$^{-1}$ dw. (400 µg kg$^{-1}$), in soil samples, limits in the JRC report of the ENSURE Action, and the Romanian limits for some individual OCPs [142], other limits could not be identified; thus, these limits were used to highlight the importance of the OCPs concentrations found globally. These limits were compared with the total OCP concentration, and the case of exceedance is presented. For hardly 15% of the studies, all

data were available (analytical and extraction method, LOD, LOQ, and total OCPs) at the same time. However, the available data were adequate to provide an overview of OCPs (Table S1). As can easily be seen, in some cases, fields were left empty due to a lack of initial data. The reported surveys indicated the distribution of OCPs in soil at national or regional levels, but the various sampling times and sampling depths, the numerous analytical protocols, and different OCPs listed amongst these surveys prevent a thorough overview of the distribution of OCPs in soils. However, it can be easily concluded that HCHs, DDTs, and heptachlors are the most frequently detected compounds.

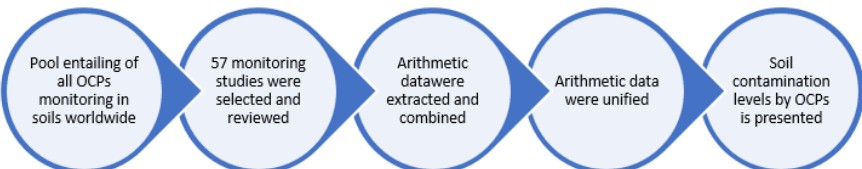

**Figure 4.** The concept of data analysis and processing of the available monitoring studies.

## 6. Conclusions

Due to their extensive usage in the past all over the world, the existence of OCPs in the natural environment is not surprising. In this article, all the available issued monitoring studies examining OCPs were revised, and all existing data in each study were reported. Due to variation in approaches used in each research for acquiring the data and variation in showing results, along with the differences in the number and the nature of the studied compounds, the reported results have been lacking valuable information. Despite all the work, the results encountered some inadequacies which should be taken into account while using it, such as variations in sampling methods, LOQs, extraction procedures, etc.

The results presented from all continents, except Oceania, proved soil contamination by OCPs during the last several decades. The upcoming potential of soil monitoring research for OCPs is emphasized to fill in the present gaps in many countries.

A cornerstone tool to address the residual prevalence of these contaminants is the implementation of powerful analytical methods. These, as presented in this extended review, vary from classical approaches using sensitive detectors such as ECD to the exploitation of advanced mass spectrometry, especially in its tandem mode, minimizing matrix interferences while increasing sensitivity. Yet, one critical step that should not be disregarded is the sample preparation, especially for cumbersome matrices such as soil. These approaches are also presented, and even though they are extensively elaborated, room for improvement when new techniques emerge cannot be excluded.

Monitoring pollution levels regularly is crucial in order to understand the occurrence and fate of OCPs in different environmental compartments. Soil pollution by OCPs should be an essential aspect in the characterization of whole soil quality. However, there is no legislation for tolerances or quality standards for pesticide residues in soil that accounts for possible impacts on soil biota in the broadest potential meaning. Regrettably, no sufficient soil security policies are yet in place to fight and reverse this hidden threat.

**Supplementary Materials:** The following supporting information can be downloaded at: https://www.mdpi.com/article/10.3390/agriculture12050728/s1, Table S1: Integrated data on available numerical results of OCPs from review articles.

**Author Contributions:** Conceptualization, H.K.; methodology, E.N.T. and H.K.; software, E.N.T. and H.K.; validation, E.N.T. and H.K.; formal analysis, E.N.T. and H.K.; investigation E.N.T. and H.K.; resources H.K.; data curation, E.N.T. and H.K.; writing—original draft preparation, E.N.T. and H.K.; writing—review and editing, E.N.T. and H.K.; visualization, E.N.T. and H.K.; supervision, H.K.; project administration, H.K.; funding acquisition, H.K. All authors have read and agreed to the published version of the manuscript.

**Funding:** This research received no external funding.

**Institutional Review Board Statement:** Not applicable.

**Informed Consent Statement:** Not applicable.

**Data Availability Statement:** Not applicable.

**Conflicts of Interest:** The authors declare no conflict of interest.

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
