# Peer review of "A Comprehensive Review of Organochlorine Pesticide Monitoring in Agricultural Soils: The Silent Threat of a Conventional Agricultural Past"

_agriculture, doi:10.3390/agriculture12050728_

Round 1

Reviewer 1 Report

Comments:

The manuscript entitled “A comprehensive review of organochlorine pesticides monitoring in agricultural soils. A Silent threat of a Conventional Agricultural Past” has so many grammatical and space errors that need to be corrected. Authors need to proofread the article thoroughly. In addition, this article has some ambiguous statements which do not fully depict the information provided. I suggest the following changes and improvements:

  1. What is the research significance and novelty of this work?
  2. Author should add a table or figure which depicts key findings.
  3. There is a need to add a Figure in the research methodology that shows the inclusion and exclusion criteria.
  4. It will be better if the author adds a Figure on the sampling method.
  5. Figure 1 needs to be modified and typo errors need to be removed.
  6. A comparison Table should be added on the global occurrence of OCPs' occurrence in agricultural soil.
  7. The conclusion section should be revised considering the key findings. Moreover, future recommendations should be proposed.

Author Response

  1. What is the research significance and novelty of this work?

We thank the reviewer for this comment. The significance and the novelty of this work is to provide the reader with a broader feature on OCPs occurrence in the world, the concentration levels of the individual substances/metabolites and to examine the pollution levels based on the existing limits. A new paragraph (World-wide data combination’ has been added to highlight it.

  1. Author should add a table or figure which depicts key findings.

We thank the reviewer for this constructive comment. A Table S1: Integrated data on available numerical results of OCPs from review articles, has been provided as supplementary material including all key findings.

  1. There is a need to add a Figure in the research methodology that shows the inclusion and exclusion criteria.

We thank the reviewer, for this constructive comment. A new paragraph (1.1 Research methodology-inclusion/exclusion criteria and a figure) and a figure (figure 1) has been added in the manuscript to explain the inclusion/exclusion criteria.

  1. It will be better if the author adds a Figure on the sampling method

Thank you for this comment. A new figure (figure 2: General Soil sampling strategy) has been incorporated in the manuscript to better describe the sampling methodology.

  1. Figure 1 needs to be modified and typo errors need to be removed.

Figure 1 (Figure 3 in the revised version) has been modified and typo errors has been removed.

  1. A comparison Table should be added on the global occurrence of OCPs' occurrence in agricultural soil.

We thank the reviewer for this constructive comment. A comparison Table, Table S1: as regards the global occurrence in agricultural soils of OCPs from review articles, has been provided as supplementary material including all key findings.

  1. The conclusion section should be revised considering the key findings. Moreover, future recommendations should be proposed.

The conclusion section has been revised accordingly. Thanks for this comment.

Reviewer 2 Report

Please see complete edits in attached manuscript.

However, the manuscript is well organised but it's would be better is metabolic pathowys of degradation products of OCPs were added. Also, effects of different types of soils might affect OCPs residues.

Author Response

  1. Please see complete edits in attached manuscript.

We thank the reviewer for the constructive comments. The manuscript has been revised according to the proposed comments by the reviewer (track changes).

  1. However, the manuscript is well organized but it would be better is metabolic pathways of degradation products of OCPs were added. Also, effects of different types of soils might affect OCPs residues.

We thank the reviewer for the constructive comments. Indeed, the metabolic pathways of OCPs degradation products and the effects of different soil types that may affect OCPs residues are two considerable factors involved in pesticides distribution. However, the purpose of the current review is to provide the reader with a broader feature on OCPs occurrence in the world, the concentration levels of the individual substances/metabolites and to examine the pollution levels based on the existing limits. In addition, this article reviews the analytical methods and sample preparation techniques for determining OCPs and some of their transformation products in soils.

Given the already extensive review we believe that the degrading behavior of OCPs in relation to the historical / current inputs of OCPs in (different type of) soils, may be deciphered in a separate study in the future.

Reviewer 3 Report

Organochlorine pesticides pose significant threats to human health and biodiversity due to their persistence, ubiquity, bioaccumulation and high toxicity to humans and non-target organisms. Research on OCPs has been widely pursued over the past 40-50 years. This manuscript summarized sampling methods, extraction techniques, analytical methods and occurrence of OCPs in agricultural soil on a global, but provided little new understanding or direction to study of OCPs in agricultural soil as the topic and the discussion is not novel. Thus, I would suggest the manuscript to be  revised. Specific comments are following:

1. The logic and organization within the section is chaos, especially for Scetion 4. OCPs' occurrence in agricultural soil. The authors listed the studies, but without any summarizing and reviewing.

2. The length is too long for the manuscript, but did not provide much new understanding for the research area. Please shorten the length. There are too many redundant introductions repeated in this manuscript, like the toxicity of OCPs, control of OCPs, etc. It is superfluous to dwell on these.

3. More tables and figures should be used to summarize the studies and make the statement clear and specific.

4. There are many writing mistakes in this manuscript. Follows are part of the mistakes in this manuscript:

Line 14: “over the last 30 years”.

Line 88: “have been”.

Line 97: “enough” here conflicts with “few” in the next row.

Line 163-164: “fit-for-purpose”.

Line 194: “, e.g.,”

Line 238: “p,p’-DDT” and other similar expressions should be revised.

Line 272: “the properties selected”.

Line 306: “freeze-drying”.

Line 325: “air-dried”.

Lines 326, 337, 381, 408, 489, 572 and many other places, the periods before citations are redundant.

Line 339: “initially”.

Line 345: “i.e.,”.

Line 456: “In recent years”.

Line 494: “studied”.

Line 529: “was determined”.

Line 580: “offers”.

Line 590: “from”.

Author Response

This manuscript summarized sampling methods, extraction techniques, analytical methods, and occurrence of OCPs in agricultural soil on a global but provided little new understanding or direction to study of OCPs in agricultural soil as the topic and the discussion is not novel. Thus, I would suggest the manuscript to be revised. Specific comments are following:

  1. The logic and organization within the section is chaos, especially for Scetion 4. OCPs' occurrence in agricultural soil. The authors listed the studies, but without any summarizing and reviewing.

We thank the reviewer for the constructive comment. A new comparison table has been added in the supplementary materials in order to summarize the key findings in each review article.

  1. The length is too long for the manuscript but did not provide much new understanding for the research area. Please shorten the length. There are too many redundant introductions repeated in this manuscript, like the toxicity of OCPs, control of OCPs, etc. It is superfluous to dwell on these.

We thank the reviewer for this valuable comment. The introduction section has been shortened in length.

  1. More tables and figures should be used to summarize the studies and make the statement clear and specific.

Thank you for this comment. New figures and a Table have been added to summarize the studies and to provide a clear statement on pollution levels.

  1. There are many writing mistakes in this manuscript. Follows are part of the mistakes in this manuscript:

All writing mistakes has been revised according to the reviewer proposal:

Line 14: “over the last 30 years”. The correction has been accepted.

Line 88: “have been”. The correction has been accepted.

Line 97: “enough” here conflicts with “few” in the next row. The correction has been accepted

Line 163-164: “fit-for-purpose”. The correction has been accepted

Line 194: “, e.g.,” The correction has been accepted

Line 238: “p,p’-DDT” and other similar expressions should be revised. The correction has been accepted

Line 272: “the properties selected”. The correction has been accepted

Line 306: “freeze-drying”. The correction has been accepted

Line 325: “air-dried”. The correction has been accepted

Lines 326, 337, 381, 408, 489, 572 and many other places, the periods before citations are redundant. The correction has been accepted

Line 339: “initially”. The correction has been accepted

Line 345: “i.e.,”. The correction has been accepted

Line 456: “In recent years”. The correction has been accepted

Line 494: “studied”. The correction has been accepted

Line 529: “was determined”. The correction has been accepted

Line 580: “offers”. The correction has been accepted

Line 590: “from”. The correction has been accepted

Round 2

Reviewer 3 Report

To be honest, I don't think the manuscript has been improved according to my comments, such the chaos of the logic, shortening the length. The logical problems are still there, the length is even longer. Besides these, there are many writing mistakes in this manuscript (listed below). The manuscript should be edited by someone with expertise in technical English editing, paying particular attention to English grammar, spelling, and sentence structure. Follows are part of the mistakes in this manuscript. Based on these issues, I have to suggest rejection of this manuscript!

Line 110: Figure title should be written below the figure.

Line 119: “HCH” and “BHC” are both abbreviation for hexachlorocyclohexane, and “BHC” appears for the first time here, and it’s full name should be given.

Line 178: “practice, ”.

Line 220: “p, p’-DDT”, and other similar expressions, which are not only in manuscript but also in supplementary information, should be revised.

Line 279: “physicochemical”.

Line 285: “it ignores”.

Line 285: “field”.

Line 288: “organized.”.

Line 289: “factors”.

Line 291: “relationships”.

Line 319: “periods”.

Line 285: In this figure, “Gather equipment” is followed with a strange icon.

Line 749: “extract the analytes of interest satisfactorily”.

Line 751: “origins”.

Line 766: “have”.

Line 789: “Besides Lesueur”.

Line 794: “Soxhlet”.

Line 798: “florisil,”.

Line 813: “solvent than the”.

Line 830: “In particular”.

Line 849: “was developed”.

Line 852: “was”.

Line 853: “compounds”.

Line 863: “residues in soil”.

Line 873: “require”.

Line 884: “determining”.

Line 891: “were reported”.

Line 1354: “were collected”.

Line 1355: “observed”.

Line 2285: “are presented”.

Line 2292: “processing are presented”.

Line 2303: “were left”.

Author Response

We thank the reviewer for his comments.

  1. The manuscript should be edited by someone with expertise in technical English editing, paying particular attention to English grammar, spelling, and sentence structure.

The manuscript has been revised properly and all grammar, spelling and sentence structure errors have been corrected.

Specifically:

Reviewer comment: Line 110: Figure title should be written below the figure.

The figure title has been moved below the figure.

Reviewer comment: Line 119: “HCH” and “BHC” are both abbreviation for hexachlorocyclohexane, and “BHC” appears for the first time here, and it’s full name should be given.

Thank you for this valuable comment. As HCH and BHC are both abbreviations for hexachlorocyclohexane, it was used the abbreviation HCH through the whole text to avoid confusion.

Reviewer comment: Line 178: “practice”.

The correction has been accepted.

Reviewer comment: Line 220: “p, p’-DDT”, and other similar expressions, which are not only in manuscript but also in supplementary information, should be revised.

This comment is not clear. The use of e.g. “p, p’-DDT” and similar expressions is used to differentiate the DDT isomers.

Reviewer comment: Line 279: “physicochemical”.

The correction has been accepted.

Reviewer comment: Line 285: “it ignores”.

The correction has been accepted.

Reviewer comment: Line 285: “field”.

The correction has been accepted.

Reviewer comment: Line 288: “organized”.

The correction has been accepted.

Reviewer comment: Line 289: “factors”.

The correction has been accepted.

Reviewer comment: Line 291: “relationships”.

The correction has been accepted.

Reviewer comment: Line 319: “periods”.

The correction has been accepted.

Reviewer comment: Line 285: In this figure, “Gather equipment” is followed with a strange icon.

The strange icon is an equipment which is used in the fields.

Reviewer comment: Line 749: “extract the analytes of interest satisfactorily”.

The correction has been accepted..

Reviewer comment: Line 751: “origins”.

The correction has been accepted.

Reviewer comment: Line 766: “have”.

The correction has been accepted.

Reviewer comment: Line 789: “Besides Lesueur”.

The correction has been accepted.

Reviewer comment: Line 794: “Soxhlet”.

The correction has been accepted.

Reviewer comment: Line 798: “florisil,”.

The correction has been accepted.

Reviewer comment: Line 813: “solvent than the”.

The correction has been accepted.

Reviewer comment: Line 830: “In particular”.

The correction has been accepted.

Reviewer comment: Line 849: “was developed”.

The correction has been accepted.

Reviewer comment: Line 852: “was”.

The correction has been accepted.

Reviewer comment: Line 853: “compounds”.

The correction has been accepted.

Reviewer comment: Line 863: “residues in soil”.

The correction has been accepted.

Reviewer comment: Line 873: “require”.

The correction has been accepted.

Reviewer comment: Line 884: “determining”.

The correction has been accepted.

Reviewer comment: Line 891: “were reported”.

The correction has been accepted.

Reviewer comment: Line 1354: “were collected”.

The correction has been accepted.

Reviewer comment: Line 1355: “observed”.

The correction has been accepted.

Reviewer comment: Line 2285: “are presented”.

The correction has been accepted.

Reviewer comment: Line 2292: “processing are presented”.

The correction has been accepted.

Reviewer comment: Line 2303: “were left”.

The correction has been accepted.
